# Acquiring new memories in neocortex of hippocampal-lesioned mice

Wenhan Luo [1,7], Di Yun[1,7], Yi Hu[1], Miaomiao Tian[1], Jiajun Yang[2], Yifan Xu[2], Yong Tang[3], Yang Zhan [3], Hong Xie[4,5] & Ji-Song Guan [1,6 ✉]

The hippocampus interacts with the neocortical network for memory retrieval and consolidation. Here, we found the lateral entorhinal cortex (LEC) modulates learning-induced cortical long-range gamma synchrony (20–40 Hz) in a hippocampal-dependent manner. The long-range gamma synchrony, which was coupled to the theta (7–10 Hz) rhythm and enhanced upon learning and recall, was mediated by inter-cortical projections from layer 5 neurons of the LEC to layer 2 neurons of the sensory and association cortices. Artificially induced cortical gamma synchrony across cortical areas improved memory encoding in hippocampal lesioned mice for originally hippocampal-dependent tasks. Mechanistically, we found that activities of cortical c-Fos labeled neurons, which showed egocentric map properties, were modulated by LEC-mediated gamma synchrony during memory recall, implicating a role of cortical synchrony to generate an integrative memory representation from disperse features. Our findings reveal the hippocampal mediated organization of cortical memories and suggest brain-machine interface approaches to improve cognitive function.

[1] School of Life Science and Technology, Shanghai Tech University, 201210 Shanghai, China. [2] School of Life Sciences, Tsinghua University, 100084 Beijing, China. [3] Brain Cognition and Brain Disease Institute, Shenzhen Institutes of Advanced Technology, Chinese Academy of Sciences, 518055 Shenzhen, China. [4] Institute of Photonic Chips, University of Shanghai for Science and Technology, 200093 Shanghai, China. [5] Centre for Artificial-Intelligence Nanophotonics, School of Optical-Electrical and Computer Engineering, University of Shanghai for Science and Technology, Shanghai, China. [6] CAS Center for Excellence in Brain Science and Intelligence Technology, Chinese Academy of Sciences, 200031 Shanghai, China. [7] These authors contributed equally: Wenhan Luo, Di Yun ✉email: guanjs@shanghaitech.edu.cn

Hippocampus (HPC) is believed to retain recent memories and gradually consolidate them into remote memories by interacting with the neocortex[1,2]. Hippocampus (HPC) is also involved in the retrieval of memories by reinstating patterns of cortical activity[3–5]. Conflicting hypotheses about how the interactions between HPC and neocortex contribute to memory storage and retrieval remain. The standard memory consolidation hypothesis proposes HPC transiently stores the memorized information[6], while the memory indexing theory argues that HPC only maintains the pointers to memories stored in the neocortex[7,8]. Decades of researches revealed that HPC and its adjacent regions, including subiculum, medial and lateral entorhinal cortex (LEC), are tightly connected brain structures. While the medial entorhinal cortex encodes spatial information[9], the LEC is critical to associated memory[10–12]. How the information is stored or modulated by the HPC and associated structures remains an open question.

Cortical activities are critical for learning and memory. For example, immediate early gene expression marked neurons are identified as elements of the memory engrams[13], which are responsive to a specific context. The reactivation of engram causally induces memory retrieval[14,15]. Memory engrams are also found in brain-wide cortical regions, including prefrontal cortex, retrosplenial cortex, and other cortices[4,16,17]. During memory formation and retrieval, a small population of neurons distributed in layer2 (L2) of neocortex showed engram-like response properties[16–18]. Such distributed cortical hubs may constitute a brain-wide neural network for memory storage[19].

Besides engram activities, oscillatory activities in neocortex have long been implicated in mnemonic functions. Oscillatory brain waves are engaged in high-level cognitive function[20–22] and may conduct communication functions across brain regions[23] with unknown neural mechanisms[24–26]. For example, cortical theta waves are phase synchronized to hippocampal theta and are related to the cortical gamma power[27,28]. In humans (note the difference of frequency definition, rodent: theta, 4–12 Hz, gamma, ~25–150 Hz; human, theta, 4–8 Hz, gamma, ~30–150 Hz), theta[29,30] and gamma[31] oscillations show large-scale phase synchronization, which is positively correlated to the performance in memory encoding[29] and retrieval[31]. The short firing delay between neurons during phase synchronization is proposed to strengthen their synaptic connections and to promote the interaction across various cortical regions for cognitive functions[32,33]. Recently, a report showed that multi-sensory stimuli at gamma frequency ameliorates cognitive function deficits in Alzheimer's disease (AD)[34]. However, it was still not clear whether the oscillatory activity causally engaged memory processing in neocortex. In this study, we discovered that upon learning and memory retrieval, long-range gamma synchrony was induced across multiple regions in neocortex, mediating mnemonic functions. While HPC is critical to modulate cortical synchrony during new memory encoding and retrieval, artificially inducing long-range gamma synchrony in neocortex alone restored the capacity of new memory formation and storage in HPC-lesioned animals. These data indicate that cortical networks alone, which are coordinated by long-range gamma synchrony, are able to store and retrieve hippocampal-dependent memories, such as spatial or contextual memories. Our observations suggest that the HPC-LEC complex modulates cortical memory units by providing synchronized oscillatory potentials to coordinate memories and implicate approaches to treat memory defective diseases.

## Results

### Learning-induced cortical long-range gamma synchrony is coupled to the theta rhythm.
First, we surveyed the multiregional cortical synchronization during a spatial memory task (Fig. 1). Mice were trained to forage buried food (reward) in a sandbox with glass blocks (place cues) for 4 days (Fig. 1a, Supplementary Fig. 1a). In recall trials (day 5), trained mice preferred to search and dig within the food zone, indicating successful memory retrieval (Fig. 1b). We recorded the local field potential (LFP) signals simultaneously from superficial layers of multiple cortices, including secondary motor cortex (MO, AP: −1 mm, ML: −0.7 mm, DV: −0.25 mm), primary visual cortex (VIS, AP: −3 mm, ML: −3 mm, DV: −0.25 mm) and retrosplenial cortex (RSC, AP: −3 mm, ML: −0.7 mm, DV: −0.25 mm, dorsal RSC) (Fig. 1c, Supplementary Fig. 1b) to study the coordination between motor, visual and spatial information related regions in this task. Inter-cortical projections have been identified between those cortical regions[35–37]. It has been reported that spatial navigation induces a strong elevation of theta LFP power in HPC[38,39]. Similarly, we found that the recorded regions showed a higher level of cortical theta (7–10 Hz) and gamma (low gamma: 20–40 Hz, high gamma: 60–80 Hz) oscillation power in the maze than that in homecage (Fig. 1d, e, Supplementary Fig. 1c). While, the cortical oscillatory powers were correlated to the animals' locomotion (Supplementary Fig. 1e), as reported in HPC[40], we found that learning specifically induced the coupling of gamma power to the theta phase during encoding and recall trials (Fig. 1f, Supplementary Fig. 3), akin to the results reported in HPC[41].

By subsampling LFPs to theta trough and calculating the synchrony (indicated by phase-locking value, PLV) between each cortical area of interest (Fig. 1c, see details in method), we found low gamma synchrony between each pair of the recording electrodes (pairwise PLV between MO, RSC, and VIS) became stronger during training and recall than that in the resting state. The cortical long-range gamma synchrony was coupled to theta phase (Fig. 1g, h, Supplementary Fig. 1d). In contrast to the power of theta and low gamma oscillation, the low gamma synchrony was not modulated by the locomotion speed in all conditions (Supplementary Fig. 1f), suggesting the cross-regional synchrony at gamma band might represent a moving-independent feature of the network dynamics in memory encoding and retrieval trials.

### HPC-LEC complex mediates memory associated long-range cortical gamma synchrony.
Because the HPC is closely involved in the modulation of theta waves in the brain[20,27,39], we asked if the cortical gamma oscillation during memory processing is modulated by HPC. In the loss-of-function test, we examined the behavioral and oscillation changes by a neurotoxic hippocampal lesion in the spatial memory task. When both the dorsal and ventral parts of HPC were damaged (Fig. 2a, b; Supplementary Fig. 2a, b), HPC-lesioned mice spent significantly longer time to find the food and learned much slower than the control group to obtain the food during the learning phase of the spatial memory task (Fig. 2c). In the probe trial to test memory retrieval, compared with the control group, HPC-lesioned mice did not prefer to search within the food zone (Fig. 2d). These results indicated a strong memory deficit in these mice.

In addition to the behavioral deficit, we observed increased theta and gamma power during memory encoding and recall, but the oscillation powers were substantially reduced compared to those in normal mice (Fig. 2e, Supplementary Fig. 2e). Prominently, learning-induced cortical theta-gamma coupling was abolished in HPC-lesioned mice (Figs. 1f vs. 2f, Supplementary Fig. 3 for all kinds of trials). Consistently, the coupling of phase synchrony was reduced significantly (Fig. 2g). The learning-induced cross-regional gamma synchrony vanished in those HPC-lesioned mice (Fig. 2h, i). Thereby, HPC is required for spatial memory and is essential for the modulation of learning-associated cortical oscillation.

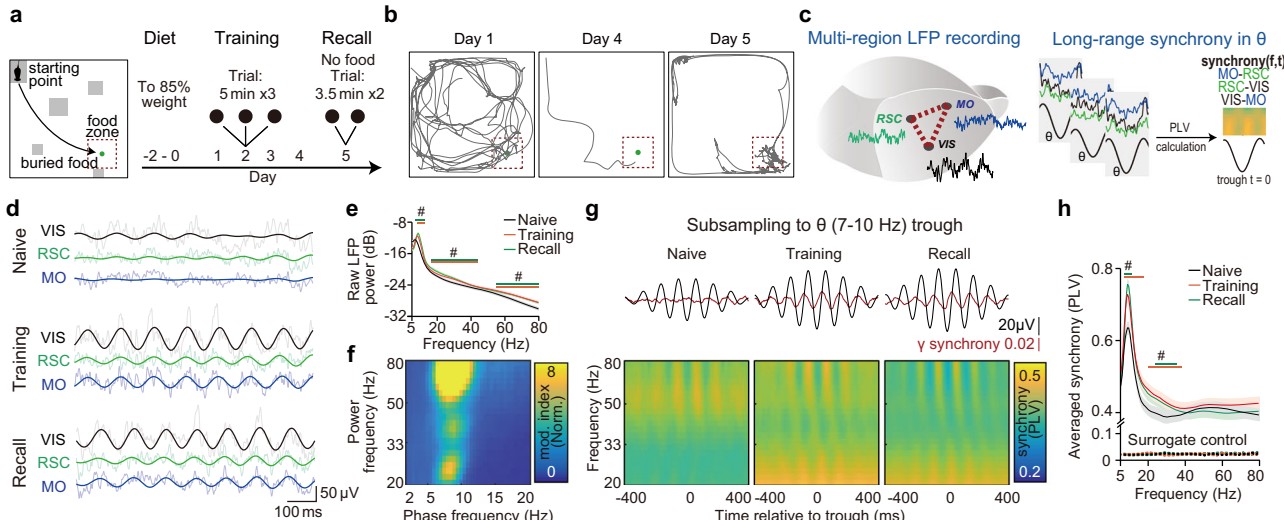

**Fig. 1 Learning-induced cortical long-range gamma synchrony is coupled to the theta rhythm. a** A schematic of the spatial memory task. Mice were trained to find the place of hidden food (buried in sands) for 4 days. Recall trials were set on day 5 without any food reward. The arena was designed with landmarks (gray blocks) and buried food (green dot). Mice start exploring from the starting point in all trials. **b** Representative trajectories of one mouse during learning and recall. **c** Simultaneously recording of LFPs from three distinct cortices during the task (left) and calculation of phase-synchrony spectrogram by subsampling LFPs to theta trough (right). **d** Representative LFP raw traces (tint lines) and filtered traces (theta (θ) band, 7–10 Hz, dark lines) from three recording sites at different stages of the task. Here training shows the data from the first day of training, same as the following graphs. See results for each region for each day in the supplementary Fig. 1c. **e, f** Theta and gamma power increased in training/recall trials and the gamma power was coupled to cortical theta phase (78 electrodes from 28 mice, $N_{MO}$ = 26, $N_{RSC}$ = 26, $N_{VIS}$ = 26). **e** Averaged raw cortical LFP power from three regions of different trials. Encoding showed here is data from training day1. Significance was assessed by two-way ANOVA followed by false discovery rate (FDR) corrected multiple comparisons at each frequency comparing with data of homecage trail. The trial factor but not the frequency factor was repeatedly measured. Statistical significant frequency range ($q < 0.05$) is noted on the graph in corresponding colors. **f** Phase-power modulation index comodulograms of training, averaged from all regions. (See comodulograms of each region for each day and corresponding quantification in Supplementary Fig. 3). Modulation index was normalized by the element-wise division of the raw comodulogram by surrogated control. **g, h** Theta and gamma synchrony elevated in training/recall trials and the gamma synchrony was coupled to cortical theta phase (72 electrode pairs from 28 mice. $N_{MO-RSC}$ = 24, $N_{RSC-VIS}$ = 24, $N_{MO-VIS}$ = 24). **g** Gamma synchrony was specifically coupled to theta phase during training and memory recall. Top, averaged theta wave (black) and 30 Hz cortical synchrony (red). Bottom, averaged phase(theta)-synchrony(gamma) spectrogram from all pairs. Each pixel of the spectrograms showed averaged synchrony of all pairs. **h** Comparison of averaged overall synchrony in three kinds of trials (See result for each pair for each day in the supplementary Fig. 1d. $q < 0.05$, FDR corrected, significantly changed frequencies were noted on the graph).# $q < 0.05$, Shadow of line plot shows S.E.M.

LEC is the gateway from the HPC to the cortex[42] and is also essential for memory encoding. To dissect the relationship between HPC, LEC and other cortical areas, we recorded electrical signals and calculated their synchronies in RSC, VIS, MO, LEC (AP: −4.35 mm, ML: −4.0 mm, DV: −4.0) and dorsal HPC (dHPC-CA1, AP, −2 mm; ML, −1.5 mm; DV, −1.3 mm) simultaneously during the spatial memory task (Supplementary Fig. 2j). While the theta power of LEC and dHPC (CA1 region) was significantly increased during learning, the cross-regional synchronies between LEC/dHPC and CTX (MO, RSC, and VIS) were detected at theta band (Supplementary Fig. 4). We found that the power of cortical oscillation (in MO, RSC, and VIS) at gamma band were coupled to the theta rhythm in LEC. However, gamma power in LEC did not couple to the cortical theta oscillation (Supplementary Fig. 5). Thus, the theta rhythm in LEC modulates gamma oscillation across multi-sensory cortices, but not in the reverse order. Moreover, the cross-correlation analysis[43] showed that theta rhythm in dHPC/LEC occurred earlier than that in cortical regions during learning (Fig. 2j). While HPC-LEC communicated via theta oscillation[44–46], we found that dHPC theta wave led theta waves in LEC (Fig. 2j, Supplementary Fig. 6).

To rule out the possible influence of volume conduction in our synchrony analysis, we directly induced a current sink in one of five brain regions (CA1, LEC, RSC, VIS, MO) and quantified the evoked amplitude (volume conduction) of potentials in other brain regions (Supplementary Fig. 7), we found that <1% of the source LFP could be detected in distant regions, so the volume conduction has a

negligible effect on the observed cortical synchrony phenomenon. These data suggest the learning-associated cortical long-range gamma synchrony is regulated by HPC.

**LEC layer 5 neurons project to widely distributed neuron ensembles in layer 2 of visual cortex and association cortices to induce synchronized gamma oscillations.** Next, we found that LEC projects to L2/3 of multiple cortical areas. We used virus approaches to drive the expression of EYFP in L5 Rbp4$^{Cre}$-derived neurons (Rbp4 is specifically expressed in L5 neurons) in LEC to trace their efferent fibers and found that LEC connected to many cortices including PTLp (posterior parietal cortex), ACC (Anterior cingulate cortex), SS (Somatosensory cortex), RSC, MO and VIS (Fig. 2k). Moreover, besides cortical axonal sprouting in L2/3, the anterograde trans-synaptic tagging in LEC also confirmed the projections from LEC L5 neurons to cortical L2/3 neurons. After injecting anterograde tracer virus (AAV2/1: hSyn-Cre) to trans-synaptically express Cre protein in the direct downstream neurons[47] of LEC in the reporter mouse line (Ai9 strain), we observed sparsely labeled neurons mainly located in L2/3 of the VIS, SS, MO and many other cortical regions (Supplementary Fig. 8). Hippocampus (HPC) does not have widespread projections to cortices. Both dorsal CA1 and ventral CA1 project to LEC deep layers[46,48,49]. Therefore, those observed LEC projections in numerous cortical areas, which are originated from deep LEC neurons to L2/3

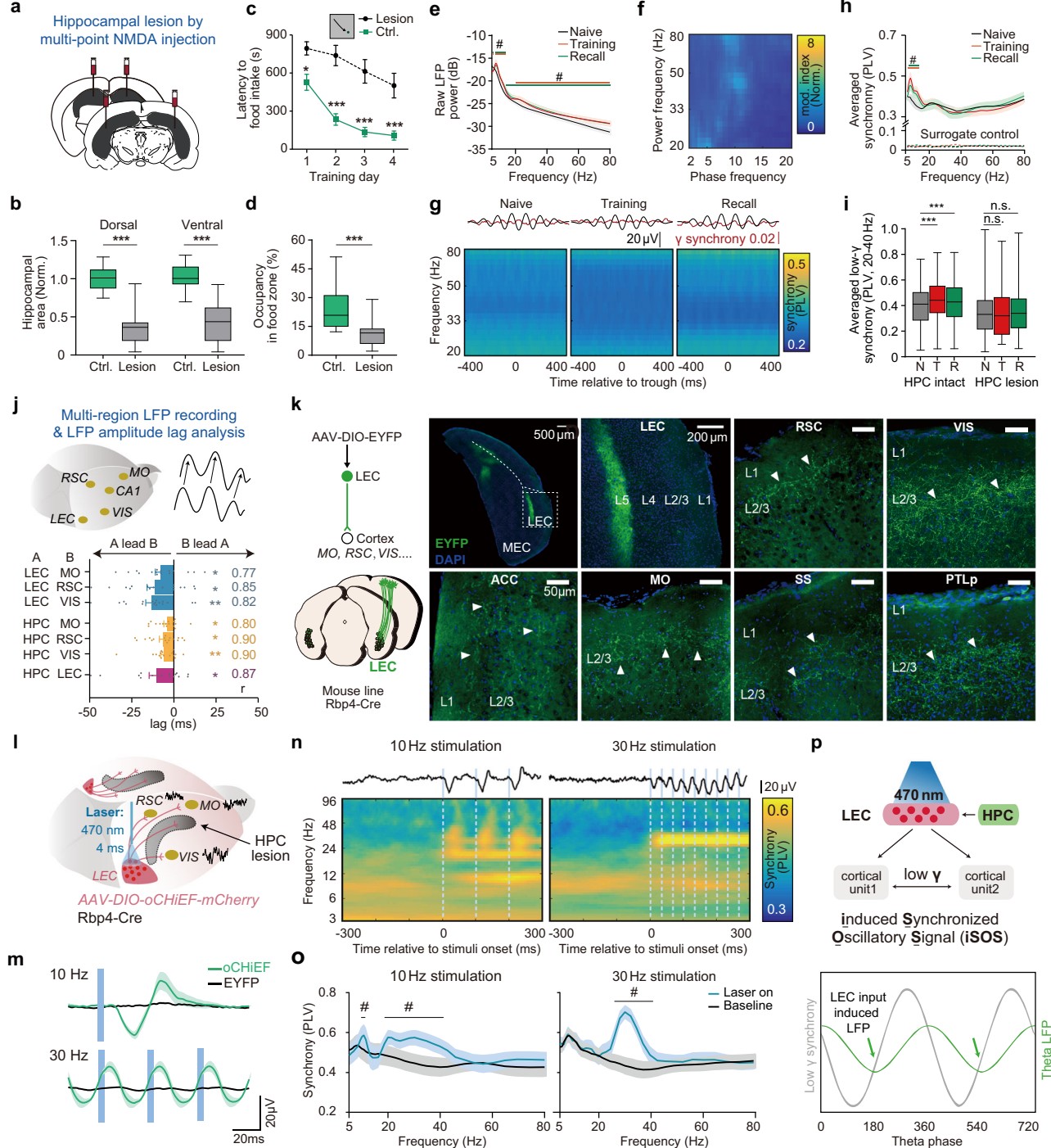

neurons, are fundamental for the signals from the HPC complex to reach cortical areas simultaneously. This anatomical structure implicates the role of LEC in mediating the HPC-modulated cortical synchronization.

To further test if the LEC L5 inter-cortical axons play a role in the long-range cortical synchrony for memory encoding and retrieval, we used virus approaches to drive the LEC expression of oCHiEF, which allows fast optogenetic control of neural activity[50]. We stimulated the LEC efferent fibers from the cortical surface to directly control the LEC targeted cortical neurons or stimulated the LEC cell bodies to test if it was able to affect cortical synchrony indirectly in HPC lesioned mice (Fig. 2l, Supplementary Fig. 9a). While HPC-lesioned mice were recorded in the homecage, LFP signals were collected simultaneously from

MO, VIS, and RSC. Activating the cell bodies of L5 LEC neurons or their efferent fibers induced reliable cortical LFP signals in recorded cortices (Fig. 2m, Supplementary Fig. 9b). Upon optogenetic activation in LEC (470 nm, 4 ms), LFP response latencies were similar across cortical regions (MO: 25.50 ± 1.87 ms, RSC: 25.25 ± 1.46 ms, VIS: 25.40 ± 2.11 ms, Supplementary Fig. 9d, e). The engaged actives showed significant bursts on low gamma band, no matter the frequency of the optic stimuli (Fig. 2n). Such LEC-mediated cortical oscillation in the low gamma band is an intrinsic property of each cortical unit, as each activation-induced LFP response lasted for ~36 ms (corresponding to low gamma frequency). Such phenomenon was observed by either activation of LEC-axons or activation of the L5 cell bodies in LEC (Fig. 2o). As a result of the simultaneous

**Fig. 2 HPC-dependent long-range cortical gamma synchrony is regulated by layer 5 LEC cortical projects and restored by iSOS in HPC-lesioned mice.**
**a, b** Effective bilateral hippocampal lesion in mice by multipoint NMDA injection. **a** Experimental scheme. **b** Quantification of the hippocampal residue size. ($N_{Control\ mice} = 14$; $N_{HPC-lesioned} = 15$; Dorsal: two-sided $t$-test, $t(27) = 7.9$; $P < 0.0001$; Ventral, two-sided $t$-test, $t(27) = 6.8$, $P < 0.0001$). **c** Learning curves show impairment of HPC-lesioned mice in the spatial memory task ($N_{Control\ mice} = 24$; $N_{HPC-lesioned} = 15$; ANOVA, Time factor: $F(3, 148) = 12.61$, $P < 0.0001$. Group factor: $F(1, 148) = 87.7$, $P < 0.0001$; Interaction, $F(3, 148) = 1.484$, $P = 0.2213$; Bonferroni post-hoc test, $P_{Day1} = 0.0111$, $P_{Day2} < 0.0001$, $P_{Day3} < 0.0001$, $P_{Day4} < 0.0001$). **d** HPC-lesioned mice did not prefer the food zone in the probe trials (same mice number as (**c**), two-sided $t$-test, $t(37) = 4.1$, $P = 0.0002$). **e, f** Theta and gamma power increased in training and recall trials while gamma power was no longer coupled to cortical theta phase in HPC-lesioned mice (35 electrodes from 12 mice, $N_{MO} = 11$, $N_{RSC} = 12$, $N_{VIS} = 12$). Here training shows the data from the first day of training, same as the following graphs. See results for each region for each day in the supplementary Fig. 2c. **e** Averaged raw cortical LFP power from three regions of different trials of HPC-lesioned mice ($q < 0.05$, FDR corrected, significant frequencies were noted on the graph). **f** Phase-power modulation index comodulograms of training of HPC-lesioned mice, averaged from all regions. **g, h** Neither the coupling between gamma synchrony and cortical theta phase (Fig. 1g) nor the elevation of long-range cortical synchrony (Fig. 1h) can be detected during training and memory recall in HPC-lesioned mice (34 electrode pairs from 12 mice, $N_{MO-RSC} = 11$, $N_{RSC-VIS} = 12$, $N_{MO-VIS} = 11$). **g** Cortical gamma synchrony no longer coupled to the cortical theta phase in HPC lesion mice. Top, averaged theta wave (black) and 30 Hz cortical synchrony (red). Bottom, averaged phase(theta)-synchrony(gamma) spectrogram from all pairs. **h** Comparison of averaged overall synchrony in three kinds of trials ($q < 0.05$, FDR corrected, significant frequencies were noted on the graph. See raw power, synchrony, comodulograms of each regions for each day and corresponding quantification in Supplementary Figs. 2–3). **i** Comparison of the overall low gamma synchrony between HPC intact mice and HPC lesion mice across the training process. N, naive; T, training day1; R, recall. (same data as Fig. 1g–h and Fig. 2g–h. Two-way ANOVA, $F_{Interaction}\ (2,208) = 9.56$, $P < 0.0001$, $F_{Lesion\ vs.\ Intact}\ (1,204) = 3.406$, $P = 0.0678$; $F_{Naive\ vs.\ Training\ vs.\ Recall}\ (2,208) = 5.705$, $P < 0.0001$. Bonferroni post-hoc test, HPC intact: $P_{Training\ vs.\ Naive} < 0.0001$, $P_{Recall\ vs.\ Naive} < 0.0001$). **j** Top, multi-region LFP recording and diagram for theta oscillation amplitude-based cross-correlation analysis. Bottom, dHPC theta lead LEC and LEC theta lead theta of cortical regions in training trials (training day1). Correlation coefficients of the max lag are noted on the right. (Wilcoxon signed-rank test, two-sided, compared to 0. $N_{LEC-MO} = 18$, $N_{LEC-RSC} = 18$, $N_{LEC-VIS} = 18$, $N_{dHPC-MO} = 17$, $N_{dHPC-RSC} = 17$, $N_{dHPC-VIS} = 17$, $N_{dHPC-LEC} = 16$ from 19 mice). See lag summary of other learning state in Supplementary Fig. 6. **k** Efferent axons from L5 neurons of LEC were detected in a wide range of cortical areas in L2/3, including, MO, RSC, VIS, SS, PTLp (Posterior parietal cortex) and ACC (Anterior cingulate cortex). Axons were labeled by EYFP via virus (AAV2/9-DIO-EYFP) injection in LEC of Rbp4[Cre] mice. Scale bar, 50 µm. (three mice brain were sectioned and show similar results, represenive images here are from one of them). **l–m** Artificial co-activation of LEC axons induced long-range cortical gamma synchrony. To detect the pure effect of optogenetic stimulation, cell body activation was done in homecage using awake HPC-lesioned mice to minimize the task-engaged cortical synchrony. **l** 470 nm laser-activated neurons expressing oCHiEF-mCherry in LEC (4 ms per pulse) induced synchronization oscillatory signals (iSOS) of LFP in multiple cortical areas simultaneously. **m** Blue light stimulation at either 10 Hz or 30 Hz in LEC could induce LFP responses simultaneously in MO, VIS and RSC in oCHiEF-expressing mice (green lines) but not EYFP expressing mice (oCHiEF group: $n = 21$ electrodes, including electrodes in RSC, MO, and VIS, EYFP group: $n = 14$ electrodes, including electrodes in RSC, MO, and VIS, lines are the averaged evoke potential of RSC, MO, and VIS). The evoked potential period (~36 ms, gamma frequency, ~30 Hz) did not change across stimuli frequencies. See surface axonal stimuli in Supplementary Fig. 9a–b. **n** Averaged cortical synchrony heatmap before and after stimulation. Top black lines, averaged LFP traces for each stimulation. Blue lines, laser stimuli. The spectrogram is the average of all cortical pairs including, RSC-MO, RSC-VIS, and VIS-MO. **o** Quantification of panel n. Baseline, averaged PLV before laser stimulation. Laser on, averaged PLV at 10–50 ms after each pulse, significant frequency ranges (lines above) lie at gamma band ($q < 0.05$, FDR corrected, 16 electrode pairs). **p** Illustration summarizes that long-range gamma synchrony is mediated by HPC-LEC and coupled to the theta-rhythm during memory encoding and retrieval (top). Endogenous synchronized LEC axonal activation may induce cortical synchronization oscillatory signals (iSOS), which is phase lock to theta phase, during memory encoding and retrieval (bottom). Shadow of lines and error bar shows lines show S.E.M. ***$P < 0.001$, #$q < 0.05$. For all box plot, whiskers show min and max, box shows 25th, median and 75th percentile.

activation for each cortex, long-range cortical synchrony was induced, especially at the low gamma band.

Consistent with this view, LEC stimuli at 25 Hz and 30 Hz induced a much stronger cross-regional cortical synchrony at the gamma band than that under the stimulus at 10 Hz (Fig. 2n, o; Supplementary Fig. 9f–i). The enhanced signal was probably due to the resonance. The LEC-stimuli-induced synchrony on the low gamma band reached the peak at ~30 ms, the time of which is similar to the delay time occurred in the learning period between theta trough and the peak of gamma synchrony (Supplementary Fig. 9j, k). These data suggested the stimuli-induced PLV rhythmicity was similar to endogenous PLV rhythmicity from theta modulation during memory encoding and retrieval. While HPC damage abolished the theta-coupled gamma synchrony in the neocortex, applying optogenetic stimuli on L5 LEC axon terminals was able to generate the induced-Synchronized-Oscillatory Signal (iSOS) to reestablish the long-range cortical synchrony in HPC-lesioned mice (Fig. 2p).

**Long-range cortical iSOS rescued memory deficits in HPC-lesioned mice for the contextual fear memory task.** Next, we asked if the LEC-mediated gamma synchrony in neocortex is functionally critical for the HPC-mediated contextual fear memory. In mice with bilateral hippocampal lesions, we applied artificial iSOS (470 nm, LED) through a cranial window ($d = 6$ mm) by activating the oCHiEF-expressing LEC axons in the upper layers of MO, VIS, SS, PTLp, and RSC (Fig. 3a) to simultaneously activate the circuits from LEC to the target cortices. After applying iSOS during learning trials to restore the long-range gamma synchrony (Fig. 2n, o), we tested the behavioral performance of mice in the contextual fear memory task (Fig. 3b). Activating LEC axons with iSOS did not affect locomotor activities, freezing, or exploring levels in those mice in the open field box (Fig. 3c; Supplementary Fig. 10a, b). In the contextual fear conditioning task after a weak foot-shock, while HPC-lesioned mice showed significantly reduced freezing, the iSOS groups of HPC-lesioned mice (10 Hz or 30 Hz, only in training trials) showed no impairment both in the immediate recall trial and in the 24 h recall trial (Fig. 3d, e). In the control experiment, rescued mice showed the same freezing level in a unfamiliar context as HPC intact mice, indicating rescued mice have the same memory fidelity as normal mice (Supplementary Fig. 10c–f). These data indicated that the iSOS application induced a full rescue of encoding and storage of long-term contextual fear memory in mice with hippocampal lesion.

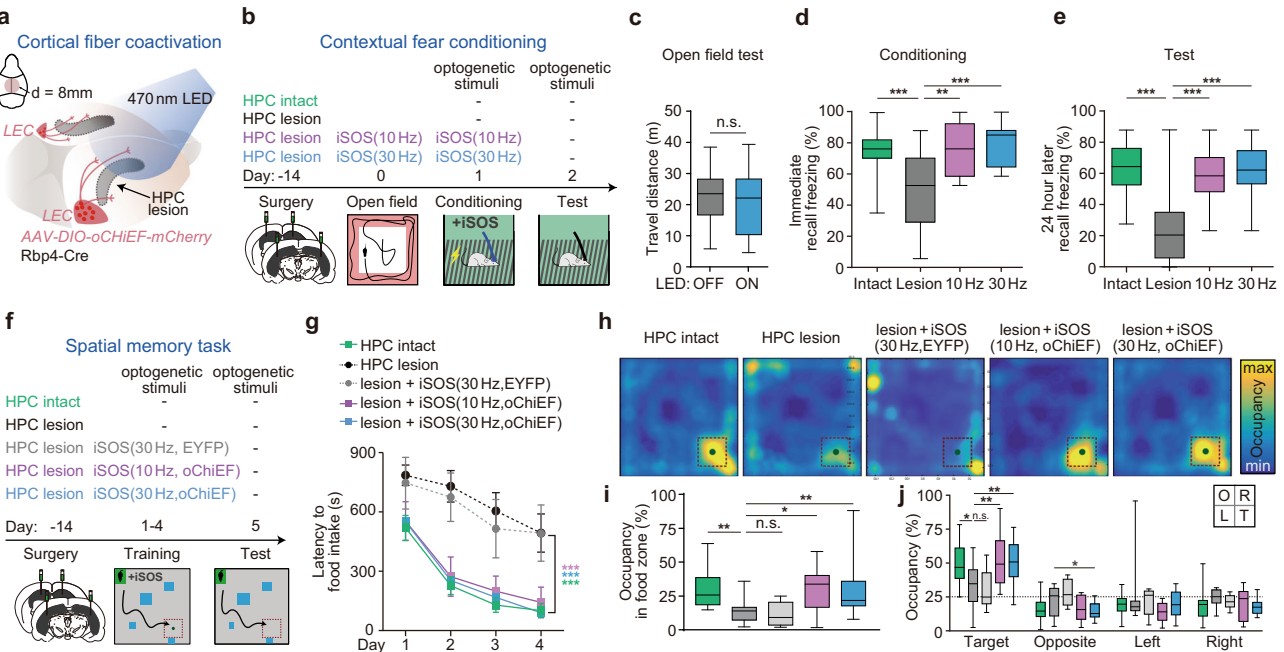

**Fig. 3 Cortical application of iSOS rescued memory deficits in HPC-lesioned mice. a–e** Artificial iSOS in neocortex during training can rescue fear memory deficits in HPC-lesioned mice. **a** Scheme of the co-activation LEC fiber during encoding. The iSOS were induced by LED on cortical surface to activate oCHiEF-expressing widespread axons from LEC L5. Activation of cortical fibers but not LEC cell bodies is to avoid the unspecific activation of circuits from LEC to other brain regions. **b** Flow of the behavioral experiment. All four groups of mice received contextual fear conditioning (CFC) training. Two groups of HPC-lesioned mice were given iSOS during training trials. Long-term memory was tested 24 h later without iSOS. HPC intact group means no HPC lesion, no iSOS applied and no virus infected. **c** Travel distance of open field test upon given iSOS or not. (HPC-lesioned mice, $N = 13$, two-sided paired $t$-test, $t(12) = 1.6$, $P = 0.1449$). **d** Immediate freezing after foot shock ($N_{HPC\ intact} = 15$, $N_{HPC\ lesion} = 19$, $N_{iSOS-10Hz} = 8$; $N_{iSOS-30Hz} = 8$; ANOVA, $F(3, 46) = 9.1$, $P < 0.0001$; Bonferroni post-hoc test, $P_{Control\ vs.\ HPC-lesion} = 0.0005$, $P_{iSOS-10Hz\ vs.\ HPC-lesion} = 0.0035$, $P_{iSOS-30Hz\ vs.\ HPC-lesion} = 0.0005$). **e** Memory test in the conditioned context (same mice as **d**, ANOVA, $F(3, 46) = 13.5$, $P < 0.0001$; Bonferroni post-hoc test, $P_{Control\ vs.\ HPC-lesion} < 0.0001$, $P_{iSOS-10Hz\ vs.\ HPC-lesion} < 0.0001$, $P_{iSOS-30Hz\ vs.\ HPC-lesion} < 0.0001$). **f–j** Artificial iSOS rescued spatial memory deficit in HPC-lesioned mice. **f** Flow of the behavioral experiment. **g** Learning curves for the spatial memory task (mice number: $N_{HPC-intact} = 24$, $N_{HPC-lesion} = 15$, $N_{iSOS\ (30Hz,EYFP)} = 6$, $N_{iSOS\ (10Hz,oCHiEF)} = 11$; $N_{iSOS\ (30Hz,oCHiEF)} = 22$; ANOVA, Time factor: $F(3, 219) = 52.5$, $P < 0.0001$; Group factor: $F(4, 73) = 10.6$, $P < 0.0001$; Interaction, $F(12, 219) = 1.4$, $P = 0.1453$; Bonferroni post-hoc test, $P_{HPC-intact\ vs.\ HPC-lesion} < 0.0001$, $P_{iSOS\ (30Hz,EYFP)\ vs.\ HPC-lesion} > 0.9999$, $P_{iSOS\ (10Hz,oCHiEF)\ vs.\ HPC-lesion} < 0.0001$, $P_{iSOS\ (30Hz,oCHiEF)\ vs.\ HPC-lesion} < 0.0001$). **h** Averaged occupancy maps for memory recall in day5. **i** Quantification of occupancy in food zone (same mice as **g**. ANOVA, $F(4, 73) = 5.0$, $P = 0.0012$; Bonferroni post-hoc test, $P_{HPC-intact\ vs.\ HPC-lesion} = 0.0048$, $P_{iSOS\ (30Hz,EYFP)\ vs.\ HPC-lesion} > 0.9999$, $P_{iSOS\ (10Hz,oCHiEF)\ vs.\ HPC-lesion} = 0.0243$, $P_{iSOS\ (30Hz,oCHiEF)\ vs.\ HPC-lesion} = 0.0098$). **j** Quantification of occupancy in four quadrants (same mice as **g** ANOVA for each quadrant, Target quadrant: $F(4, 73) = 5.0$, $P = 0.0013$; Bonferroni post-hoc test, $P_{HPC-intact\ vs.\ HPC-lesion} = 0.0159$, $P_{iSOS\ (30Hz,EYFP)\ vs.\ HPC-lesion} = 0.9803$, $P_{iSOS\ (10Hz,oCHiEF)\ vs.\ HPC-lesion} = 0.0102$, $P_{iSOS\ (30Hz,oCHiEF)\ vs.\ HPC-lesion} = 0.0070$, Opposite quadrant: $F(4, 73) = 4.3$, $P = 0.0038$, Bonferroni post-hoc test, $P_{iSOS\ (30Hz,oCHiEF)\ vs.\ HPC-lesion} = 0.0190$; Left quadrant: $F(4, 73) = 1.4$, $P = 0.2529$; Right quadrant: $F(4, 73) = 1.0$, $P = 0.4043$). *$P < 0.05$, **$P < 0.01$, ***$P < 0.001$. Each dot represents one mouse. Error bar shows lines shows S.E.M. For all box plot, whiskers show min and max, box shows 25th, median and 75th percentile.

**Long-range cortical iSOS rescued memory deficits in HPC-lesioned mice in the spatial memory task**. In the task to retrieve the food in a maze (Fig. 3f), HPC-lesioned mice showed significant learning impairment both in the initial phase of training and in the progressive learning days. The iSOS-induced groups of the HPC-lesioned mice (restoring the long-range gamma synchrony during the learning phase) showed a full capacity to retrieve food in term of learning speed and the short latency to reach the spot, similar to those in the HPC-intact group (Fig. 3g). In contrast, in a control experiment, no rescue effect was observed when applying iSOS to the HPC-lesion mice expressing EYFP (Fig. 3g). In the memory retrieval trials, when iSOS was not applied, the HPC-lesioned mice trained under iSOS application showed a strong preference to the food zone (Fig. 3h), similar to the HPC-intact group. While, HPC-lesioned mice showed a significant reduction of occupation time in the food zone (Fig. 3i), indicating long-range gamma synchrony during learning was critical for memory encoding and storage. HPC-lesioned mice after iSOS treatment spent more time in the target quadrant of the maze (Fig. 3j), suggesting they were able to form and retrieve the spatial information of the hidden food.

These results showed a complete recovery of HPC-lesioned mice to acquire and retrieve new spatial memories after engaging cortical long-range gamma synchrony during learning. Taken together, both the artificial iSOS and LEC axon-mediated long-range gamma synchrony are able to coordinate cortical units to store the contextual and spatial information in mouse neocortex, allowing a successful memory recall during the test trial.

**The cortical iSOS, but not asynchronous LEC axonal activation rescued spatial memory deficits in HPC-lesioned mice**. To ask whether the synchronization of cortical units or the activation of LEC axons is essential for the hippocampal-dependent cortical memory storage, we performed a double dissociation test. To this end, two groups of HPC-lesioned mice were trained sequentially either with the iSOS (30 Hz pulses by two synchronized laser beams) or asynchronous LEC axonal activation (two laser beams

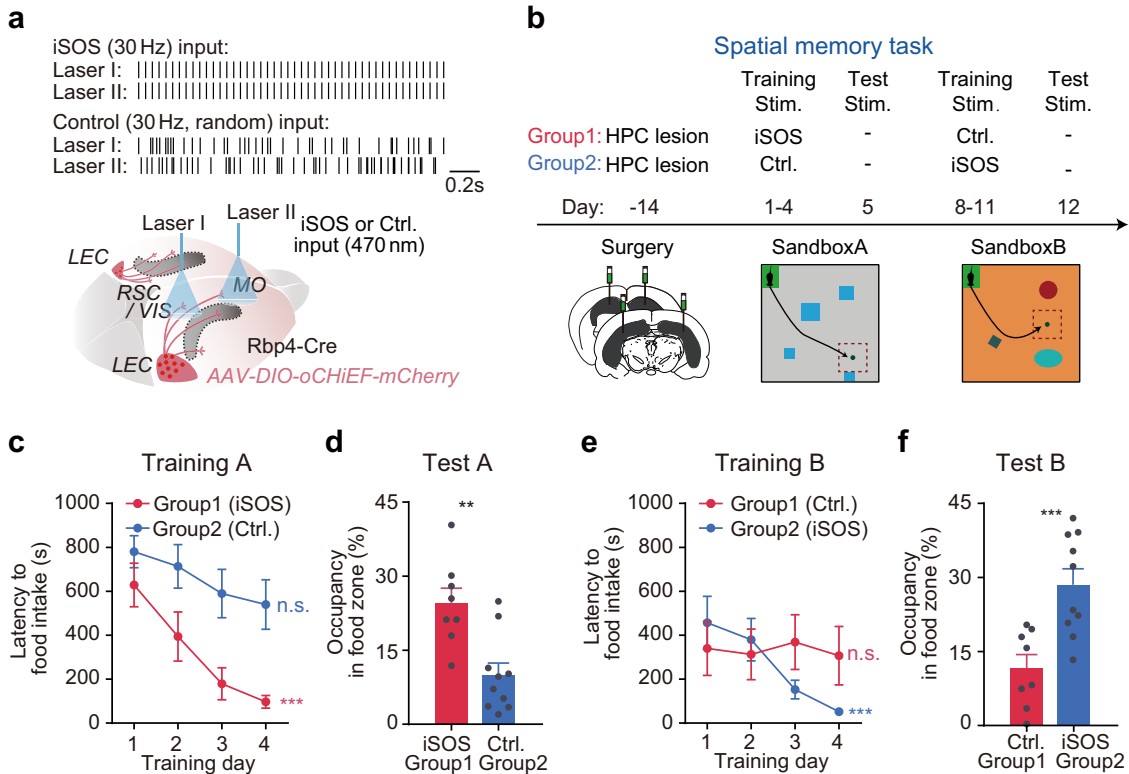

**Fig. 4 Application of iSOS, but not asynchronous cortical axonal activation rescued spatial memory deficits in HPC-lesioned mice. a, b** Scheme of the double dissociation experiment. **a** During training, LEC-axons in two cortical areas in HPC lesioned mice were activated by two independent fibers with either synchronized signal (iSOS$_{30Hz}$) or asynchronous signal (Ctrl. random pulses of each, same total pulse number). **b** HPC-lesioned mice were divided into two groups, both of which underwent the sequential maze tasks in two boxes in the same order. Group 1 mice were only applied with cortical iSOS during learning in the first maze and group 2 mice were only applied iSOS in the second maze. **c** Learning curves for two groups in CtxA. (N$_{Group1}$ = 8; N$_{Group2}$ = 10; ANOVA, Time factor: $F(3, 48)$ = 11.3, $P$ < 0.0001; Group factor: $F(1, 16)$ = 8.2, $P$ = 0.0113; Interaction, $F(3, 48)$ = 1.3, $P$ = 0.2976; Bonferroni post-hoc test, $P_{Group1:Day4\ vs.\ Day1}$ < 0.0001, $P_{Group2:\ Day4\ vs.\ Day1}$ = 0.0689). **d** Probe trials of the two groups in CtxA (two-sided t-test, $t(16)$ = 3.8, $P$ = 0.0017). **e** Learning curves for two groups in CtxB (ANOVA, Time factor: $F(3, 48)$ = 4.7, $P$ = 0.0056; Group factor: $F(1, 16)$ = 0.36, $P$ = 0.5582; Interaction, $F(3, 48)$ = 4.7, $P$ = 0.0062; Bonferroni post-hoc test, $P_{Group1:Day4\ vs.\ Day1}$ > 0.9999, $P_{Group2:\ Day4\ vs.\ Day1}$ < 0.0001). **f** Probe trials of the two groups in CtxB (two-sided t-test, $t(16)$ = 3.9, $P$ = 0.0013). Each dot represents one mouse. Error bar shows lines shows S.E.M.

with asynchronous stimuli on distinct areas). We used virus approaches to express oCHiEF in LEC L5 Rbp4[Cre]-derived neurons (Fig. 4a) and stimulated LEC-to-L2 axons on the cortical surface of MO and VIS (or RSC) during sequential learning for two place memory tasks in two distinct boxes (Fig. 4b, Supplementary Fig. 1a). In great contrast to the synchronized signal (iSOS), applying asynchronous laser stimuli on neocortex increased cortical LFP power (Supplementary Fig. 11a, b) and reduced the cross-regional synchrony (~10–100 Hz, Supplementary Fig. 11c, d).

In box A, when group 1 mice were given iSOS and group 2 were applied with the asynchronous signal, group 1 mice showed a quick reduction of the latency to retrieve the food during 4 days' training. In contrast, group 2 mice learnt much slower than group 1 (Fig. 4c). In the test trial of memory recall in box A, the group 1 mice spent significantly longer time searching in the food zone than that of the group 2 mice (Fig. 4d). Later on, in box B, group 2 mice, which now received the iSOS stimuli, improved their performance and significantly reduced their searching time (Fig. 4e). In contrast, group 1 mice, which received the asynchronous signal, did not show improvement in performance during the training period (Fig. 4e). In the probe trial in box B, group 2 mice spent significantly longer time in the food zone than that of the group 1 mice (Fig. 4f). Thus, the long-range cortical synchronization, rather than the LEC axonal activation per se, is essential for spatial memory storage.

Consistent to these observations, in the HPC-intact mice, we found that applying asynchronized signals in multiple cortical regions largely reduced the learning ability in the same task (Supplementary Fig. 11e–g) and mice showed impaired performance on the test trial of memory retrieval (Supplementary Fig. 11h). However, when the asynchronized signals were not given in the same group of mice in a second spatial memory task, their performance appeared normal in learning and memory retrieval, indicating that cortical synchrony is required for the formation of spatial memory in the brain.

Furthermore, as the artificially applied iSOS definitely did not contain any information about the context and the place clue in the task, the fact that those HPC-lesioned mice were still able to encode and retrieve memories about the spot of food location strongly implicates the fact that spatial information was stored and retrieved within the coordinated cortical network, at least in those HPC-lesioned mice. Thereby, given the fact that HPC was critical for memory-engaged cortical synchronies (Fig. 2), one of the essential roles of the HPC and associated structures in memory could be acting as the coordinator to allocate the memory storage in the distributed cortical networks.

**Long-range cortical gamma synchrony was tightly coupled to the activation of cortical memory-related neurons during memory retrieval.** How does the long-range cortical synchrony

modulate memory formation and retrieval? We simultaneously recorded calcium activities in cortical memory-related neurons (engram) in RSC and the LFP synchrony between RSC and VIS in the contextual fear conditioning task. We used virus approaches to express GCaMP6f, a calcium indicator, in cFos-Cre[ER] derived memory-related neurons in RSC by TRAP technology[51] and used optrode to simultaneously record the LFP and photometry of labeled neurons located in the superficial layer of cortex in free-moving mice (Supplementary Fig. 14a–d). Our previous in vivo imaging study suggest the superficial layer 2 neurons expressing immediate early gene encode context-dependent memories during learning trials[16]. Mice were injected with tamoxifen 24 h before learning to specifically induce the expression of GCaMP6f in neurons activated during training in context A (CtxA, Supplementary Fig. 12a, b).

The activity of labeled neurons in the same region increased robustly and repeatedly in the recall trials when back to the learned context (CtxA), compared to that in the unfamiliar context (CtxB) and in homecage (Supplementary Fig. 12c). in the test trial of memory retrieval, while neurons remained at low activities in the NO-freezing period, the frequency of the calcium events (CEs) in the labeled neuronal ensemble increased significantly at 2 s before the onset of the freezing behavior. The activities of engram neurons in VIS and RSC cortices in the pre-freezing period was significantly higher than those in the freezing period or in the post-freezing period (Supplementary Fig. 12d). We found that during this pre-freezing period, the rising time of calcium signals of labeled neurons of RSC was strongly correlated with the increase of gamma synchrony (Supplementary Fig. 12e, f). In contrast to c-Fos labeled engram neurons, activities of randomly labeled neurons of RSC did not show preference to the memorized context and were not coupled to the LFP synchrony between VIS and RSC (Supplementary Fig. 12g–i).

In another group of mice, we tested the correlation between activity of memory-related cells (labeled on the 4th day of training) and long-range synchrony in the spatial memory task. Similar to the pre-freezing period in the contextual fear task, a pre-exploring phase was set in the spatial memory task: mice were placed in the start region and constrained by a non-transparent door for 1 min before the door was open and mice were then allowed to explore and search for the hidden food (Fig. 5a). Consistent with the results of the contextual fear conditioning task, labeled cells in RSC and VIS showed a higher level of activities, especially during the pre-exploring period in the learned context than those in the unfamiliar context or the homecage conditions (Fig. 5b, Supplementary Fig. 13a). Only labeled neurons in the superficial layer (L2, within 200 μm to pia surface), but not deep-layer neurons (L4/5), showed context-specific activation during recall trials (Supplementary Fig. 14e–g). By cross-correlation analysis, comparing the whole time-series of low gamma synchrony with the activities of labeled neurons in RSC or VIS (Fig. 5c, d, Supplementary Fig. 13b, c), we found that such cross-correlation strongly occurred during the pre-exploring phase in recall trails, but not in the homecage trial. Those data indicate the activities of engrams were induced by memory retrieval, particularly in the free recall time when the sensory information of the learned context was not presented yet.

These observations were further confirmed by averaging synchrony near each calcium event. For the CEs in the pre-exploring and the exploring period, gamma synchrony showed peak events ahead of the peak of the CEs in all recorded cortical regions (RSC, Fig. 5e, VIS, Supplementary Fig. 13d). These data indicated that while cortical activities of labeled neurons were context-specific and distributed in different brain regions, long-range gamma synchrony was strongly coupled to the activation of the activities of memory-related neurons, especially during the retrieval of those memories.

**Cortical memory-related neurons in RSC and VIS exhibit egocentric coding to objects in the learned context.** In fact, to successfully retrieve the food, mice need to refer to a map. Studies have revealed neurons coding place information for allocentric maps in hippocampal formation, i.e., the place cells in HPC and grid cells in the medial part of EC[52]. Besides, recent discoveries showed neurons coding egocentric information in neocortex during spatial navigation. Such egocentric coding neurons have been detected in LEC[53], RSC[46], sensory and motor cortices[54,55]. We speculated that mice navigate in the context also with their egocentric map which may encoded in those c-fos labeled neurons (cortical engram). To this end, we re-analyzed the behavioral data together with the online activity of cortical engram to explore whether the activities encoded the spatial information in an egocentric way. For each recorded memory-related population, we constructed the egocentric object ratemaps (EORs) for each object in the context (see details in methods, supplemental movie 1). Here objects included both real entities and virtual ones, such as the food zone region (Fig. 5f). Particularly, taking the food zone as an example, for each of the detected CEs in the exploring phase of recall, the position of food zone relative to mouse head within 30.0 cm at the rising time point of the CE was built as the egocentric map for this CE. Superposition of egocentric maps for all CEs constructed a raw EOR, then EOR was normalized by dividing the raw EOR in each bin by the 'EOR' constructed using all time points during traveling (Fig. 5g). In over half of the labeled ensembles in RSC and VIS, we detected significant egocentric object sensitivity to at least for one object in the context (Fig. 5h, Supplementary Fig. 15a, RSC, $n = 4/8$; VIS, $n = 4/5$; in total, 61.5%), called egocentric object ensembles (EOEs). The most frequently responsive object of EOEs was the biggest glass block (block3, probably as the spatial cue in the context, $n = 4/13$) and the block close to the foodzone (block5, Supplementary Fig. 15a, b). We determined 15 significant EORs in the labeled population from eight mice. Most of the preferred distances were over 15 cm and the tuning angles fell on the contralateral side of the recorded region (Supplementary Fig. 15c, d), implicating they were generated via a sensory input-associated process.

In contrast to objects, the CEs did not show significant egocentric tuning to non-object regions (Supplementary Fig. 15e, f, see details in Method: "Egocentric Tuning to Non-object Regions"), suggesting the egocentric map were encoded via a sensory-driven process. Furthermore, relocation of the landmark objects in the spatial memory task impaired memory retrieval (Supplementary Fig. 16, see details in Method: "Memory Test of Landmark Relocation"), implicating that mice rely on the stored egocentric information to perform this spatial task.

Calcium activities of labeled neurons could show significant egocentric object sensitivity to more than one object, which leads to disperse allocation of spatial specificity, applying reported criteria (Mean Resultant Length, MRL was greater than the 99th percentile of the random distribution of resultants computed following repeated shifted CEs randomizations)[54] could result in a lower detection rate. Therefore, to compromise, we applied a loose significance threshold (MRL was greater than the 95th percentile of the random distribution). More than 60% of recorded ensembles were determined to be EOEs. When applying the reported criteria (99th percentile), still 30.7% of recorded ensembles were determined to be EOEs. The significance of egocentric tunings has been further validated by random simulation of the CE events in the moving density space

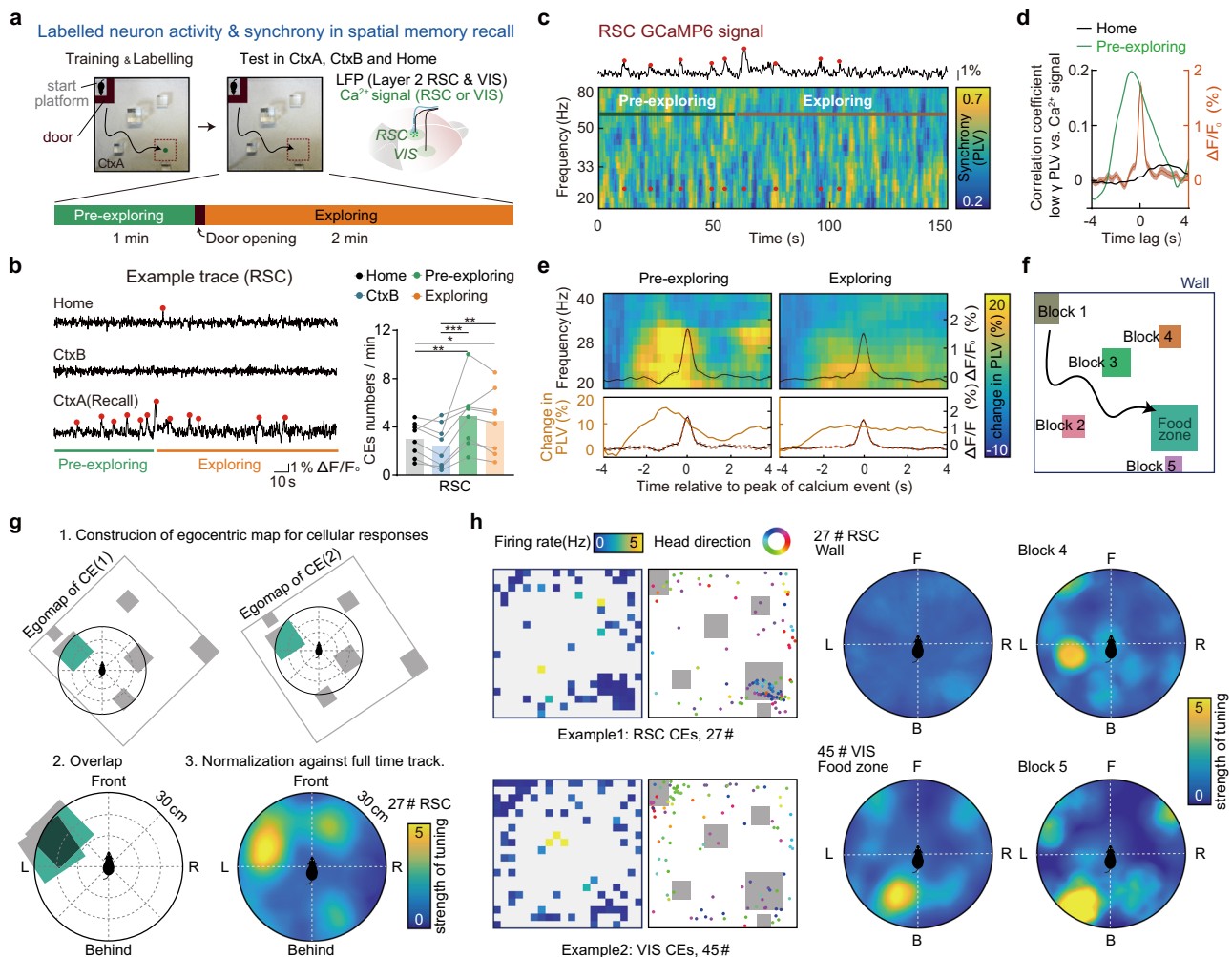

**Fig. 5 Activation of the cortical memory-related population was tightly coupled to the long-range cortical gamma synchrony and show egocentric coding to objects in the learned context during memory retrieval. a–e** Cortical gamma synchrony is associated with spatial memory-related neurons activation. **a** Experiment scheme to monitor the activity of cortical spatial memory-related neurons and the long-range gamma synchrony simultaneously. Activities of memory-related neurons were monitored by calcium signal (GCaMP6f labeled by TRAP using c-Fos-CreER mice during spatial memory task training day4) in RSC or VIS, while LFPs were recorded both in RSC and in VIS. In recall trials (day11), mice were confined on the start platform for 1 min by an L shape door (pre-exploring) before freely exploring in the sandbox (exploring). See other recording details in Supplementary Fig. 14a–d. **b** Left, examples of RSC calcium signals of labeled neurons. Right, labeled RSC neurons showed context selectivity. Red dots indicate the peaks of the detected RSC calcium events ($N = 8$ mice, ANOVA, $F(3, 15) = 7.1$, $P = 0.0034$, Tukery post-hoc test, $P_{Pre\text{-}exploring\ vs.\ CtxB} = 0.007$, $P_{Pre\text{-}exploring\ vs.\ Home} = 0.0308$, $P_{Exploring\ vs.\ CtxB} = 0.0257$). **c** An example of synchrony spectrogram (RSC-VIS) plotted together with calcium signal from labeled RSC neurons, showing the correspondence between engram activity and synchrony. **d** Cross-correlation analysis. Averaged correlation coefficient between low gamma synchrony and engram activity of RSC ($n = 6$) in pre-exploring phase (green) and in homecage (black). Orange line, averaged calcium events ($n = 263$ RSC CEs). **e** Averaged low gamma synchrony (RSC-VIS) spectrogram. Spectrograms are aligned to each peak ($t = 0$) of RSC calcium event during the pre-exploring phase or exploring phase. Synchrony was normalized to baseline synchrony of each calcium event (mean PLV from −4s to −3s). The black curve inside the graph shows the averaged curve of all calcium events within pre-freezing or freezing. Bottom, change in synchrony (averaged across low gamma band) and same averaged curve of CEs are plotted for clarity of peak time of the change in synchrony. 133 CEs of RSC within the pre-exploring phase and 263 CEs of RSC within exploring from 6 animals. (VIS calcium signal owns similar properties as RSC, both RSC and VIS signals showed most frequent firing in the pre-exploring phase and coupled to long-range low gamma synchrony. See VIS data in Supplementary Fig. 13). Dash lines plot around the CEs curve show S.D. **f–h** Labeled neurons in RSC and VIS show egocentric coding to objects in the learned context. **f** Diagram for definitions of objects in the spatial training context, including entity (walls, block 1–5) and virtual regions (food zone). **g** Schematic for construction of egocentric object ratemap (EOR) for a specific object, for example, the food zone. **h** 2D ratemaps of CEs, location of CEs plotted together with head directions (left), and EORs for two examples (right). Firing rate was calculated by total CEs numbers in each bin divided by total time spent in that bin. *$P < 0.05$, **$P < 0.01$, ***$P < 0.001$.

(Supplementary Fig. 15g, h, see details in Method: "Egocentric Significance of Random CE Simulation"). According to the statistical analysis, cortical activities of memory-related neurons reflected the egocentric map-like coding for each landmark within the context. Furthermore, coordination of those representations constituted the entire view about the context, forming an egocentric view of the path to the food pellet.

**LEC is critical for memory engram formation and its mediated long-range gamma synchrony is essential for cortical engram reactivation and memory retrieval.** Finally, we dissected the role of LEC in mediating the memory-retrieval associated with long-range gamma synchrony and retrieval-induced reactivation of memory-related neurons. We found that optogenetic inhibition of LEC-cortical axons impaired memory retrieval. Optogenetic

stimulating (589 nm LED) LEC-cortical axons expressing NpHR-EYFP on the surface of VIS, RSC, and SS induced less freezing in the light-on trials than that in the light-off trials in the conditioned context (Supplementary Fig. 17a–d). Inhibiting LEC axons in a single functional region (~0.5 mm²) did not alter the retrieval behavior (Supplementary Fig. 17e), implicating that loss of LEC-mediated synchrony in individual units of the cortex did not block the retrieval of contextual fear memory. Similarly, in the spatial memory task, when the LEC-L2 axons activities were inhibited during test trial, mice showed a significant reduction of their time spent in the target area in probe trials (Supplementary Fig. 17f–h), indicating LEC-mediated cortical synchrony is essential for the spatial memory retrieval.

Furthermore, in the spatial memory task, we expressed hM4D(Gi), a modified human M4 muscarinic receptor, to inhibit activities in LEC under the control of clozapine-N-oxide (CNO)[56], to inhibit activities in LEC under the CNO. Similar to the NpHR mediated inhibition in axon terminals, inhibiting LEC activities by hM4D(Gi) during memory retrieval impaired the preference of mice to the food zone in the recall trial and increased the latency for the mice to reach the food in the retraining trials (Fig. 6a–c). Consistent with the behavioral deficits, inhibition of LEC activity also reduced the frequency of cortical activities of labeled neurons in the pre-exploring phase of recall trials (Fig. 6d, e). Moreover, in the pre-exploring phase, LEC-inhibition induced reduction of reactivation of labeled neurons (Fig. 6e). Consistently, LEC-inhibition induced reduction of VIS-RSC long-range gamma synchrony and triggered a complete abolishment of the coupling between each of the CEs and cortical synchrony (Fig. 6f–h), when aligning the LFP signals to each of the engram calcium event. By correlation analysis, comparing the whole time-series of gamma synchrony with the calcium activities of labeled neurons in RSC and VIS, we found that LEC-inhibition totally abolished the correlation between engram activity and long-range gamma synchrony in the pre-exploring phase (Fig. 6i).

Moreover, we investigated whether the egocentric cortical representation is influenced by LEC during retrieval. We found that the number of objects which were encoded in an egocentric manner was decreased after LEC inhibition (Supplementary Fig. 18a, b), and egocentric maps of those labeled cortical neurons were more dispersive when LEC was inhibited by CNO (Supplementary Fig. 18c). This data suggests that the LEC-mediated cortical gamma synchrony is required during memory retrieval for the precision coding of the egocentric map.

We found LEC activity is essential to the formation of spatial memory. We inhibited part of the LEC activity (as the hM4Di were not detected in all the LEC neurons) throughout all the training days of the spatial memory task, and mice showed significantly reduced performance in recall trials (Supplementary Fig. 19, see details in Method: "LEC Inhibition During Spatial Memory Encoding"). We also labeled the learning activated cortical neurons on day 4 and found that they were not activated much in the recall trial, indicating the LEC activity was essential for the formation of context-selective cortical neurons (Supplementary Fig. 19d). Our data indicated that LEC-mediated gamma synchrony is critical for both memory encoding and engram reactivation during memory retrieval. While engram activities were representing the "egocentric" map of each landmark in individual cortical regions, LEC (HPC complex)-mediated cortical long-range synchrony integrated them to form an intact egocentric representation of the behavioral context. Thereby, HPC-LEC activities are serving as a coordinator to allocated individual cortical memory units for contextual and spatial memory storage (Fig. 6j).

## Discussion

The gamma synchrony is associated with cognitive function in neocortex, especially for sensory processing and associative learning[34,45,57,58]. Our data revealed the long-range gamma synchrony is mediated by axons from LEC neurons, which project to layer 2 of cortical regions, including RSC, VIS, SS, and MO, and synchronize activities between each region for memory storage. While memory engrams in individual cortical regions encode egocentric maps of dispersed features, successful encoding and retrieval of memories require the LEC-mediated gamma synchrony between multiple cortical regions to generate an integrative memory representation. Thereby, the long-range gamma synchrony, which is modulated by hippocampal theta rhythm could serve as the coordinator to organize the collective memory representation in neocortex (Fig. 6j). Our studies implicate distinct roles of two brain structures in memory processing: the cortical networks embed memorized information, while hippocampal-associated structures engage the coordination of the cortical networks to access or modify specific memories.

Long-range phase synchronization has been identified in human[31]. Such a conserved phenomenon might play critical roles in the brain. The synchronization coordinates the timing of neuronal firing in functional distinct areas[59,60], thus refining the long-range connections between cortical areas[23,61,62]. Alternatively, oscillation activities might directly affect engram activities via modulation on oscillatory frequencies[63]. Dominant frequencies of cortical LFP are related to the brain states and found to be essential for state-dependent memory retrieval[64]. Such LFP events might also contribute to system consolidation of the long-term memory[65]. Consistent with these reports, our observations indicated a strong linkage between the cortical oscillations and the memory processes, especially with the L2 engram activities.

Besides the learning-associated long-range synchrony, our data revealed neural mechanisms that memory retrieval is closely associated with the reactivation of cortical memory engrams in layer 2. While studies indicated the artificial reactivation of those memory engrams induces the retrieval of specific memories[17,65], we further demonstrated that activities of the cFos-labeled neurons, which were recruited during the learning phase, encode egocentric place information in the behavioral context. In contrast to those detected in the single-unit recording, the photometry recording revealed a populational egocentric coding specific in the labeled population (15–20 neurons under the tip). Remarkably, activation of those neurons was detected during the pre-exploring period, when no landmark cues were presented. It is likely reflecting the active memory retrieval of the spatial information as inhibition of LEC/HPC complex, which impairs memory retrieval, significantly reduced cortical engram activities before the door opening.

Our discovery on the HPC-regulated distributive cortical engrams shines a light on the memory index theory, which proposes that memory is stored in distributed cortical and subcortical modules, while HPC registers extrahippocampal network connectivity patterns[66–68]. We found HPC/LEC do regulate activities of engrams in multiple cortices via providing gamma synchrony, suggesting the traveling waves in the brain and their synchrony could take a role as the index to register cortical memory units. These observations are also consistent with a recent report that Tanaka and colleagues[15] found the hippocampal CA1 engrams encode the behavior context rather than the specific place information and were coupled to theta rhythm. It is interesting to speculate that allocentric spatial map in place cells and egocentric information coupling to the hippocampal engrams might interact in CA1 to provide integrative information for spatial memory retrieval. In addition, the memory index theory suggests the HPC

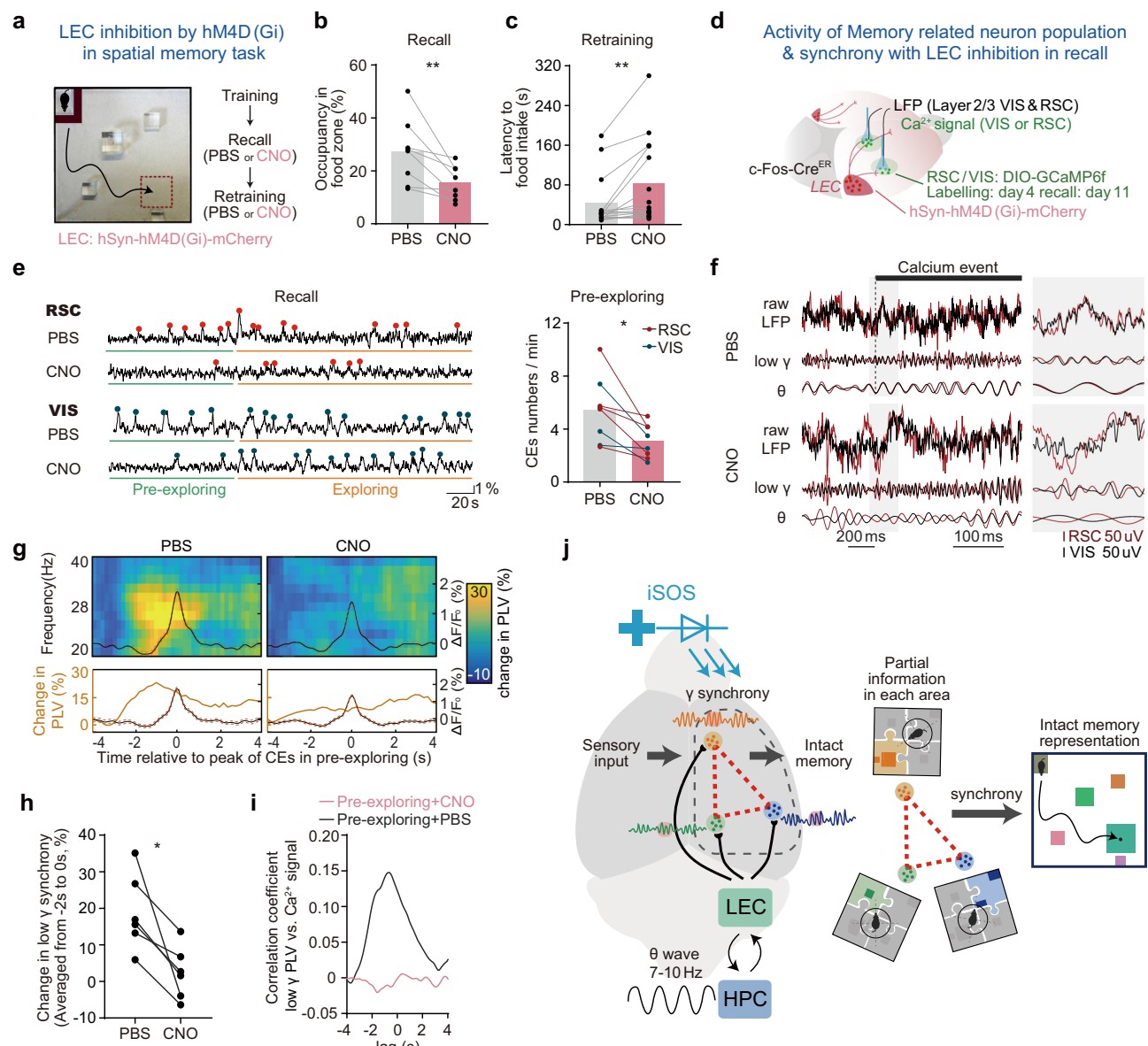

**Fig. 6 LEC-mediated long-range gamma synchrony is essential for cortical memory-related neurons reactivation and memory retrieval in HPC-intact mice. a** Experimental scheme for LEC inhibition by hM4D(Gi) in spatial memory recall and retraining to test if LEC is essential for successful memory recall. **b–c** Inhibition of LEC activity impair memory recall. **b** Occupancy in the food zone significantly decreased in recall trail ($N = 15$ mice, two-sided paired $t$-test, $t(14) = 3.5$, $P = 0.0036$). **c** Latency to the reward in the following retraining trail increased ($N = 15$ mice, two-sided paired $t$-test, $t(14) = 3.2$, $P = 0.0067$), CNO dosage: 2 mg/kg. **d** Similar to Fig. 5a, experimental scheme shows calcium signals from labeled neurons (labeled in training day4) and LFPs were recorded simultaneously during memory recall with or without LEC inhibition. **e** Left, calcium signal examples of labeled neurons in RSC and VIS with or without LEC inhibition. Right, quantification of CEs frequency in pre-exploring phase of memory recall (N total = 8 mice with three recorded in VIS and 5 in RSC, two-sided paired $t$-test, $t(7) = 3.5$, $P = 0.0105$). Red and blue dots indicate the peaks of the detected RSC and VIS calcium events, respectively. **f** Examples of LFPs around calcium events from the pre-exploring phase of recall trials with PBS and with CNO, the black bar above LFPs indicate the time range from rising time to the peak of the calcium event, illustrating theta ($\theta$) and gamma ($\gamma$) LFPs transiently synchronized with zero-phase lag at the rising time of the calcium event while it is not the case in CNO administration recall trial. **g** Averaged synchrony (RSC-VIS) spectrogram, spectrograms were aligned to each peak of RSC or VIS calcium events in the pre-exploring phase during memory recall with or without LEC inhibition. ($N_{PBS} = 90$ CEs, $N_{CNO} = 75$ CEs, from six animals). **h** Change in low gamma synchrony in panel g (averaged PLV from −2s to 0 s) was quantified for each mouse. LEC inhibition impaired the coupling between engram activity and cortical low gamma synchronization (N total = 6 mice, paired two-sided $t$-test, $t(5) = 3.541$, $P = 0.0165$). **i** Cross-correlation analysis. Averaged correlation coefficient between low gamma synchrony and RSC or VIS engram activity in pre-exploring phase with CNO administration (red, $N = 6$ mice) or not (black, $N = 6$ mice). *$P < 0.05$. **j** Model for the mechanism in which long-range cortical gamma synchrony mediates memory encoding and retrieval by HPC/LEC. The illustration shows that the cortical long-range gamma synchrony is coupled to the theta wave and engram activities. Such a process might underlie hippocampus-mediated memory encoding and recall in the neocortex in a highly coordinated way. Engram coding partial context information in each cortical region was integrated by gamma synchronization and this integration contribute to intact memory representations.

might not retain and deliver memorized information, but keep the internal index to access that information. This theory is consistent with our finding that artificially imposed iSOS signal is able to encode and retrieve memory, probably via providing indexing role to the cortical memory units.

Our study suggests that brain machine interface (BMI) devices could improve cognitive function in health.. Studies have demonstrated memory enhancement through deep brain stimulation[69] or transcranial direct current stimulation (tDCS)[70]. However, the stimulation settings are highly variable with electrode placements, stimulation waveforms, and spatial-temporal scales[71]. Our biological insight into the memory process might inspire BMI designs targeting LEC and proposed specific stimulation protocols with the gamma synchrony. On the other hand, while decoding neural oscillation signals usually employ power or single units[72], instantaneous large-scale multi-pair cortical phase synchrony could be used to decode neural information via BMI equipment. We propose that the revealed biological mechanism of memory encoding and retrieval could improve the efficiency of BMI for future technologies.

## Methods

**Mouse subjects**. The Laboratory Animal Facility at the Tsinghua University and ShanghaiTech University are accredited by Association for Assessment and Accreditation of Laboratory Animal Care International. Wild type (C57BL/6), Rbp4-Cre (Tg-KL100Gsat), Ai9 (007909, The Jackson Laboratory), and cFos-CreER mice (021882, The Jackson Laboratory) were utilized in experiments. All animals were socially housed in a 14 h/10 h (7 a.m.–9 p.m.) light/dark cycle, with food and water *ad libitum*. All experimental animals were 3–5 months old (22–35 g) and male. After surgery, they were housed individually in homecage in a humidity- and temperature-controlled environment. The mice recovered from surgery for at least 1 week before all behavioral tasks. All animal protocols and experiments and were evaluated and approved by the Institutional Animal Care and Use Committee at Tsinghua University (license 15-GJS1) or ShanghaiTech University (license 20201218002) based on Guide for the Care and Use of Laboratory Animals (Eighth Edition), and conducted in agreement with Chinese law (Laboratory animal -Guideline for ethical review of animal welfare, GB/T 35892). This study complies with all relevant ethical regulations animal testing and research, and received ethical approval from Laboratory Animal Resources Center at Tsinghua university and from Scientific Research Ethics Committee at ShanghaiTech University.

**Electrode implantation and stereotaxic injection**. Recording electrodes were custom made by platinum-iridium (90–10%) wires (coat: PTFE diameter, 33.020 μm, A-M system, 775003) and tungsten (coat: PTFE diameter, 105 μm, A-M system, 795500). Only electrodes with an impedance lower than 2 MΩ were employed. During the implantation surgery, animals were anesthetized with isoflurane. The speed of airflow was kept at 1.2 L/min with 1.5% (v/v) isoflurane. Local sterilizing (75% alcohol) was applied to the skin before making the incision. Holes were drilled and optrodes were implanted in the retrosplenial cortex (RSC, AP: −3 mm, ML: −0.7 mm, DV: −0.25 mm), secondary motor cortex (M2, AP: −1 mm, ML: −0.7 mm, DV: −0.25 mm), primary visual cortex (V1, AP: −3 mm, ML: −3 mm, DV: −0.25 mm), CA1(AP: −2.0 mm, ML: −1.5 mm; DV: −1.3 mm) and LEC (AP: −4.35 mm, ML: −4.0 mm, DV: −4.0 mm relative to the bregma). Four screws were inserted above the cerebellum, two of them were used as reference and the other two ground electrodes were used as ground. Two references were interconnected and isopotential, two ground screws were also interconnected and isopotential. The implantation was secured by black dental cement. In activation/ inhibition of LEC neuron fiber/cell body experiment, AAV2/9-DIO-NpHR-EYFP, AAV2/9-DIO-oChiEF-mCherry or AAV2/9-hSyn-hM4D(Gi)-mCherry was injected into LEC region (1 μL, 0.1 μL/min, AP: −4.35 mm, ML: −4 mm, DV: −3.85 mm relative to the bregma). Optical fiber (d = 200 μm, N.A. = 0.48) was implanted in LEC (AP: −4.35 mm, ML: −4 mm, DV: −3.65 mm relative to the bregma).

**Cranial window opening**. Cranial windows were opened for shining 470 nm excitation light by LED, mice were immobilized in custom-built stage-mounted ear bars and a nosepiece. Mice were anesthetized with 1.5% isoflurane. A 1.5 cm incision was made between the ears, and the scalp was removed to expose the skull. One circular craniotomy (6–7 mm in diameter) was made using a high-speed drill and a dissecting microscope for visualization. A glass-made coverslip was attached to the skull. A 3D printed headset (for LED holding during the experiment) was attached on top of the coverslip.

**Hippocampal lesion**. Mice were anesthetized with 1.5% isoflurane and placed into a stereotaxic frame. A midline incision was made on the scalp, the skin was

removed and the skull overlying the targeted region was removed. Injections of NMDA (10 mg/ml, dissolved in PBS) were injected to induce hyperactivity for killing cells in the target region (Dorsal: AP, −1.5 mm; ML, −2 mm; DV, −2 mm; Ventral: AP, −3 mm; ML, −2.5 mm; DV, −3 mm). Injection of 0.25 μl was given in 2 min at each site. Mice show seizure behavior and abnormal mobility after NMDA injection, as the evaluation whether the NMDA was injected successfully. Once the mouse show seizure behavior, it was put back under anesthesia for 3 h and then recover in the homecage for 2 weeks. The learning and memory behavioral experiment was performed after recovery.

**Open field test**. To test whether the activation or inhibition of LEC cortical fiber will cause some side effects in mice, the open field test was performed upon 470 nm or 589 nm LED stimulation. Mice were placed in the center of an open-field apparatus (46 cm wide × 46 cm long × 40 cm high) and were allowed to move freely for 12 min (LED: OFF → ON → OFF → ON, 3 min each section) after a 5 min habituation. Mice activities were video recorded. Total travel distance and other parameters were analyzed by a computerized mouse tracking system (MATLAB).

**Spatial memory task**. Before training, mice were put on a food deprivation regimen that kept them at ~85% of their free-feeding body weight. Mice were put into the box to forage for sunflower seeds as food reward in the TSE system (Multi-conditioning system, TSE system). The sandboxes A and B were the same sizes as the mentioned open-field context (46 cm × 46 cm × 40 cm) but with different objects and foodzone locations. There were no cues on the surrounding walls (A: identical pure white; B identical pure black) to provide allocentric information so that mice have to position the food by objects in the context (Supplementary Fig. 1a). Mice were trained three trials per day and underwent four days of training followed by two recall trials on day5. The trial interval was 2 min. In training trials, once the mouse finished the food or the mice cannot find the food in five minutes, we ended the trial. In recall trials, there was no food reward and mice were allowed to freely explore in the context for 3.5 min. The food zone was defined as a 10 cm × 10 cm rectangle with the center located at the food reward. The spatial memory task in Figs. 5 and 6 and Supplementary Fig. 19 was slightly different, training mice were confined on the start-region 1 min by an L-shape door (Pre-exploring,1 min) before freely exploring in the context (Exploring phase, 2 min). L-shape door was non-transparent, mice recall in the pre-exploring phase is indicating a recall without major sensory input from the context.

The y-axis range from 0 to 900 s in Fig. 2c is because the 'latency to food intake' means the accumulation time spent to reach the first food throughout three trials in a training day. For example, if a mouse cannot obtain the food in all three trials in a training day (300 s at most in one trial), the latency here most would be 300 s + 300 s + 300 s = 900 s, and if the mouse obtains the food in the first trail by 50 s, that the latency would be 50 s. The food pellets were air-drying to reduce the odor and buried under the sand. These operations made it harder to reach the food, thus mice needed a longer time to recognize landmarks and dig in the right place to reach the food.

**Memory test of landmark relocation**. If mice are using an egocentric map during the spatial memory task, changed object locations will impair their performance. To verify this, a group of mice trained with the same spatial task paradigm (learn to find food in sandbox A for 4 days, Supplementary Fig. 16) followed by a performance test in the same box with altered object location (A'). The result showed that altering the object foodzone location largely decreased the foodzone occupancy in test trials (Supplementary Fig. 16b). This indicated that locations of objects are critical for mice to locate the food reward. Noted that the context has little allocentric information (unlike water maze with signs on the wall) so that mice have to rely on their relative position (egocentric information) to objects to locate the food.

**Video recognition for occupancy and head direction**. We identify the location of the mice by video recognition and then calculated the total time the mice spend in the foodzone to quantify the occupancy. Here is the entire video recognition process: 1. Frames were extracted every 0.2 s from the video file and then resize to 1/4 size and changed to grayscale. Each frame was subtracted by a frame without mice (usually it is an average of all frames) to produce frame difference for the determination of the mice location. One recall trial in total is 210 s (1050 frame) and starts when mice get off from the start block. 2. Each frame difference was binarized, the average of all points of x or y was determined as the mice body location. Head direction was determined by Skeleton algorithm 3. The context was divided into 46 × 46 grid, the foodzone was defined as a 10 × 10 region around the food position. Each bin of occupancy map was calculated as the proportion of body positions in that bin. For the occupancy of the foodzone, the total number of frames in which the mouse is located in the foodzone is $T_{foodzone}$, and the total frame number of the trail is $T_{all} = 1050$. Therefore [Eq. 1]:

$$\text{Occupancy}_{foodzone} = 100\% \times \frac{T_{foodzone}}{T_{all}} \tag{1}$$

**Contextual fear memory task**. Contextual fear conditioning was performed in a fear box (square chamber, 20 × 20 × 39.5 cm) with a metal-gridded floor and yellow

environment illumination. Experiments were performed in the TSE system. Training trials consist of a 3 min' exposure of mice to the conditioning box followed by a foot shock (2 s, 0.8 mA, constant current). The memory test in Fig. 3a–e was performed 24 h later by re-exposing the mice for 3 min into the conditioning context. In Supplementary Fig. 12 memory tests were in 7 days after TRAP labeling (day8), 1st recall and the 2nd recall were performed on the same day, the second one was performed 10 mins after the first one. Freezing, defined as a lack of movement except for heartbeat and respiration associated with a crouching posture, was recorded by video and rated every 10 s by two blinded observers (unaware of the experimental conditions) for 3 min (a total of 18 sampling intervals). The number of observations indicating freezing obtained as a mean from both observers was expressed as a percentage of the total number of observations. The results were cross-validated by the automatic freezing counting in TSE.

**LEC neurons and their cortical fibers stimulation or inhibition by optogenetics.** We utilized the Arduino module to control the laser. During the electrophysiology coupled with optogenetics experiment, we activated LEC neurons by 470 nm laser (12.45 mW/mm$^2$, Aurora-300-470, Newdoon technologies) and activated LEC cortical fiber by the same laser and irradiance. In the behavior experiment, we activated LEC cortical fiber by 470 nm LED (5.4 mW/mm$^2$, model: 5050-470), inhibition of LEC fiber by 589 nm LED (6.8 mW/mm$^2$, 5050–589).

**iSOS rescue experiments.** For rescue experiment of contextual fear conditioning (Fig. 3a–e). The experiment procedure is basically the same as described above (Method section: Contextual fear memory task), the distinct point of the rescue experiment is that mice were trained with or without iSOS (5.4 mW/mm2, 10 Hz or 30 Hz, 4 s on, 6 s off, 4 ms), depends on each group design. For rescue experiment of spatial memory task (Fig. 3f–j). The experiment procedure is in principle the same as described above (Method section: Spatial memory task). In each training trial, mice were delivered iSOS (5.4 mW/mm2, 10 Hz or 30 Hz, 4 s on, 6 s off, 4 ms pulse) or not, depending on each group design. In the experiment of reverse training of random signal ('30 Hz', pulses are randomly distributed but maintain 120 pulses in 4 s) and iSOS (30 Hz, 12.45 mW/mm2, 4 s on, 6 s off, 4 ms), two groups of mice underwent the opposite training order (Fig. 4). One group first trained with iSOS in sandboxA followed by random signal training in sandboxB, the other group underwent random signal in A, and then iSOS in B. After training in sandboxA, mice were back to homecage to recuperate with food and water ad libitum for 3 days before training in sandboxB. To ensure we can activate LEC cortical fibers separately, two optical fibers (diameter = 200 μm) with a low numerical aperture (N.A. = 0.37) were placed right on the surface of the cortex to confine the emission light to the local region.

**Immunohistochemistry.** Mice were overdosed with 400 μl 2% phenobarbital sodium and perfused transcardially with cold PBS, followed by 4% paraformaldehyde (PFA) in PBS. Brains were extracted from the skulls and kept in 4% PFA at 4 °C overnight, then transferred to 20% sucrose in PBS. 50 μm thick coronal slices were taken using a vibratome and were collected in cold PBS. For DAPI staining, each slice was placed DAPI in PBS (1:10000 dilution) for 40 min at 37 °C. Slices then underwent three times of washing for 5 min each in PBS, followed by adding the Anti-fade Mounting Medium (P0126, Beyotime) and coverslip on microscope slides.

**Microscopy and cell counting.** Sections (50 μm) were imaged using a ZEISS (LSM710META) confocal microscope. All imagings were done using standardized laser settings held constant for samples from the same experimental dataset. LEC downstream neurons were quantified in four cortical regions (RSC, MO, VIS, SS), all slices were quantified including cell amount and cell depth, for cell depth quantification, all tdTomato positive cells were imaged and calculated shortest distance from the cell body to the cortex edge by ImageJ (v1.52p) measure tool.

**Memory-related neurons labeling and photometry.** Activated neuron labeling. The TRAP system[51] was adopted to label the activated neurons in a specific time window (task-related neuron labeling). First, AAV-DIO-GCaMP6f virus was injected into the superficial layer of retrosplenial cortex or visual cortex and optrodes (200 or 300 μm core diameter, 1.2 mm length, N.A. = 0.48, Newdoon Inc., hand-made by gluing the electrode to the optical fiber) were implanted above the virus injection sites and sealed by black dental cement. Noted that the tip of optical fiber was placed on the surface of the cortex (DV = −0.0 mm) and the protruding electrode was implanted into the cortex with the depth of 0.25 mm (DV = −0.25 mm, see schematic diagram in Supplementary Fig. 14a), mice could recover for 2 weeks before all subsequent experiments. Second, c-Fos-Cre$^{ER}$ mice were injected with tamoxifen for labeling memory engram (dosage: 100 mg/kg) 24 h before training day 4 (labeling day), released Cre protein enables the GCaM6f expressed in activated neurons. Third, 7 days after labeling (PBS: day11/13; CNO: day12/14, CNO was injected intraperitoneally 1 h before the behavioral experiment), engram activities were recorded by fiber photometry system in homecage (Home), unfamiliar context (CtxB), and learning context (Recall x2) followed by a retraining trial to avoid memory extinction. All fiber placements and virus injection sites were histologically verified post experiments. In practice, the surface of the optical fiber

and electrode vary across cases, only those cases with optical fiber and electrode located in target place were included (fiber surface, depth <200 μm; electrode, depth <350 μm). Images in Supplementary Fig. 14e show cases with optical fiber located on the surface of the cortex. In addition, we also quantified the CEs from cases with optical fiber depth more than 200 μm (Supplementary Fig. 14f), we found neuronal activity from these cases shows no context specificity (Supplementary Fig. 14g).

**The fiber photometry system.** The fiber photometry system was bought from Thinker Tech Nanjing Biotech Limited Co. Excitation light from a 488 nm semiconductor laser (Coherent, Inc. OBIS 488 LS, tunable power up to 60 mW) was reflected by a dichroic mirror with a 452–490 nm reflection band and a 505–800 nm transmission band (Thorlabs, Inc. MD498), and then coupled to a fiber (Thorlabs, Inc., 200 μm in diameter and 0.48 in N.A.) by an objective (JiangNan, Inc. 20×, N.A. 0.4). The emission fluorescence was collected with the same optical fiber and then detected by a highly sensitive photomultiplier tube (PMT, Hamamatsu, Inc. H10720-210) after being filtered by a GFP bandpass emission filter (Thorlabs, Inc. MF525-39). The laser intensity at the interface between the fiber tip and the animal was adjusted to around 30 μW to minimize bleaching. Signals were collected at a sampling frequency of 100 Hz and further filtered through a 10 Hz IIR low pass filter. The analog voltage signals were digitalized and collected using FiberPhotometry software (Thinker Tech Nanjing Biotech Limited Co.) and were further analyzed in MATLAB.

Signal analysis. At the beginning of our experiment, we set the cutoff of $\Delta F/F_0$ according to a set of control experiments. We tested the noise of our recording system by measuring the signal from unchanged EGFP signals (data not shown). This control signal (EGFP) was detected as small variations only (<1% $\Delta F/F_0$), far smaller than the GCaMP6f signal. According to this control experiment, we set the threshold of calcium event identification to 1.25% $\Delta F/F_0$ for 470 nm signals (GCaMP6f). Thus, we perform our subsequent GCaMP6f recording by the system. Here $\Delta F/F_0 = (F − F_0)/F_0$, where F is the fluorescence intensity at any time point, and $F_0$ is the averaged F in five seconds time window with a center on the corresponding time point. To further verify the specificity of recorded signal, we recorded the signals from tissue simultaneously with 405 nm and 470 nm light, we found the signals are specific to GCaMP and only 470 nm light can detect obvious fluorescence changes in different tasks (Supplementary Fig. 14d). Therefore, our recorded signals represent the activities of labeled neurons.

**LEC inhibition during spatial memory encoding.** cFos-Cre$^{ER}$ Mice were trained with inhibited LEC by hM4D(Gi) (AAV2/9-hSyn-hM4D(Gi)-mCherry with CNO, which was injected intraperitoneally 1 h before the behavioral experiment) during every training day in the spatial memory task and memory-related neurons were labeled on day4 (Supplementary Fig. 19a). Same as the behavioral protocol of spatial memory task before. Behavioral tests and recoding of calcium activities were set on day 11 (also with CNO injection for LEC inhibition). Although these mice could find the food during training, they showed poor memory recall in recall trials (Supplementary Fig. 19b, c), indicating the LEC is essential for the storage of long-term memory. In the cortical neurons labeled with GCaMP6f during training, CEs were detected in homecages and untrained unfamiliar context and reduced in the trained context (Supplementary Fig. 19d). This result is in great contrast to the LEC activated condition.

**LFPs data collection and analysis.** The raw data was amplified during the behavioral experiment with a 1000 Hz sampling rate (Apollo Neural Data Acquisition (DAQ) Systems, v1.0.0, Bio-Signal Technologies, USA), LFP signals were bandpassed at 1–200 Hz. Notch filters were not applied. Unless indicated otherwise, analyses were performed using chronux toolbox (2.12v03) and MATLAB code written by the authors. Channels with a low signal-to-noise ratio (SNR) were identified and deleted. Reasons for low SNR included 50 Hz line interference, electromagnetic noise from surrounding equipment, and poor connection between screws and cortical tissue. Mice with large moving artifacts or 50 Hz line noise were excluded from the final dataset.

To guarantee that the LFP measures were collected as mice behaved similarly. LFP segment was sectioned with the following criteria: For a naive trial, it's a segment of LFP data from a 3 min period when the mice were awake in the homecage before spatial memory training (day0), while training trials were divided into several parts: 1. Preparing, on the start region. 2. Exploring, from the timepoint of getting off the start block to finally food eating. 3. Food eating. We found that some major LFP differences between these parts, there is little theta power in preparing and stronger delta power when mice are eating the food. During exploring, LFPs show similar power patterns with higher theta and gamma, mice actively explore the environment and find food, which is a good time period to study memory encoding. Therefore, only the exploring phase as the encoding trial was employed for subsequent analysis including synchrony, power, modulation index, all spectrograms and theta amplitude base cross-correlation. For recall trials (3.5 min, 210 s), it's a 210 s LFP segment from the timepoint of getting off the start block to the end of the trial, because there is no food reward in the sandbox.

Bandpass-filtering. A *filtfilt* function in MATLAB is utilized (zero-phase shift, iir) for bandpass-filtering, different bands defined as: δ (1–4 Hz), low theta (4–7 Hz), theta (7–10 Hz), β (11–20 Hz), For gamma definition, we defined low

gamma (20–40 Hz) and high gamma (60–80 Hz) as PLV calculation may be contaminated by 50 Hz inference. Strictly, although most researchers define 25–40 Hz as low gamma band, we define 20–40 Hz as the low gamma band in this article for convenient description.

**Wavelet analysis.** Given that the filtered LFP is not a perfect sinusoid, wavelet is more suitable for analysis transient power and phase difference (wavelet toolbox v1.0)[73,74]. LFPs were transformed into the time-frequency domain $\mathbf{w}(t,f)$ for further analysis, using complex Morlet wavelets (wavenumber = 6, evaluated at 0.25 octave intervals from 5 to 80 Hz). The amplitude, phase, and power can be calculated by following equations [Eqs. 2–4]. The amplitude of the LFP for each frequency $A(t, f)$ is corresponding to the length of $\mathbf{w}(t, f)$

$$A(f,t) = abs(\mathbf{w}(t,f)) \tag{2}$$

And the phase of the LFP for each frequency $P(t, f)$ is corresponding to the angle of $\mathbf{w}(f, t)$:

$$P(t,f) = angle(\mathbf{w}(t,f)) \tag{3}$$

**Time-frequency analysis of power.** Since we got the instantaneous amplitude of the LFP for each frequency, the LFP power can be estimated by:

$$Power(t,f) = 10 \times \log_{10}(A(t,f)^2) \tag{4}$$

**Phase-power spectrogram.** To analyze the dynamics of LFP power underlying theta (Supplementary Fig. 3). We need to identify all theta (7–10 Hz) troughs of the LFP data and extract data segments locked to theta trough (1000 ms data centered on theta troughs). To identify the theta troughs, we need to construct the analytic phase time series $P_{theta}(t)$. First, the raw signal $X_{RAW}(t)$ was filtered from 7 to 10 Hz to create the real-valued bandpass filtered signal $X_{theta}(t)$. Second, the complex-valued analytic signal $Z_{theta}(t)$ was created by applying the Hilbert Transform[28,75], where $Z_{theta}(t) = X_{theta}(t) + iH[X_{theta}(t)]$. Third, the phases at each time point were extracted from $Z_{theta}(t)$ to create the analytic phase time series $P_{theta}(t)$. $P_{theta}(t)$ assumed values within $(-\pi, \pi]$ radians, where $\pi$ radians corresponds to a theta trough and 0 radians corresponds to a theta peak. theta troughs were identified as all local maxima of $P_{theta}(t)$ higher than 3.14. This produced more than 2000 theta segments in a 3 min data. The average of raw data segments can produce the LFP-trough locked potential (traces in the top panel of Supplementary Fig. 3), and the average of power segments produced the phase-power spectrogram. Noted that we keep on continuously resampling and averaging data. For example, a minute of data would contain about 60*8 = 480 theta cycles, 480 data segments range from −500 ms to 500 ms. Data was subsampling ±500 ms around theta troughs. But only the result of ±400 ms time range was shown in the Figure. For a more clear comparison of the power change between frequencies, normalization is required. Power was normalized to a z-score-like statistic by subtracting the mean value and dividing by the s.d.

**Phase-power modulation index comodulograms.** To measure the coupling intensity between high frequency power and low frequency phase, we used modulation index (MI) developed by Tort and his colleges[76], corresponding Figure is showed in Fig. 1f. The MI can be calculated in all combinations of low frequencies and high frequencies, thus we can construct a two-dimension map (MI comodulograms) of with low frequency 2–20 Hz and high frequency range from 20 to 80 Hz. The MI comodulogram was normalized by the element-wise division of the raw comodulogram by surrogated control, which is the averaged one from 100 comodulogram calculated with random shifts of the phase of the low frequency.

**Phase-synchrony spectrogram and overall synchrony estimation.** We have considered two LFP metrics in this study, one is the LFP power (local neuronal synchrony) and another is the synchrony of the LFP phase between cortical regions (long-range phase synchrony). To analyze the dynamics of phase synchrony underlying theta wave (Fig. 1g) and estimate the overall synchrony for comparison among different kinds of trial (Fig. 1h). We followed the method developed by Jean-Philippe Lachaux et ai.[74]. In this method, the strength of neural synchrony was quantified by the PLV, and based on multi-trials, it needs to align phase-difference data to the onset of trials and then calculated the likelihood of the phase-difference for the given delay in time, it represents the phase synchrony level around the onset. Inspired by this approach, to estimate the phase synchrony change with respect to the theta wave we can align timepoints to the trough of theta wave instead of onset of trials, and then the strength of neural synchrony was quantified by the PLV, the length of the vector average of cross-electrode differences [Eq. 5].

$$PLV_\theta(f,t) = \left| \frac{1}{n_{\theta trough}} \sum_{k=1}^{n} e^{i\left(P_{electrod1,k}(f,t) - P_{electrod2,k}(f,t)\right)} \right|, t \in (t_{\theta trough} - 500\ ms, t_{\theta trough} + 500\ ms) \tag{5}$$

Where $f$ is the frequency, $t$ is time, $n$ is the number of theta troughs, $P_{electrod1,k}(f,t)$ and $P_{electrod2,k}(f,t)$ corresponding the phase for frequency f at time t of two electrodes, which is obtained from wavelet analysis mentioned above, k is the iteration of theta trough ID. Akin to the phase-power spectrogram, we can construct the phase-synchrony spectrogram (Fig. 1g) by $PLV_\theta(f, t)$, see method above for detailed theta phase extraction. Noted that we keep on continuously resampling and averaging data. Phase data was subsampling ±500 ms around theta troughs. But only result of ±400 ms time range is showed in the Figure. In the calculation of phase-synchrony spectrogram and overall synchronization, we define

alignment to theta as follows: the MO-RSC synchrony was aligned to theta wave of MO, the RSC-VIS synchrony was aligned to theta wave of RSC, and the VIS-MO synchrony was aligned to theta wave of VIS.

There are two ways to estimate the overall synchrony of a trial (Fig. 1h) [Eqs. 6, 7], one way is to average $e^{i(Phase\ difference)}$ of all time points of the trial:

$$PLV_{overall}(f) = \left| \frac{1}{T_{triallength}} \int_{t_{trial-start}}^{t_{trial-end}} e^{i(P_{electrod1}(f,t) - P_{electrod2}(f,t))} dt \right| \tag{6}$$

Another way is to simply average PLV in theta cycle:

$$PLV_{overall}(f) = \frac{1}{2\pi} \int_{-\pi}^{\pi} PLV_\theta(f,t) dt \tag{7}$$

We decided to employ the latter approach because it represents not only the overall level of the synchronization throughout the trail but also represent the average synchronization level in the theta cycle. For the surrogate control, the PLV is calculated by the same procedure but aligned the phase data to randomly shifted theta trough. This preserves both the number of locking indices and the interval between sequential indices, assuring that the real and surrogate indices both have the same statistical properties. For calculation of induced cortical synchrony (iSOS) by activation of LEC neurons (Fig. 2l–n), phase difference data were aligned to the onset of each stimulation (±300 ms). To monitor the synchrony change along with time (to analysis together with calcium signal or animal locomotion), we calculated overall synchrony by the above equation with 2 s time window (from −1s to +1 s, contain about 14–20 theta troughs) by each 0.2 s step.

**Construction of EORs and significance detection.** Construction of EORs was similar to the generation of egocentric boundary ratemap (EBR), which was previously reported[54]. Although EORs were designed based on the same principle as EBR, the EORs were considered the CEs of GCaMP6f signal from each labeled engram population instead of considering the firing rate of a single neuron. Data were considered in an egocentric reference frame where an object position was considered relative to a static mouse position (Fig. 5f, g, seven objects, including five glass blocks, wall and intangible food zone). The primary components used to generate the EOR are the animal's head direction and a specific object position relative to the animal. The position of a specific object relative to the mouse was calculated on a frame-by-frame (0.2 s per frame) basis and we calculated the relative position for each object separately. Head direction determination were calculated based on the bwmorph function in MATLAB, please see custom code for detail. The 360° space around the animal was evenly divided into 3° angular bins centered (0°) on the animal's instantaneous heading and the distance from the animal's current position was divided into 0.5 cm distance bins up to a 30 cm maximum distance, yielding 3° × 0.5 cm bins. First, for each frame during exploring in the recall trial, the presence of an object in each bin is counted resulting in an egocentric object occupancy map ($Map_{Allframe}$). Second, for a given engram population, a density plot of object location at the rising time of calcium event is generated ($Map_{CEs}$). Rising time is defined as the time of the peak minus the width of half-prominence. Third, occupancy normalized to EOR were then made as the element-wise division of the $Map_{CEs}$ by $Map_{Allframe}$. Therefore, normalized EOR for a specific object is [Eq. 8]:

$$EOR_{normalized}(i,j) = \frac{Map_{CEs}(i,j)}{Map_{Allframe}(i,j)} \tag{8}$$

Finally, the EOR was smoothed by a 2D moving average (with a circle $r = 5$ bins). To further validate the significance of the tuning strength of an EOR. we need to calculate the MR (mean resultant, [Eq. 9]) of each EOR:

$$\mathbf{MR} = \left( \sum_{\theta=1}^{n} \sum_{D=1}^{m} F_{\theta,D} * e^{i*\theta} \right) / (n * m) \tag{9}$$

where $\theta$ and $D$ is the orientation and distance relative to the mouse, $F_{\theta,D}$ is the firing rate in a given orientation-by-distance bin, $n$ is the number of orientation bins, $m$ is the number of distance bins, $e$ is the Euler constant, and $i$ is the imaginary constant. Then MRL [Eq. 10], used as a measure of object orientation specificity, is calculated as

$$MRL = abs(\mathbf{MR}) \tag{10}$$

and the mean resultant angle, which is used as the preferred egocentric object orientation is calculated as [Eq. 11]:

$$MRA = arctan2 \left( \frac{imag(\mathbf{MR})}{real(\mathbf{MR})} \right) \tag{11}$$

From each EOR, we computed the MRL of angular tuning for each recall trial (each mouse has 2 or 3 recall trials after training). Labeled memory-related neurons were determined to display significant egocentric object sensitivity if they met the following standards: (i) they had an MRL that was greater than the 95th percentile of the distribution of resultants computed following repeated shifted CEs randomizations, (ii) they had an absolute difference of mean directional tuning between recall trials that was <45°, and (iii) the change in preferred distance relative to the full session was <50% for trials. All standard-compliant EOR was considered the significant EOR in this paper (Supplementary Fig. 15). The same

criterion for detecting significant egocentric coding was employed in LEC inhibition by CNO or PBS (Supplementary Fig. 18).

**Egocentric tuning to non-object regions**. To further validate whether these activities are reliably tuned to an object egocentrically. We tested if these CEs show significant egocentric tuning to non-object regions. Except for 6 objected regions, 30 non-object regions were defined and tested by the same criterion as mentioned (Supplementary Fig. 15e). The result showed that these CEs are more likely to show significant tuning to objects but not non-object regions (Supplementary Fig. 15f).

**Egocentric significance of random CE simulation**. To statistically test our method for detection of significant tunings, a simulation was performed: position of the mice and its head direction, as well as the egocentric tuning and non-egocentric CEs were randomly generated 500 times (same region probability and total time point as real recall trials $3.5*4 = 14$ min, 4200 points), followed by a calculation for the detected rate of significant egocentric coding (Supplementary Fig. 15g, h). Egocentric CEs were generated with a distance range from 5 to 10 cm and angle $1/2\pi$. Only 6 (or >60% of total) CEs that always fire in the same position relative to the animal are able to make a neuron population detected as having significant egocentric tunning (Supplementary Fig. 15h). Moreover, when there was no egocentric CE, only 6.7% of simulations found significant egocentric coding (false positive rate = 6.74%). This data suggested that the method for the detection of significant tunings is statistically reliable.

**Statistics**. Results are presented in a bar or line plots as mean ± s.e.m, GraphPad Prism version 7.00 (GraphPad Software, La Jolla, California, USA) and OriginPro version 9.0 (OriginLab Software, Northampton, Massachusetts, USA) were used for statistical analyses. Statistical significance was assessed by two-tailed paired Student's $t$ tests, two-tailed unpaired Student's $t$ tests, one-way ANOVA ($P \le 0.05$) or two-way ANOVA ($P \le 0.05$) where appropriate, multiple comparisons are calculated if necessary. Significance levels were set to $P = 0.05$. Significance for comparisons: $*P < 0.05$; $**P < 0.01$; $***P < 0.001$, $\#q < 0.05$; For all box plot, whiskers show min and max, all box shows 25th, median and 75th percentile.

**Reporting summary**. Further information on research design is available in the Nature Research Reporting Summary linked to this article.

## Data availability

All data supporting the findings of this study are provided within the paper and its supplementary information. A source data file is provided with this paper and can be publicly accessed, we deposited data on figShare repository (https://doi.org/10.6084/m9.figshare.19221969) or one can directly download via the link this link https://figshare.com/ndownloader/files/32333996. All additional information will be made available upon reasonable request to the authors. Source data are provided with this paper.

## Code availability

All codes used in this study are available and can be publicly accessed. Custom Matlab codes were deposited data on figShare repository (https://doi.org/10.6084/m9.figshare.19222011). One can directly download them at https://figshare.com/ndownloader/files/32179832.

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

## Acknowledgements

We gratefully acknowledge helpful discussion and suggestions with Dr. Wei Lu, Dr. Longnian Lin, and Dr. Ji Hu. We thank Xiaoming Li, Rui Wang, and Ziwei Yang from the Molecular Imaging Core Facility (MICF) at the School of Life Science and Technology, ShanghaiTech University, for providing technical support. This work was supported by grants from the Science & Technology Innovation 2030 Project of China (2021ZD0203500) to J.-.S.G. This work was also supported by grants from NSFC (31970903, 31671104,31371059), and by Shanghai ministry of science and technology (19ZR1477400) to J.-.S.G. The work was also funded by the National Natural Science Foundation of China and the German Research Foundation (DFG) in project Cross-modal Learning, NSFC (61621136008) /DGF TRR-169 to J.-S.G. W.H.L. is supported by STCSM (18JC1420301).

## Author contributions

J.-S.G. conceived the study and experiments. W.H.L. built the experimental setup, performed recording experiments, optogenetic stimulation, and data interpretation. D.Y. studied the correlation between LFP and engram activity. Y.T., Y.Z. assisted in the setup of in vivo electrical recording. Y.H. performed the surgery. M.M.T. helped with animal behavior experiments. W.H.L. and all authors interpreted the data under the supervision of H.X. and J.-S.G. The paper was written by W.H.L., J.-S.G., and commented by all authors.

## Competing interests

The authors declare no competing interests.
