## [Peer Review File · Nature Communications]

Reviewers' Comments:

Reviewer #1:

Remarks to the Author:

The authors showed that spatial learning and retrieval induce cortical theta-gamma coupling in RSC, MO, and VIS. Strikingly, optogenetically inducing this coupling by stimulating LEC cell bodies or projections to the cortex during training restored spatial memory in HPC-lesioned mice. This indicates that we can bypass HPC by stimulating the LEC-cortex circuit. The strength of the manuscript is that the authors performed an impressive array of behavior and techniques. It has the potential of high interest in the learning and memory field. However, the manuscript is very poorly written, and I had to read it multiple times to figure out when, how, and from which structure the data were collected. LFP recordings need to be validated. Also, there are several logical gaps between data presentation. Unless these gaps are filled, this is a very impressive shopping list of data that are not tightly held together for a complete story for publication.

Major concerns.

1. Fig.1d. VIS, RSC, and MO show different ranges of amplitude (power) of oscillations. However, all raw power plots from each structure provided throughout the paper are in the same range. Also, all SEM of the average of all 3 structures throughout the paper is negligible (e.g. Fig.1e). How is this possible?
2. Related to the first comment, LFP recordings suffer from volume conduction and are coarse measures of neural activities. The fact that LFPs from all cortical structures were very similar and the latency of evoked signals from all cortical structures after optogenetic stimulation were all in the same range (supp. Fig. 5), I suspect volume conduction. Another evidence of cortical volume conduction is Supp. Fig. 4b. Here, power in LEC and HPC have much bigger SEM, considering the scale of the graph. Ideally, the authors should have done single-unit recordings instead of LFP recordings. However, it will be impractical, and I ask the authors do something to ensure that their data are not affected by volume conduction.
3. Also, LFPs are affected by behavior. For example, when mice are digging, there are huge jitters in LFPs. Thus, all LFP analyses should be done in a stereotypical behavior (e.g. running in a certain range of speed in one direction, not exploring objects).
4. Fig.2l-n. Were these done in home cage? This should be done in a stereotypical behavior.
5. Were training data collected on Day4? To claim that cortical theta-gamma coupling is induced by learning, authors should compare the coupling between Day1 and Day4 to show that it evolves through learning.
6. Fig.3. Why not LEC cell body stimulation? Because Fig2. is showing cell body stimulation, it is logical to stick to cell body stimulation. Or, authors should show that fiber stimulation also induces theta-gamma coupling.
7. Fig.4. The authors claim that theta from the HPC/LEC modulates cortical theta that couples with cortical gamma. Also. Supp. Fig. 4c shows that LEC-MO synchronizes at theta, but not gamma. Therefore, cortical stimulation should be done in theta (10 Hz), not in gamma. Same thing for Supp. Fig. 7e.
8. Fig.5 & 6. These are very impressive data. However, they are not well tied together with the rest of the data and can be used for a separate paper. Fig. 1-4 focuses on cortical theta-gamma coupling during training (especially with optogenetic stimulation). However, Fig. 5 & 6 are showing cortical calcium activity during memory recall (testing) and focuses on cortical gamma synchrony during pre-testing. To fill this gap, the authors should show whether asynchronous cortical theta stimulation (as in Fig.4) or inhibition of LEC during 'training' disrupts engram activity during 'testing'. Also, does theta-gamma coupling modulate cortical gamma synchrony? Finally, these are fiber photometry data of collective calcium activity of the population of neurons. Conventionally, engram means single neurons encoding memory. I would not call these collective calcium activity engrams.
9. Related to #8, gamma modulated calcium activity occurs during pre-testing. Then, what does it have to do with egocentric mapping during testing? This logical gap needs to be filled.
10. Fig.5b-d. Are calcium signals between VIS and RSC the same? Why only RSC is shown?
11. If cortical neurons indeed form an egocentric map, this can be tested. In experiments in Fig3f, test mice in the same box with altered object locations. If mice are using an egocentric map strengthened by optogenetic stimulation, this change will impair their performance.
12. Supp.1f. This should show theta-gamma coupling in each structure.

13. The biggest gap in the logic is the switch from HPC to LEC. After showing data in HPC lesion, the authors suddenly switched to LEC. To fill the gap, the authors should show that HPC leads/modulates LEC theta in the main figure to link it to Fig. 3. showing that LEC theta stimulation can bypass HPC.
14. Related to #13, does HPC directly project to the cortical area? Can we directly manipulate HPC projections? Why should LEC relay the signal?
15. Supp. Fig.4. Are these done on day 4? It will be better to show the signal evolves through day1-4.
16. Supp. Fig. 9e. It looks like there are more CEs during freezing, contradictory to Supp. Fig. 9d. showing that there are more CEs during pre-freezing
17. Line 79-80. The location of theta trough and gamma peak is relative. Depending on where you start to look, either one leads the other. Lag analyses should be done to claim this.
18. Line 124-127. This is critical information and should be presented in a figure. Show correlation for HPC-RSC, HPC-VIS, HPC-MO, LEC-RSC, LEC -VIS, and LEC -MO.
19. HPC lesion. Line 726. "behavioral assessments...were performed immediately after surgery to determine the effect of hippocampal lesion." Aren't the learning task and recordings done 2 weeks after surgery? This should also be done 2 weeks after surgery.

Minor concerns.

1. Fig.2 l. Is this the average of RSC, MO, and VIS recordings after LEC cell body stimulation?
2. Fig.2 m & n. Are these the average of RSC-MO, RSC-VIS, and VIS-MO after LEC cell body stimulation?
3. Supp. Fig. 5l. Show individual data points.
4. Supp. Fig. 9b. It looks like the first recall was 7 days after training. When did the second recall happen?
5. Supp. Fig. 9g. When were the cells labeled?
6. Figures are not described in the text in order. It is better not to have readers go back and forth.
7. There are many typos and grammar errors.
8. Line 95. Fig. 2e is not described.
9. Line 119. Specify that dorsal HPC CA1 was targeted.
10. Line 121-122. Why not showing cortical gamma power is coupled to HPC theta?
11. Line 142-143. This statement is not supported by the data. LEC was engaged with HPC at theta?
12. Line 233. Overall -> random.

I am strongly against the notion that authors must do whatever the reviewer mentions to publish their work. Reviewers can be wrong. My comments are suggestions that can make the paper more logical and meaningful. If the authors feel that my suggestions are inappropriate or there are better ways to address my concerns, I am open to the authors' opinions.

Reviewer #2:

Remarks to the Author:

Nature Communications 309301_0_rel_ms_0_qs7zx4.pdf

"Acquiring new memories in neocortex of hippocampal-lesioned mice" is a potential tour-de-force, a combination of modern methods to address the classic problem of how the brain stores remembered information. Mice were trained to find food that was hidden under sand in a consistent place relative to several objects in a testing arena. Intact animals learned to find the food faster with several training trials, mice with neurotoxic lesions of the hippocampus were impaired. Local field potentials (LFPs) recorded simultaneously from the secondary motor, primary visual, and retrosplenial cortices revealed prominent theta and gamma oscillations with power and power-phase coherence that increased with training and recall. Both LFP measures were reduced by hippocampal lesions. Optogenetic stimulation of neurons in the lateral entorhinal cortex (LEC) evoked synchronized LFPs in the same regions, rescued performance in the just described spatial memory task as well as in contextual fear conditioning in rats with hippocampal lesions. Asynchronous optogenetic stimulation of LEC fibers innervating pairs of other cortical regions during training impaired spatial learning. Fiber photometry combined with TRAP (Targeted Recombination in Active Populations) detected calcium fluorescence in groups of retrosplenial and

visual cortical neurons that were activated during learning. The tagged neurons were most active in the training context, and the activity was correlated with proximity and heading angle to the more prominent objects and where food had been hidden. Memory performance, LFP synchrony, and context triggered calcium signals were impaired by chemogenetic inactivation of LEC. Together, the results suggest that gamma synchronized activity across cortical modules that is normally conveyed by LEC and coordinated by hippocampal theta rhythm may be necessary and sufficient for encoding spatial memories.

Presentation errors and missing details make it difficult to evaluate the paper. Below I've listed a few examples of such errors.

More detail is needed about the behavioral correlates of the electrophysiological measurements, in particular how the recording data were selected for analysis, and how the methods guaranteed that the LFP measures were collected as mice behaved similarly, e.g. walking within a particular speed range, in different task stages (e.g. figure 1d-h) you know. What defined "naïve"? The only description in the methods is for the phase power spectrograms-- what about the other LFP measures? How is the occupancy in food zone % quantified (figure 2D)? The dimensions of the spatial memory testing apparatus should be in the methods section (46x46 cm was only in Supp. Fig. 1a).

The statistic in several places are not appropriate. E.g. figure 2i claims that the HPC lesion reduces synchrony, but should be supported by a significant interaction between lesion group and training stage synchrony. The same issue holds for fig. 3i, supplemental figure 4j, supplemental figure 7h, supplemental figure 11h (ANOVA vs t-tests).

Language errors should be fixed throughout. E.g., Supp. Fig 8 e,f "recoding" should be "recording;" Line 46, causally, not casually; Line 714 "carinal" for "cranial".

Line 27. The interconnections among MTL areas have been known for decades, and hence are not recent.

Figure 3 caption, lines 604-605: EYFP is missing, oChIEF is repeated. "P iSOS (10Hz,oChIEF) vs. HPC-lesion > 0.9999, P iSOS (10Hz,oChIEF) vs. HPC-lesion = 0.0243,"

Reviewer #3:

Remarks to the Author:

Although theories of the hippocampal role for memory assume functional reactivation of the neocortical network during memory retrieval (e.g., Teyler and DiScenna, 1986), our understanding of how the hippocampus coordinates the activities of multiple neocortical areas still remains scarce. The authors found the lateral entorhinal cortex (LEC) mediates the hippocampal-neocortical interaction and coordinates long-range gamma-synchrony among the retrosplenial, visual, and motor cortices. Surprisingly, optogenetically induced gamma-range synchrony of these cortical structures rescued memory impairments in the hippocampus lesioned animals. These findings are strong supports for the memory index theory, which proposes the primary role of the hippocampus to be the reinstatement of cortical activity, and would be of great interest from general readers. I would say this is one of the most interesting papers I read this year, and I believe, if published on Nature Communications, this paper will have a substantial impact in the field.

To further improve the quality of the manuscript, I would like the authors to address several concerns.

1. When the authors analyzed coupling between cortical long-range gamma-synchrony and theta rhythm, it is unclear which brain area they used for theta oscillation. Because theta coherence across cortical structure varies depending on the animal's state, behavior, etc (e.g., Young & McNaughton, 2009), it would influence their interpretation of 'coupling' if they choose the reference area randomly.

2. The authors need to provide a stronger rationale for the construction of egocentric object ratemaps (EORs). In the previous study the authors referenced (Alexander et al., 2020), animals

were allowed to explore the environment freely. In the present study, I would imagine the animals' trajectories during the recall trials are more stereotyped due to their better performance to navigate towards the goal location, which potentially results in poor occupancy of angular/distance bins. Did each animal fill all angular/distance bins? If not, the authors need to provide 1) percent occupancies for each animal and 2) statistical support that the observed activities are still reliably tuned to an object.

3. I would like to hear more about the author's thoughts about BMI. I feel the authors wrote BMI without sufficient background. What are the current issues of BMI in application to memory defective diseases? And how does the finding solve them?

4. Also, it would be great if the authors provide more discussions on the functional roles of HPC, especially in the context of the Memory Index Theory. This theory assumes the memory index in the hippocampus does not store any information about experiences, which is in line with the authors' findings. If that is the case, what is the contribution of the allocentric spatial map in the hippocampus to episodic memory? I understand addressing this question might not be a scope of the current study, and the findings do not provide a conclusive view, but I think having a deeper discussion on this matter would interest more readers in the broader fields.

5. This is optional. Related to Fig6, I am very interested in how the egocentric cortical representation is influenced by LEC manipulation.

Some minor issues:

6. Line 99. I think "is essential for the cortical long-range gamma-synchrony" is a bit overstating because HPC-lesion substantially reduced the powers of the oscillations. This sentence needs to be softened, like "is essential for the cortical oscillations."

7. Line 707-708. "Four screws inserted above the cerebellum were used as reference and ground electrodes during recordings." Does this mean two of four screws?

8. Line 714. Typo "Carinal" -> "Cranial"

I don't see any severe issue in statistical analyses, and the level of the detail is sufficient enough for researchers to reproduce the study (except for a few described above).

Reviewer #1 (Remarks to the Author):

The authors showed that spatial learning and retrieval induce cortical theta-gamma coupling in RSC, MO, and VIS. Strikingly, optogenetically inducing this coupling by stimulating LEC cell bodies or projections to the cortex during training restored spatial memory in HPC-lesioned mice. This indicates that we can bypass HPC by stimulating the LEC-cortex circuit. The strength of the manuscript is that the authors performed an impressive array of behavior and techniques. It has the potential of high interest in the learning and memory field. However, the manuscript is very poorly written, and I had to read it multiple times to figure out when, how, and from which structure the data were collected. LFP recordings need to be validated. Also, there are several logical gaps between data presentation. Unless these gaps are filled, this is a very impressive shopping list of data that are not tightly held together for a complete story for publication.

Thanks a lot for your supportive and insightful comments. According to these, we revised the manuscript to clearly demonstrate the experiments. We also have added some new experiments and data to address these concerns. New experiments and data further confirm our conclusion and filled some logical gaps between data presentations.

Major concerns:

1. Fig.1d. VIS, RSC, and MO show different ranges of amplitude (power) of oscillations. However, all raw power plots from each structure provided throughout the paper are in the same range. Also, all SEM of the average of all 3 structures throughout the paper is negligible (e.g., Fig.1e). How is this possible?

We are sorry for the confusion brought by the inappropriate way of data presentation. Due to the limited space of the manuscript, we did not show the comparison of LFP powers between regions and only compared LFP power among home, training, and recall. To correct it, we now show all the raw data including LFP power (3 regions and all kinds of trials in all days) and compared between regions. Please see the figure below (**Figure 1 to reviewers**, already integrated into Supplementary Fig.1). Statically, we found that RSC (green) showed the strongest and MO (red) showed the weakest overall power across all conditions (Home, Training day1-4 and Recall), but noted that not all mice showed the same power differences, some mice had the

Figure 1 to reviewers (Supplementary Fig. 1c). c, Raw LFP powers. Significance was assessed by two-way ANOVA followed by false discovery rate (FDR) corrected multiple comparisons between regions, corresponding, P values were noted on the figure.

strongest brain wave in the visual cortex, while some had the strongest wave in the motor cortex.

However, we did find all three brain regions showed the same power elevation in the frequency band of theta and gamma during training and recall trials, therefore we pooled data together as cortical power and showed it in the main figure (Fig.1e) and showed LFP of separated single structure in the supplementary figure. Although there is some individual variance, we have 26 electrodes for each region from 28 mice (in total 78 electrodes to plot Fig.1e now), and the power was measured in dB unit ($10 \cdot \log_{10}$), that's probably why the SEM is small in the figure.

2. Related to the first comment, LFP recordings suffer from volume conduction and are coarse measures of neural activities. The fact that LFPs from all cortical structures were very similar and the latency of evoked signals from all cortical structures after optogenetic stimulation were all in the same range (supp. Fig. 5), I suspect volume conduction. Another evidence of cortical volume conduction is Supp. Fig. 4b. Here, power in LEC and HPC have much bigger SEM, considering the scale of the graph. Ideally, the authors should have done single-unit recordings instead of LFP recordings. However, it will be impractical, and I ask the authors do something to ensure that their data are not affected by volume conduction.

Thank you very much for this important comment, LFP recording is thought to be suffering from volume conduction. Yet, the effect on the LFP from volume conduction varies with recording conditions. e.g., the placement of electrodes, screws number and positions¹⁻⁴. To minimize this effect, we put 2 ground screws and 2 reference screws upon the transverse sinus (supposed to make the electric field of the dipole weaker and more dispersive, and less affected by electric potential from screws) and made the recording sites far from each other ($\geq 2\text{mm}$). To access the effect of volume conduction, we directly induced a current sink in one of five brain regions (CA1, LEC, RSC, VIS, MO) and quantified the evoke amplitude (volume conduction) of potentials in other brain regions, please see the result on the figure below (**Figure 2 to reviewers**).

Figure 2 to reviewers (Supplementary Fig. 7). Volume conduction has little effect on multi-regional recoding LFP data. **a**, Experimental diagram. One of five regions was electrically stimulated and the effect of volume conduction was quantified in other brain regions during mice awake in home cage. Stimulation voltage: 6V DC 10ms, 2Hz. Screws was implant upon the transverse sinus. We chose transverse sinus for reference screw because the position below is less neuron and more vascular tissue. **b**, Top, averaged raw LFP traces in 5 brain regions with one of them was stimulated. Bottom, quantification of normalized evoke potentials in upper panel. 50 electrodes from 10 mice.

We found that less than 1% of the source LFP could be detected in distant regions, these data suggest that the recorded LFP signal mainly represents the local activities on our recording system but not by volume conduction. This part of the data has been wrapped into supplementary figure 7 now.

- Also, LFPs are affected by behavior. For example, when mice are digging, there are huge jitters in LFPs. Thus, all LFP analyses should be done in a stereotypical behavior (e.g., running in a certain range of speed in one direction, not exploring objects).

Thanks for this suggestion, we had considered these issues and made improvements during our study. We found that the observational jitters and 50Hz line noise in LFPs could only be detected in those mice with poor ground or reference connection (screws did not fully touch the brain fluid). Here are two LFP trace examples to show the noise in the spatial memory task (**Figure to 3 reviewers**). In practice, mice with large moving artifacts or 50Hz line noise were excluded from the final dataset.

For the following LFP data analysis, we divided the trial into several parts: 1. Preparing, on the start region. 2. Exploring, from the timepoint of getting off the start block to finally food eating. 3. Food eating. We found some major LFP differences between these parts, there was little theta power in preparing and stronger delta power when mice were eating the food. During exploring, LFPs showed a similar power pattern with higher theta and gamma, mice explored the environment and searched for the food actively, which is good period to study memory encoding. Therefore, to guarantee that the LFP measures were collected as mice behaved similarly, only the exploring phase was employed to calculate synchrony and power and other analysis. We did

Figure 3 to reviewers only. Examples of LFP data. a, Low signal to noise LFP example. **b,** High signal to noise example. LFP data was from training day3 and training day2, respectively. Top, behavioral a timeline. Middle, raw LFPs of three brain regions. Bottom, power spectrograms of three brain regions.

not observe any jitters or other noise during the exploring in these mice. To make it LFP analysis and selection more clear, these details were added to the method section “LFPs data collection and analysis.”

However, the moving speed during the exploring is correlated with LFP power and cortical phase synchrony (Figure 4 to reviewers, also available in Supplementary Fig. 1e-f). We compared the power and synchrony under different moving speeds. Consistent with results reported previously⁵, we found that the cortical LFP theta powers were correlated to the locomotion of mice, as well as the cortical high theta synchrony. However, the low gamma synchronies were not affected by the moving speed. Thus, gamma synchrony is comparable in all three kinds of trials. And it may actually be correlated with the cognitive process. That is also one of the reasons why we focused on the synchrony in the gamma band in the following study, instead of the power of the oscillation.

Figure 4 to reviewers (Supplementary Fig. 1e-f). **e**, Curves of cortical LFP power with respect to the moving speed of the animal ($n = 41$ electrode). **f**, Curves of cortical LFP synchrony with respect to the moving speed of the animal ($n = 40$ electrode pairs). Low θ and low γ synchronization were not affected by the locomotion of mice. All statistical significances were assessed one-way ANOVA. Statistical parameters are noted on the graph in the corresponding color.

4. Fig.2l-n Were these done in home cage? This should be done in a stereotypical behavior.

Thank you for pointing out this. Yes, these experiments (now Fig. 2m-o). were done in homecage. The missing experimental detail is now added to the figure legend and method part. To directly address this concern, we have tested a group of mice (also hippocampus lesioned) both in homecage and in sandboxA (Figure 5 to reviewers). We found that activation of LEC layer 5 neurons was able to induce synchronized potential in three cortical regions. Thus, even

Figure 5 to reviewers only. Evoke potentials by activation of LEC neurons in homecage or during exploring in the sandboxA. Line shows averaged evoke potential from 13 brain regions in 5 mice. Blue bar indicate the activation time period, 4ms.

in stereotypical behavior conditions, optogenetic activation of L5 neurons in LEC engaged similar effects as those in homecage. The reason why we choose to activate LEC neurons in homecage is that we want to minimize the task-engaged cortical synchrony to detect the pure effect of optogenetic stimulation. Data in Fig.1-2 suggested that gamma synchrony increase when mice are exploring in the context and this elevation is hippocampal-dependent. To alleviate the endogenous synchrony elevation and measure the pure LEC activation effect on cortical synchrony, this test was done in homecage using HPC-lesioned mice.

5. Were training data collected on Day4? To claim that cortical theta-gamma coupling is induced by learning, authors should compare the coupling between Day1 and Day4 to show that it evolves through learning.

The data showed in the figure was collected on training day1, we have collected LFP data of all training days, this missing detail was added into the corresponding legend now. Follow your suggestion, we analyzed and showed these data in the manuscript now, (Figure to reviewers 6-

Figure to reviewers 6. (Supplementary Fig. 1c-d). Raw LFP power and synchrony of HPC intact mice. **c**, Raw LFP power. Two-way ANOVA. **d**, Raw LFP synchrony. Greyline, homecage synchrony for comparison with training and recall trials, Two-way ANOVA. For more details, please see supplementary figures.

Figure to reviewers 7. (Supplementary Fig. 2c-d). Raw LFP power and synchrony of HPC lesion mice. **c**, Raw LFP power. Two-way ANOVA. **d**, Raw LFP synchrony. Greyline, homecage synchrony for comparison with training and recall trials, Two-way ANOVA. For more details, please see supplementary figures.

Figure to reviewers 8 (Supplementary Fig. 3). The coupling between cortical gamma power and theta phase elevated during memory encoding and recall, which is hippocampal-dependent.

a, Results for HPC intact mice. **b**, Results for HPC lesion mice. Top, averaged theta wave (black) and 30Hz power (red). Middle, averaged phase(θ)-power(γ) spectrogram from all regions. Bottom, phase-power modulation index comodulograms of all regions. Modulation indexes were normalized by the element wise division of the raw comodulograms by surrogated control. **c**, Quantification of modulation index in a and b. Significance measured by one-way ANOVA, * $P < 0.05$, ** $P < 0.01$, n.s., not significant. For more details, please see the supplementary figure legend.

7, power and synchrony; **Figure to reviewers 8**, gamma power-theta phase coupling). We found that the theta-gamma coupling strength decreased gradually with training days and it became the same level as training day1 in recall day, which cannot be observed at all in HPC-lesioned mice indicating this coupling is also hippocampal-dependent. One possible idea for the coupling decrease during training days is that the mice gradually learn where the food is and are less dependent on HPC-LEC. But in recall trials, once mice fail to find the food in the recall trail, they have to actively recall the location of the food and it perhaps needs some involvements from HPC and LEC, leading to a high theta-gamma modulation index again. This is interesting and we will look into it in the next research paper.

6. Fig.3. Why not LEC cell body stimulation? Because Fig2. is showing cell body stimulation, it is logical to stick to cell body stimulation. Or, authors should show that fiber stimulation also induces theta-gamma coupling.

The reason we chose to activate cortical fiber but not the LEC cell body in Fig3 experiment is that activation of fibers could avoid the unspecific LEC downstream activation. In fact, except cortical regions, LEC is also projected to other brain regions like basolateral amygdala,

Figure 9 to reviewers only. LEC projects to various regions in the brain. a, Fluorescent source. **b**, 3D demonstration of the fiber projection. **c**, Projection to basolateral amygdala and caudoputamen. **d**, Projection to accumbens. Images is from Allen Brain database. <https://atlas.brain-map.org/> Experiment number: 232311959.

caudoputamen, accumbens and so on (**Figure 9 to reviewers**, from Allen Brain database). LEC may also mediate the synchronization between these regions. However, to be more specific, we aimed to focus on one circuit (LEC-neocortex) in this study, thus we chose activation cortical fibers. We modified the main text part and the legend, accordingly.

When simultaneously stimulating cortical fibers and recording electrical signals at that location, cortical LFP signal is sometimes affected by light stimulation artifact (~40% cases were excluded and caused a small N number, the amplitude of the noise depends on the relative location between the optical fiber and the electrode), which may affect the calculation and the interpretation of results, but we still can clearly observe the induced synchronized potentials (Supplementary Fig. 9a-b). In contrast, evoked cortical potentials by cell body activation showed no light artifact (Fig. 2l-o). Therefore, to be more clear and rigorous, we show the cell body stimulation data on the main figure 2 and the fiber stimulation traces in supplementary data 9.

The artificial-induced gamma wave by activating the cell body or cortical fiber is supposed to show no theta-gamma coupling because evoked potentials are knocked to the stimulus onset but not theta phase. However, the evoked gamma synchronization is still able to help HPC-lesion mice to form new memory, indicating the gamma synchronization is more important than the theta-gamma coupling.

7. **Fig.4. The authors claim that theta from the HPC/LEC modulates cortical theta that couples with cortical gamma. Also, Supp. Fig. 4c shows that LEC-MO synchronizes at theta, but not gamma. Therefore, cortical stimulation should be done in theta (10 Hz), not in gamma. Same thing for Supp. Fig. 7e.**

This is an essential question. In the experiment shown in main figure 2l-o, we show that 30Hz stimulation is able to induce cortical synchrony in corresponding frequency, that is 30Hz. Interestingly, when we activated LEC neurons with 10Hz, we found that not only 10Hz synchronization was induced but also the synchronization in low gamma band (20-40Hz) was induced (**Figure 10 to reviewers**, or see main figure 2l-o). Furthermore, the evoked cortical potential has a period around 36 ms, corresponding to a low gamma frequency 28Hz. In that case, though the LEC sends input to cortex in theta rhythm, it would induce not only theta synchronization but also gamma synchronization between cortices.

We then tried 10Hz as well as 30Hz cortical stimulation to rescue the fear memory defect or spatial memory deficit in HPC-lesion mice (**Figure 11 to reviewers**, or see main figure 3). We find that both 10Hz and 30Hz stimulation can rescue the memory defect. When 30Hz was applied, only gamma and no theta synchronization was induced, but it was still able to rescue the memory defect. Again, the induced gamma wave by artificial activation of cortical fiber here is not coupled to theta phase because the evoked potential is knocked to the stimulus onset but not the endogenous theta. Thereby, we argue that the gamma synchronization itself is more crucial than the theta-gamma coupling or theta synchronization during memory encoding and recall. Although gamma synchronization is important, it works by entrainment in theta rhythm in normal conditions.

Figure 10 to reviewers (Main figure 2l-o). l, 470 nm laser-activated neurons expressing oChIEF-mCherry in LEC (4 ms per pulse) induced synchronization oscillatory signals (iSOS) of LFP in multiple cortical areas simultaneously. m, Blue light stimulation at either 10Hz or 30Hz in LEC cell body could induce LFP responses simultaneously in MO, VIS and RSC in oChIEF-expressing mice (green lines) but not EYFP expressing mice. See surface axonal stimuli in Supplementary Fig. 5. n, Cortical synchrony heatmap before and after stimulation. o, Quantification of panel n. Baseline, averaged PLV before laser stimulation. see complete legend in the manuscript.

Figure 11 to reviewers (Main figure 3). Cortical application of iSOS rescued memory acquisition and retrieval deficits in HPC-lesioned mice. a-e, Artificial iSOS in neocortex during training can rescue fear memory deficits in HPC-lesioned mice. f-j, Artificial iSOS rescued spatial memory deficit in HPC-lesioned mice. Please see complete legend detail in the manuscript. Red arrows indicate the experimental data of rescuing HPC-lesion mice memory deficit by 10Hz LEC cortical fiber activation.

8. Fig.5 & 6. These are very impressive data. However, they are not well tied together with the rest of the data and can be used for a separate paper. Fig. 1-4 focuses on cortical theta-gamma coupling during training (especially with optogenetic stimulation). However, Fig. 5 & 6 are showing cortical calcium activity during memory recall (testing) and focuses on cortical gamma synchrony during pre-testing. To fill this gap, the authors should show whether asynchronous cortical theta stimulation (as in Fig.4) or inhibition of LEC during 'training' disrupts engram activity during 'testing'.

Thanks for your appreciation of our studies. We want to add the cellular mechanisms underlying gamma synchronization and memory encoding/recall into this manuscript. Here we have new experiments to directly address the concerns. As suggested, we trained a group of mice with inhibited LEC during memory encoding (by hm4d + CNO, 4 training days) and labeled activated cells on training day4, and we assessed its behavioral change and engram activity during testing (**Figure 12 to reviewers**). The results indicated that LEC is essential for memory encoding, as the mice with LEC inhibited showed poor memory retrieval on the last day. This indicates that LEC plays an important role in the formation of accurate memory. Without the LEC activity, mice could not retain the memory of the food location.

In this case, while some neurons were successfully labeled with GCaMP6, these neurons were activated and CEs did not show preference to the trained context (**Figure 12 to reviewers**).

Figure 12 to reviewers (Supplementary Fig. 19). Inhibition of LEC during training disrupts engram activity during recall. **a**, Experimental design. LEC was infected by hSyn-hM4D(Gi)-mCherry, CNO dosage: 2 mg/kg. **b**, Learning curves. (N = 5 mice). **c**, Quantification of occupancy in food zone, the normal mice group presented is the same as showed in Fig. 3g&i, it's for comparison here ($N_{LEC\ inhibition} = 5$, $N_{Normal\ mice} = 24$, unpaired t-test, $t(27) = 3.651$, $P = 0.0011$). **d**, CEs frequency on day11. Neither labelled neurons in RSC nor in VIS showed selectivity to sandbox. ($N_{RSC} = 5$; one-way ANOVA, $F(3, 12) = 19.8$, $P < 0.001$. Tukery post-hoc test, $P_{Home\ vs.\ Pre-exploring} = 0.0004$, $P_{Home\ vs.\ Exploring} = 0.0064$, $P_{CtxB\ vs.\ Pre-exploring} = 0.0002$, $P_{CtxB\ vs.\ Exploring} = 0.0027$. $N_{VIS} = 5$; one-way ANOVA, $F(3, 12) = 2.5$, $P = 0.1051$).

The absence of LEC disrupts the formation of context-selective engram in RSC, suggesting that LEC plays an important role in memory encoding. This part of the data has been wrapped into the manuscript and described in the main text now.

Also, does theta-gamma coupling modulate cortical gamma synchrony?

Although we have demonstrated that the cortical gamma synchronies were coupled to the cortical theta wave, we did not have direct evidence showing that the cortical theta oscillation LFP has impact on the gamma synchrony. However, based on our finds that LEC 10Hz activation is not only can induce 10Hz synchrony but also gamma synchrony. These data indirectly support the possibility that the cortical gamma synchrony and cortical theta may be periodically induced by LEC in theta rhythm and the LEC theta was modulated by HPC theta (please see the answer for question 13-14).

Finally, these are fiber photometry data of collective calcium activity of the population of neurons. Conventionally, engram means single neurons encoding memory. I would not call these collective calcium activity engrams.

We agree, to be more rigorous, we have changed the term “engram” into “c-Fos labeled neurons” or “memory-related population” in the manuscript.

9. Related to #8, gamma modulated calcium activity occurs during pre-testing. Then, what does it have to do with egocentric mapping during testing? This logical gap needs to be filled.

Remarkably, activation of those neurons was detected during the pre-exploring period, when no landmark cues were presented. It is likely reflecting the active memory retrieval of the spatial information as inhibition of LEC/HPC complex, which impairs memory retrieval, significantly reduced cortical engram activities before the door opening. In addition, they were not activated in the novel context. Thereby, the calcium activity of those c-Fos labeled neurons, which showed egocentric map coding during spatial task, is not only activated by those landmark cues, but also activated by HPC-LEC associated memory recall/replay events in the pre-testing phase in the sandbox.

While it is an interesting phenomenon we plan to leave it for further studies, for now, we added a discussion part on this topic to further elucidate the correlation: *“Remarkably, activation of those neurons was detected during the pre-exploring period, when no landmark cues were presented. It is likely reflecting the active memory retrieval of the spatial information as inhibition of LEC/HPC complex, which impairs memory retrieval, significantly reduced cortical engram activities before the door opening.”*

10. Fig.5b-d. Are calcium signals between VIS and RSC the same? Why only RSC is shown?

Now the VIS data was included also in this version. They are not the same, but VIS calcium signals own similar properties as RSC, their signals showed the most frequent firing in the pre-exploring phase and are coupled to long-range low gamma synchronization. RSC was thought to be more related to memory function in the mouse brain and to save room, thus we put the RSC

data on the main figure and put the VIS data on the supplementary figure. To be more clear, we added some text into the legend of figure 5: “VIS calcium signal owns similar properties as RSC, both RSC and VIS signals show most frequent firing in pre-exploring phase and coupled to long-range low gamma synchronization.” See VIS data in **Figure 13 to reviewers** (Supple. Fig. 13.)

Figure 13 to reviewers (Supplementary Fig 13). Calcium activity of labelled neurons in VIS was coupled to cortical synchrony during spatial memory recall. a, Examples of calcium signals from labelled neurons in VIS in various contexts and its selectivity. **b**, An example of γ synchrony spectrogram (pair of RSC-VIS) plotting together with calcium signal from VIS engram, showing the correspondence between calcium activity and γ synchrony. **c**, Cross-correlation analysis. **d**, Averaged synchrony (RSC-VIS) spectrogram aligned to the peak of VIS calcium events. For more details, please see the legend in supplementary Figures 13.

11. If cortical neurons indeed form an egocentric map, this can be tested. In experiments in Fig3f, test mice in the same box with altered object locations. If mice are using an egocentric map strengthened by optogenetic stimulation, this change will impair their performance.

Thanks for the suggestion. We performed control experiments according to this suggestion. A group of mice was trained with the same paradigm in Fig.1a (learn to find food in A for 4 days) followed by a performance test in the same box with altered object location (A'), please see the result on the following figures (**Figure 14 to reviewers**, Supplementary Fig. 16). The result showed that altering the object location largely decreased the foodzone occupancy in test trials. This indicated that locations of objects are critical for mice to locate the food reward. Noted that the context has little allocentric information (unlike water maze with signs on the wall) so that mice have to rely on their relative position (egocentric information) to objects to locate the food. This part of the data has been added to the manuscript (Supplementary Fig. 16) and described in the main text now.

Figure 14 to reviewers (Supplementary Fig. 16). Change the position of objects in sandbox largely impair spatial memory performance. a, Left, Two images shows the original sandbox (A) and the changed object sandbox (A'). Right, behavioral paradigm. **b,** Left, occupancy in foodzone. (N = 8 mice, paired t-test, $t(7) = 5.7$, $P = 0.0008$). Right, the average occupancy map of recall in A and in A'.

12. Supp.1f. This should show theta-gamma coupling in each structure.

You might be referring to indicating Fig.1f, showing that the average power-phase coupling of three brain regions. Please find the data in answer to question 5. We added these figures of theta-gamma coupling in each structure in Supplementary Fig. 3 of the new version (or see **figure 8 to reviewers**). We found that theta-gamma coupling in three structures showed similar elevation during training day1, and gradually decreased along with training days and then elevated again in recall trials, so we pooled the data together as the cortical theta-gamma coupling shown in Fig. 1f and Fig. 2f.

13. The biggest gap in the logic is the switch from HPC to LEC. After showing data in HPC lesion, the authors suddenly switched to LEC. To fill the gap, the authors should show that HPC leads/modulates LEC theta in the main figure to link it to Fig. 3. showing that LEC theta stimulation can bypass HPC.

Thanks for the valuable suggestion. We now have added new experiments and analyses to fill this gap. We followed the suggestion and perform the theta LFP amplitude-based lag analysis⁶ of all brain regions pairs, please see the result in the figure below (**Figure 15 to reviewers**). This result suggests that HPC theta leads the LEC theta and LEC theta leads cortical theta especially in training day1, which is supported by the neural circuit from HPC CA1 to LEC deep layers⁷⁻⁹

(Figure 16 to reviewers) and the circuit from LEC layer 5 to neocortex (our main figure 2). This result is also consistent with previous functional researches of the circuit from HPC-LEC, which reported that the HPC and LEC communicate with each other in theta and it's critical to memory encoding and recall^{8, 10}. We have added some additional description of this result to fill the logical gap in the main text now.

Figure 15 to reviewers (Figure 2j & Supplementary Fig. 6). HPC theta leads LEC theta and LEC theta leads cortical theta. a, Theta wave amplitude base cross-correlation analysis of all learning stages and all pairs between regions. Statistical significances were assessed by Wilcoxon signed-rank test, comparing to 0. $N_{MO-RSC} = 19$, $N_{RSC-VIS} = 19$, $N_{VIS-MO} = 19$, $N_{LEC-MO} = 18$, $N_{LEC-RSC} = 18$, $N_{LEC-VIS} = 18$, $N_{HPC-MO} = 17$, $N_{HPC-RSC} = 17$, $N_{HPC-VIS} = 17$, $N_{HPC-LEC} = 16$ from 19 mice. * $P < 0.05$, ** $P < 0.01$, *** $P < 0.001$

14. Related to #13, does HPC directly project to the cortical area? Can we directly manipulate HPC projections? Why should LEC relay the signal?

The HPC does not have widespread projections to neocortex. According to the mouse brain connectivity database provided by Allen Brain, there are only a few projections to ventral RSC from dorsal CA1 and some projections to prefrontal cortex (PFC) from ventral CA1 (Figure 16 to reviewers). Nevertheless, both dorsal CA1 and ventral CA1 projected to LEC deep layers, and the LEC layer 5, shows widespread projections to neocortex so that makes LEC an ideal candidate for mediating synchronization between multiple cortical regions. That's the reason why the LEC draw our attention. We now modified the main texts to state the logic. In addition, our LEC inhibition experiments have demonstrated that LEC is important for both memory encoding

Figure 16 to reviewers only. HPC does not have widespread projections to neocortex but has some projection to mPFC, vRSC and LEC. a-d, Projection from dorsal CA1 of HPC. a, Fluorescent source. b. 3D demonstration of the fiber projection. c, Projection to ventral RSC. d, Projection to dorsal LEC. e-f. Projection from ventral CA1 of HPC. e, Fluorescent source. f. 3D demonstration of the fiber projection. g, Projection to ventral PFC. h, Projection to ventral LEC. Images is from Allen Brain database. <https://atlas.brain-map.org/> Experiment number: 232311959 & 286610923.

(Supplementary Fig. 19) and recall (Fig.6), indicating it may be the memory mediator between HPC and cortex.

15. Supp Fig4. Are these done on day 4? It will be better to show the signal evolves through day1-4.

The experiment was performed with the same behavioral protocol as in Fig. 1a. The data used to plot supplementary 4 is from naive (Home), training day1, and recall. Now we show all data from all training days, please see the figure below (**Figure 17-18 to reviewers**, supplementary figure 4-5). Consistent with previous results shown in answer to question 5, we found that the coupling between cortical gamma and LEC theta decreased along with training days. This coupling was always stronger than the coupling between LEC gamma and cortical theta, indicating the cortical gamma was modulated by LEC theta but not vice versa (**Figure 18 to reviewers**).

Figure 17 to reviewers (Supplementary Fig. 4). The evolution of power of LEC/HPC (CA1) and synchrony between HPC/LEC and various cortices. a, The evolution of power of LEC and HPC. **b**, The evolution of synchrony between LEC/HPC and cortices. **b**, Synchrony spectrum between HPC/LEC and cortices. $N_{MO-RSC} = 19$, $N_{RSC-VIS} = 19$, $N_{VIS-MO} = 19$, $N_{LEC-MO} = 18$, $N_{LEC-RSC} = 18$, $N_{LEC-VIS} = 18$, $N_{HPC-MO} = 17$, $N_{HPC-RSC} = 17$, $N_{HPC-VIS} = 17$, $N_{HPC-LEC} = 16$ from 19 mice. **c**, Training (day1) and recall trials showed significant theta power elevation in LEC, significant theta and gamma power elevation in HPC. **d**, Averaged synchrony between LEC/HPC and all cortical regions. All significance was assessed by two-way ANOVA followed by false discovery rate (FDR) corrected multiple comparisons at each frequency comparing with data of homecage trail. Significant frequency range ($q < 0.05$) is noted on the graph. # $q < 0.05$, Shadow of line plot shows S.E.M.

Figure 18 to reviewers (Supplementary Fig. 5). Cortical gamma powers were modulated by LEC theta phase. **a**, Phase-power modulation index comodulograms between cortices and LEC (For all pairs, N = 18). The modulation indexes between cortical gamma and LEC theta were stronger than the indexes between LEC gamma and cortical theta, indicating the cortical gamma was modulated by LEC theta but not vice versa. **b**, Quantification of modulation index. Significance measured by one-way ANOVA comparing with data of homecage trail, *P < 0.05, **P < 0.01, ***P < 0.001, n.s., not significant.

16. Supp. Fig. 9e (now Supp. Fig. 12e). It looks like there are more CEs during freezing, contradictory to Supp. Fig. 9d (now Supp. Fig. 12d) showing that there are more CEs during pre-freezing

This is not intuitive, because the freezing time is much longer than the pre-freezing time, while the pre-freezing time was defined as 3s just before a freezing behavior. Consequently, though there is more CE during freezing behavior, the frequency of CE during pre-freezing can be bigger than the frequency during freezing.

17. Line 79-80. The location of theta trough and gamma peak is relative. Depending on where you start to look, either one leads the other. Lag analyses should be done to claim this. Line 79-80: *Troughs of the θ wave were always preceding to the peak of γ synchrony, implicating the θ wave might lead the cross-regional γ synchrony (Fig. 1g).*

Because synchrony evaluation needs the average phase difference between regions from more than one timepoint, that average makes it unable to obtain the instantaneous synchrony like power or phase (For more details, please refer to the method section of LFPs data collection and analysis). Therefore, it's hard to perform the lag analysis like LFP amplitude as we mentioned in question 13.

Alternatively, we analyzed latency between theta trough and the peak of the gamma synchrony base on the PLV spectrogram (Figure 19 to reviewers, also is the Supplementary Fig. 9j-k now). This latency is about 28ms, if calculated from the opposite direction, it would be about $125-28 = 97\text{ms}$ (one theta cycle is about 125ms) from the gamma synchrony peak to the next theta trough, usually speculating the direction that interact with each other with a shorter latency. However, only based on data shown on Fig.1g and these analyses do not prove that theta wave

Figure 19 to reviewers (Supplementary Fig. 9j-k), j-k, iSOS produced PLV rhythmicity that was similar to endogenous theta-modulated PLV rhythmicity. **j**, Illustrations of peak latency of low γ and high γ PLV, relative to theta trough (top) and LEC neuron stimulation onset (bottom). **k**, Comparison of these peak PLV latency. (In training trials: latency from all 72 PLV pair; low γ vs. high γ , paired t-test, $t(71) = 8.1$, $P < 0.0001$; In recall trials: latency from same 72 PLV pair; low γ vs. high γ , paired t-test, $t(71) = 6.2$, $P < 0.0001$; iSOS stimulation: low γ latency from same 16 PLV pair; Training low γ vs. recall low γ vs. iSOS low γ , one-way ANOVA, $F(2, 157) = 0.66$, $P = 0.5173$, no significance is detected). Error bars show mean \pm S.E.M. n.s., no significance. Shadow of line plot shows S.E.M.

has the casual or leading effect on gamma synchrony. But to a certain extent, these data indirectly support the idea that the cortical gamma synchrony and cortical theta may be periodically induced by LEC in theta rhythm and the LEC theta was modulated by HPC theta. To avoid misleading, we deleted this sentence.

18. Line 124-127. This is critical information and should be presented in a figure. Show correlation for HPC-RSC, HPC-VIS, HPC-MO, LEC-RSC, LEC-VIS, and LEC-MO. Line 124-127: “*In addition, the cross-correlation analysis showed that θ rhythm in LEC/HPC occurred earlier than that in RSC in the training trials (HPC-RSC: -8.12 ± 2.6 ms, mean \pm s.e.m), suggesting θ rhythm signal in HPC/LEC might modulate cortical γ synchrony during learning and memory retrieval.*”

Thanks for this suggestion, we showed all results of the LFP amplitude cross-correlation analysis between mentioned pairs in the main figure 2j, or you can see the figure below (**Figure to reviewers 20**).

Figure 20 to reviewers (Main figure 2j) HPC theta leads LEC theta and LEC theta leads cortical theta. **j**, Top, Experimental diagram show simultaneous recording of 5 brain regions (MO, RSC, VIS, LEC, CA1) during spatial memory training as show in Figure 1a. Recorded LFP was used for theta LFP amplitude based cross-correlation lag analysis. Bottom, lag summary (training day1), see lag summary of other learning state in supplementary figure 6. corresponding number in the graph note the max-lag correlation coefficient r . For more details, please see the figure legend in the manuscript

19. HPC lesion. Line 726. “*behavioral assessments....were performed immediately after surgery to determine the effect of hippocampal lesion.*” Aren’t the learning task and recordings done 2 weeks after surgery? This should also be done 2 weeks after surgery.

Sorry for making confusion here. The “behavioral assessment” here means that the

behavioral observation immediately after the NMDA injection to test if the NMDA was injected successfully. Mice show seizure behavior and abnormal mobility after hippocampal NMDA injection, by this we can evaluate if the NMDA was injected into the target point, once the mouse show seizure behavior, it was put back under anesthesia for 3 hours and then recover in the homecage for 2 weeks. The learning and memory behavioral experiment was performed after recovery.

We have revised the method section to make it clear: "... Injection of 0.25 μ l was given in 2 minutes at each site. Mice show seizure behavior and abnormal mobility after NMDA injection, as the evaluation whether the NMDA was injected successfully. Once the mouse show seizure behavior, it was put back under anesthesia for 3 hours and then recover in the homecage for 2 weeks. The learning and memory behavioral experiment was performed after recovery."

Minor concerns:

1. Fig. 2l. (now Fig. 2m) Is this the average of RSC, MO, and VIS recordings after LEC cell body stimulation?

Yes, Fig. 2l shows the averaged curves of RSC, MO, and VIS recordings after LEC cell body stimulation, and the average curves after LEC cortical fiber activation are shown in Supplementary Fig. 9a-b. To be more clear, we revised the corresponding legend: "Fig.2m... including electrodes in RSC, MO, and VIS, lines are the averaged evoke potential of RSC, MO, and VIS".

2. Fig. 2m&n (now Fig. 2n&o). Are these the average of RSC-MO, RSC-VIS, and VIS-MO after LEC cell body stimulation?

Yes, Fig. 2m&n shows the averaged synchrony spectrogram of all pairs after LEC cell body stimulation. To be more clear, we revised the corresponding legend: "...Fig. 2n&o... The spectrogram is the average of all cortical pairs including, RSC-MO, RSC-VIS, and VIS-MO...".

3. Supp. Fig. 5l (now Supp. Fig. 9j-k). Show individual data points.

Thank you for your suggestion, we have revised this figure and show every individual data point on it (**Figure 19 to reviewers**), we also fixed all other figures without individual data points in the manuscript.

4. Supp. Fig. 9b (now Supp. Fig 12b). It looks like the first recall was 7 days after training. When did the second recall happen?

The memory-related cell was labeled on day 1 (CFC training). Recalls were in 7 days after TRAP labeling (day8), 1st recall and the 2nd recall were performed on the same day, the second one was performed 10mins after the first one and the 1st recall was tested on day 8. To be more clear, the relevant information has been added into the methods section of the Contextual Fear Memory Task and the figure.

5. Supp. Fig. 9g (now Supp. Fig 12g). When were the cells labeled?.

Basically, it's the same as the previous question. The memory-related cell was labeled on

day 1 (CFC training). TM was injected 24hours before CFC (day0, It takes 24 hours for TM to metabolize into 4-OHT). Memory recall trials were done in 7 days after labeling (day8). To be clear, this missing detail was already added into the method section and written on the figure.

6. Figures are not described in the text in order. It is better not to have readers go back and forth. There are many typos and grammar errors.

Thank you very much for the careful reading, we have rearranged figures to avoid such situations, and have checked and revised throughout the manuscript to eliminate grammatical errors, typos and the manuscript was refined and polished by a professional English polishing agency.

7. Line 95. Fig. 2e is not described.

Thanks for your reminder, we have added the description in the manuscript now: “we observed increased theta and gamma power during memory encoding and recall, but the oscillation powers were substantially reduced, when comparing to those in normal mice (Fig. 2e, Supplementary Fig. 2e).”

8. Line 119. Specify that dorsal HPC CA1 was targeted.

Electrodes were target to HPC CA1 region with this coordinate: AP, -2.0 mm; ML, -1.5 mm; DV, -1.3 mm, this information was already added into the main text.

9. Line 121-122. Why not showing cortical gamma power is coupled to HPC theta? Line 121-122: “Importantly, we found that cortical γ power in MO, RSC and VIS were coupled to the θ rhythm in LEC.”

In our opinion, HPC does not has many projections to cortices like LEC dose, therefore the cortical gamma is not likely modulated by the hippocampus directly so that coupling is not that meaningful as cortical gamma to LEC theta. To save room, we did not show the cortical gamma HPC theta coupling. Nevertheless, we had calculated the modulation index, please see the figure below (**Figure 21 to reviewers**, on the next page). The result shows that cortical gamma power was coupled to the hippocampal theta. Interestingly, the hippocampal gamma showed a much stronger coupling to cortical theta.

Figure 21 to reviewers only. Cortical gamma powers are coupled to HPC (CA1) theta phase. a, Phase-power modulation index comodulograms between cortices and HPC (For all pairs, N = 17). **b,** Quantification of modulation indexes. Significance measured by one-way ANOVA comparing with data of homecage trail, *P < 0.05, **P < 0.01, ***P < 0.001, n.s., not significant.

10. Line 142-143. This statement is not supported by the data. LEC was engaged with HPC at theta?
Line 142-143 :*“Apparently, when LEC was engaged with HPC at the θ rhythm, the long-range cortical synchrony showed strong θ coupled γ rhythm (Fig. 2o).”*

Followed your suggestion, this time we provided the critical information that HPC theta is leading LEC theta (**Figure 15-16 to reviewers**) showed that LEC theta may be regulated by HPC theta, but the detailed mechanism still need to be dissected. To avoid the overstatement and be more precise, we deleted this sentence.

11. Line 233. Overall -> random. Line 233: *“While overall neural activities of RSC did not show preference to the memorized context and were not coupled to the LFP synchrony between VIS and RSC...”*.

Thanks! fixed, the word ‘random’ is more rigorous. Now the sentence is changed to *“While random neural activities of RSC did not show preference to the memorized context ...”*

I am strongly against the notion that authors must do whatever the reviewer mentions to publish their work. Reviewers can be wrong. My comments are suggestions that can make the paper more logical and meaningful. If the authors feel that my suggestions are inappropriate or there are better ways to address my concerns, I am open to the authors' opinions.

Thank you very much for reading our response and we really appreciate your comments, that help us to improve the manuscript and organized it in a more logical way.

Reviewer #2 (Remarks to the Author):

"Acquiring new memories in neocortex of hippocampal-lesioned mice" is a potential tour-de-force, a combination of modern methods to address the classic problem of how the brain stores remembered information. Mice were trained to find food that was hidden under sand in a consistent place relative to several objects in a testing arena. Intact animals learned to find the food faster with several training trials, mice with neurotoxic lesions of the hippocampus were impaired. Local field potentials (LFPs) recorded simultaneously from the secondary motor, primary visual, and retrosplenial cortices revealed prominent theta and gamma oscillations with power and power-phase coherence that increased with training and recall. Both LFP measures were reduced by hippocampal lesions. Optogenetic stimulation of neurons in the lateral entorhinal cortex (LEC) evoked synchronized LFPs in the same regions, rescued performance in the just described spatial memory task as well as in contextual fear conditioning in rats with hippocampal lesions. Asynchronous optogenetic stimulation of LEC fibers innervating pairs of other cortical regions during training impaired spatial learning. Fiber photometry combined with TRAP (Targeted Recombination in Active Populations) detected calcium fluorescence in groups of retrosplenial and visual cortical neurons that were activated during learning. The tagged neurons were most active in the training context, and the activity was correlated with proximity and heading angle to the more prominent objects and where food had been hidden. Memory performance, LFP synchrony, and context triggered calcium signals were impaired by chemogenetic inactivation of LEC. Together, the results suggest that gamma synchronized activity across cortical modules that is normally conveyed by LEC and coordinated by hippocampal theta rhythm may be necessary and sufficient for encoding spatial memories.

Presentation errors and missing details make it difficult to evaluate the paper. Below I've listed a few examples of such errors.

Thank you much for your positive evaluation and useful comment on our work, encouraging us to keep improving our manuscript. We have fixed mentioned presentation errors and added missing details in the latest version.

1. More detail is needed about the behavioral correlates of the electrophysiological measurements, in particular how the recording data were selected for analysis, and how the methods guaranteed that the LFP measures were collected as mice behaved similarly, e.g. walking within a particular speed range, in different task stages (e.g. figure 1d-h) you know. What defined "naïve"? The only description in the methods is for the phase power spectrograms -- what about the other LFP measures?

Thank you for your notification, details are missing in the last version. This time we have added more details in the figure legends and method section of the manuscript. For the LFP data selection, mice with large moving artifacts or 50Hz line noise were excluded from the final dataset, only high signal to noise LFP was included, please refer to the answer to question 3 and **Figure 3 to reviewers** (on page 3).

For naive trials, it means a segment of LFP data from a 3-minute period when mice were awake in the homecage before spatial memory training (day0), while training trials were divided into several parts: 1. Preparing, on the start region. 2. Exploring, from the timepoint of getting off the start block to finally food eating. 3. Food eating. We found some major LFP differences between these parts, there was little theta power in preparing and stronger delta power when mice were eating the food. During exploring, LFPs showed a similar power pattern with higher theta and gamma, mice actively explored the environment and searched for food, which is good period to study memory encoding (**Figure 3 to reviewers**). Therefore, to guarantee that the LFP measures were collected as mice behaved similarly, only the exploring phase was employed for subsequent analysis including synchrony, power, modulation index, spectrograms and cross-correlation analysis. For recall trials (3.5 min, 210 s), it's a 210s LFP segment from timepoint of getting off the start block to the end of the trial, because there was no food reward in the sandbox.

However, the moving speed during the exploring was correlated with LFP power and cortical phase synchrony (**Figure 4 to reviewers**, also available in Supplementary Fig. 1e-f). Thus, we compared the power and synchrony under different moving speeds. Consistent with results reported previously⁵, we found that the cortical LFP theta powers were correlated to the locomotion of mice, as well as the cortical high theta synchrony. However, the low gamma synchrony was not affected by the moving speed. Thus, gamma synchrony is comparable in all three kinds of trials. And it may actually correlate to the cognitive process. That is also one of the reasons that we focused on the synchrony in the gamma band in the following study, instead of the power of the oscillation. The missing details of this part have been added into the method section: "LFPs data collection and analysis."

How is the occupancy in food zone % quantified (figure 2D)?

Basically, we identify the location of the mice by video recognition and then calculated the total time the mice spend in the foodzone to quantify the occupancy. Here is the entire video recognition process, please see the illustration below (**Figure 22 to reviewers**). 1. Frames were extracted every 0.2s from the video file and then resize to 1/4 size and changed to grayscale. Each frame was subtracted by a frame without mice (usually it is an average of all frames) to produce frame difference for the determination of the location of the mice. One recall trail in total is 210s (1050 frame) and starts when mice get off from the start block. 2. Each frame difference was binarized, the average of all points of x or y was determined as the mice body location. Head direction was determined by the Skeleton algorithm. The body position and head direction of the mice in each frame were checked manually to make sure they are correct. 3. The context was divided into 46 x 46 grid, the foodzone was defined as a 10 x 10 region around the food position. Each bin of occupancy map was calculated as the proportion of body positions in that bin. For the occupancy of the foodzone, the total number of frames in which the mouse is located in the foodzone is T_{foodzone} , and the total frame number of the trail is $T_{\text{all}} = 1050$. Therefore:

Figure 22 to reviewers only. Working flow of recognition of mice location and head direction for occupancy calculation and egocentric analysis. **a**, Video loading and frame extraction. **b**, Body position and head direction recognition. **c**, Example trajectory and occupancy map.

$$\text{Occupancy}_{\text{foodzone}} = 100\% \times \frac{T_{\text{foodzone}}}{T_{\text{all}}}$$

This calculation detail was already added to the method section of the latest version (Video recognition for occupancy and head direction).

The dimensions of the spatial memory testing apparatus should be in the methods section (46x46 cm was only in Supp. Fig. 1a).

Thanks. We have added some detailed information into the method section: "Mice were put into the box to forage sunflower seed as food reward in the TSE system (Multi-conditioning system, TSE system). The sandboxA and B were the same sizes as the mentioned openfield context

(46cm x 46cm x 40cm) but with different object and foodzone locations. There is no sign on the surrounding walls (A: identical pure white; B identical pure black) to provide allocentric information so that mice have to position the food by objects in the context (Supplementary Fig. 1a).”

- The statistic in several places are not appropriate. E.g. figure 2i claims that the HPC lesion reduces synchrony, but should be supported by a significant interaction between lesion group and training stage synchrony. The same issue holds for fig. 3i, supplemental figure 4j, supplemental figure 7h, supplemental figure 11h (ANOVA vs t-tests).

Thank you for pointing out these statistical problems. Following your suggestion, we reanalysis figure 2i using two-way ANOVA followed by Bonferroni’s multiple comparisons between training stages (home and training, home and recall). Here is the statistical parameter: $F_{\text{Interaction}}(2,208) = 9.56, P < 0.0001, F_{\text{Lesion vs. Intact}}(1,204) = 3.406, P = 0.0678; F_{\text{Naive vs. Training vs. Recall}}(2,208) = 5.705, P < 0.0001$. Bonferroni post-hoc test, HPC intact: $P_{\text{Training vs. Naive}} < 0.0001, P_{\text{Recall vs. Naive}} < 0.0001$. We found a significant interaction between the lesion group and the training stage, indicating the elevation of the gamma synchrony is hippocampal-dependent. Now the detailed statistic parameters have been reported in the figure legends (**Figure 23 to reviewers**).

Figure 23 to reviewers (Main figure 2i). Learning and recall induced low gamma synchrony elevation is hippocampal-dependent. Pair number of HPC intact group, N, naive. T, training(day1), R, recall. $N_{\text{MO-RSC}} = 24, N_{\text{RSC-VIS}} = 24, N_{\text{VIS-MO}} = 24$ from 28 mice. Pair number of HPC lesion group, $N_{\text{MO-RSC}} = 11, N_{\text{RSC-VIS}} = 12, N_{\text{VIS-MO}} = 11$ from 12 mice. Two-way ANOVA, $F_{\text{Interaction}}(2,208) = 9.56, P < 0.0001, F_{\text{Lesion vs. Intact}}(1,204) = 3.406, P = 0.0678; F_{\text{Naive vs. Training vs. Recall}}(2,208) = 5.705, P < 0.0001$. Bonferroni post-hoc test, HPC intact: $P_{\text{Training vs. Naive}} < 0.0001, P_{\text{Recall vs. Naive}} < 0.0001$.

The same issue holds for fig. 3i, supplemental figure 4j, supplemental figure 7h, supplemental figure 11h (ANOVA vs t-tests).

For the fig. 3i, it was statically assessed by one-way ANOVA with Bonferroni post hoc test, and the ‘EYFP is missing’ mention in comment 5 is fixed. Here are the statistical parameters: $F(4, 73) = 5.0, P = 0.0012$; Multiple comparison: $P_{\text{HPC-intact vs. HPC-lesion}} = 0.0048, P_{\text{ISOS (30Hz,EYFP) vs. HPC-lesion}} > 0.9999, P_{\text{ISOS(10Hz,oCHIEF) vs. HPC-lesion}} = 0.0243, P_{\text{ISOS (30Hz,oCHIEF) vs. HPC-lesion}} = 0.0098$.

For supplementary figure 7h (**Figure 24 to reviewers**, now is the supplementary figure 11h), and supplementary figure 11h (**Figure 25 to reviewers**, now is the supplementary figure 17h). Follow your suggestion, we reanalysis it using ANOVA instated of t-test. The corresponding legend descript was revised or you can see specific statistical parameters details on the figure legend of the figures to reviewers.

h

Figure 24 to reviewers, now is the supplementary figure 11h. h, Memory test (Same mouse number as G, one-way ANOVA followed by Bonferroni's multiple comparisons to Group1@A. test. $F(2, 35) = 3.8, P = 0.0312$, $P_{\text{Group1@A vs. Group1@B}} = 0.0235$, $P_{\text{Group1@A vs. Group2@A}} = 0.0493$). The result of Group2 presented is the same as showed in Fig. 3g&i, it's for comparison here.

h

Figure 25 to reviewers, now is the supplementary figure 17h. h, quantification of occupancy in food zone (Two-way ANOVA followed by Bonferroni's multiple comparisons, $F_{\text{EYFP vs. NpHR}}(1,6) = 1.8, P = 0.2314$, $F_{\text{ON vs OFF}}(1,6) = 9.3, P = 0.0226$, $F_{\text{Interaction}}(1,6) = 1.5, P = 0.2689$, $P_{\text{EYFP ON vs. EYFP OFF}} = 0.6632$; $P_{\text{NpHR ON vs. NpHR OFF}} = 0.0106$). n.s., no significance.

- Language errors should be fixed throughout. E.g., Supp. Fig 8 e.g. “recoding” should be “recording;” Line 46, causally, not casually; Line 714 “carinal” for “cranial”.

Thank you for your careful reading, already fixed. We have checked and revised throughout the manuscript to avoid grammatical errors, typos and the manuscript was refined and polished by a professional English polishing agency.

- Line 27. The interconnections among MTL areas have been known for decades, and hence are not recent.

Thank for the notification, we have revised the text: “*Decades of researches revealed that hippocampus and its adjacent regions, including subiculum, medial and lateral entorhinal cortex (LEC), are tightly connected brain structures*”.

- Figure 3 caption, lines 604-605: EYFP is missing, oChIEF is repeated. “ $P_{\text{iSOS (10Hz,oChIEF) vs. HPC-lesion}} > 0.9999$, $P_{\text{iSOS (10Hz,oChIEF) vs. HPC-lesion}} = 0.0243$ ”.

Thanks! The text was already fixed: “ $P_{\text{HPC-intact vs. HPC-lesion}} = 0.0048$, $P_{\text{iSOS (30Hz,EYFP) vs. HPC-lesion}} > 0.9999$, $P_{\text{iSOS(10Hz,oChIEF) vs. HPC-lesion}} = 0.0243$, $P_{\text{iSOS (30Hz,oChIEF) vs. HPC-lesion}} = 0.0098$ ”.

Reviewer #3 (Remarks to the Author):

Although theories of the hippocampal role for memory assume functional reactivation of the neocortical network during memory retrieval (e.g., Teyler and DiScenna, 1986), our understanding of how the hippocampus coordinates the activities of multiple neocortical areas still remains scarce. The authors found the lateral entorhinal cortex (LEC) mediates the hippocampal-neocortical interaction and coordinates long-range gamma-synchrony among the retrosplenial, visual, and motor cortices. Surprisingly, optogenetically induced gamma-range synchrony of these cortical structures rescued memory impairments in the hippocampus lesioned animals. These findings are strong supports for the memory index theory, which proposes the primary role of the hippocampus to be the reinstatement of cortical activity, and would be of great interest from general readers. I would say this is one of the most interesting papers I read this year, and I believe, if published on Nature Communications, this paper will have a substantial impact in the field. To further improve the quality of the manuscript, I would like the authors to address several concerns.

Thanks for the careful reading and constructive suggestions, especially those positive feedbacks to our major findings. We followed suggestions from all the reviewers and provided additional experiment results in the new version of the manuscript and hope these would improve it.

1. When the authors analyzed coupling between cortical long-range gamma-synchrony and theta rhythm, it is unclear which brain area they used for theta oscillation. Because theta coherence across cortical structure varies depending on the animal's state, behavior, etc (e.g., Young & McNaughton, 2009), it would influence their interpretation of 'coupling' if they choose the reference area randomly.

Thank you for pointing this out. In the practical analysis of the coupling between theta wave and long-range gamma synchrony, the MO-RSC synchrony was aligned to theta wave of MO, the RSC-VIS synchrony was aligned to theta wave of RSC, and the VIS-MO synchrony was aligned to theta wave of VIS. To quantify the effect of theta selection on the result, we generated the spectrogram and calculated synchrony using theta wave of another region (**Figure 26 to reviewers**). We found that the spectrogram and the overall synchrony were extremely close to each other, probably due to the synchronization of theta wave.

To control the data from similar behavior states as much as possible. Not all LFP data was selected for the following data analysis. For naive trials, it is a segment of LFP data from a 3-minute period when the mice were awake in the homecage before spatial memory training (day0). For training trials, from the timepoint of getting off the start block to finally food eating. (For more details, please refer to the answer to comment 3 from review1 and **Figure 3 to reviewers**, on page3). For recall trials (3.5min, 210s), it's a 210s LFP segment from timepoint of getting off the start block to the end of the trial.

However, the moving speed during the exploring was correlated with LFP power and cortical phase synchrony (**Figure 4 to reviewers**, also available in Supplementary Fig. 1e-f). Thus, we

Figure 26 to reviewers only. Theta wave selection has little effect on phase-synchrony spectrogram and the overall synchrony. a-b, When consider theta selection, there are 6 pairs among 3 regions in total. Phase-synchrony spectrograms and overall synchronies of 6 pairs in HPC-intact mice. **a**, Phase-synchrony spectrograms of 6 pairs in 3 kinds of trials. **b**, Comparison of overall synchronies, solid line pairs represent the 3 pairs show in main figure1 while dash line represent the other three pairs. (Pair number: $N_{MO-RSC(\theta)} = 26$, $N_{RSC-VIS(\theta)} = 26$, $N_{VIS-MO(\theta)} = 26$, $N_{RSC-MO(\theta)} = 26$, $N_{VIS-RSC(\theta)} = 26$, $N_{MO-RSC(\theta)} = 26$ from 28 mice) **c-d**, Phase-synchrony spectrogram and overall synchrony of 6 pairs in HPC-lesion mice. (Pair number: $N_{MO-RSC(\theta)} = 11$, $N_{RSC-VIS(\theta)} = 12$, $N_{VIS-MO(\theta)} = 11$, $N_{RSC-MO(\theta)} = 11$, $N_{VIS-RSC(\theta)} = 12$, $N_{MO-RSC(\theta)} = 11$ from 12 mice).

compared the power and synchrony under different moving speeds. We found that low gamma synchrony was not affected by the moving speed. Thus, gamma synchrony is comparable in all three kinds of trials, suggesting it may correlate to the cognitive process. That is also one of the reasons that we focused on the synchrony in the gamma band in the following study, instead of the power of the oscillation.

To make it more clear, we have added some explanatory text in the corresponding part of the method section: “*In the calculation of phase-synchrony spectrogram and overall synchronization, we tested whether the theta selection would affect the final conclusion and found that the theta wave selection had little effect on the phase-synchrony spectrogram and the overall synchrony (data not shown). Therefore, we define alignment to theta as follows: the MO-RSC synchrony*

was aligned to theta wave of MO, the RSC-VIS synchrony was aligned to theta wave of RSC, and the VIS-MO synchrony was aligned to theta wave of VIS. ”

- The authors need to provide a stronger rationale for the construction of egocentric object ratemaps (EORs). In the previous study the authors referenced (Alexander et al., 2020), animals were allowed to explore the environment freely. In the present study, I would imagine the animals' trajectories during the recall trials are more stereotyped due to their better performance to navigate towards the goal location, which potentially results in poor occupancy of angular/distance bins. Did each animal fill all angular/distance bins? If not, the authors need to provide 1) percent occupancies for each animal and 2) statistical support that the observed activities are still reliably tuned to an object.

Thank you for your suggestion, it is a crucial question. In fact, each experimental mouse underwent 4 recall trials in 2 days (missing experimental details now have been added into the method section). Therefore, we pooled 4 trials together in the analysis of egocentric tuning strength, the total exploring time in the context is $3.5 * 4 = 14$ min (similar to the experiment by Alexander et al). Unlike training trials, there was no food in recall trials, once the mice fail to find food in the target region, the mice would travel around in the context and search for it. So, mice's trajectories covered most of the context area (on average, 73.6%, allocentric occupancy) and most of object filled > 90% of the egocentric map (egocentric occupancy), please see the table below (**Figure 27 to reviews**, now integrated into supplementary figure 15b).

To further validate whether these activities are still reliably tuned to an object egocentrically, we tested if the calcium event (CEs) show significant egocentric tuning to non-object regions

Mouse ID	Brain-Region	Allocentric Map Occupancy (%)	Egocentric map occupancy (%)						
			Wall	Food zone	Start block	Block 2	Block 3	Block 4	Block 5
16#	RSC	72.32	100.0	100.0	97.5	99.7	99.9	100.0	100.0
18#	RSC	66.24	100.0	100.0	94.9	99.9	99.8	97.2	99.5
20#	RSC	56.64	99.9	99.9	80.9	97.1	99.2	90.5	99.6
22#	RSC	82.72	100.0	100.0	100.0	99.9	100.0	98.9	100.0
27#	RSC	61.12	100.0	99.9	95.4	100.0	97.8	91.6	100.0
39#	VIS	87.36	100.0	100.0	99.7	100.0	100.0	100.0	100.0
40#	VIS	86.4	100.0	100.0	99.7	98.2	99.9	99.1	100.0
41#	VIS	80.8	99.9	100.0	100.0	99.9	98.8	100.0	100.0
42#	VIS	75.84	100.0	100.0	96.2	99.9	98.9	97.9	99.9
44#	RSC	82.72	100.0	100.0	99.8	100.0	100.0	99.6	100.0
45#	VIS	68.96	100.0	100.0	98.2	99.7	99.7	97.4	98.9
46#	RSC	70.56	100.0	99.4	98.5	99.2	99.9	93.5	99.8
48#	RSC	64.8	100.0	100.0	93.9	98.8	100.0	94.9	99.8
Average	RSC	73.6	100.0	99.9	96.5	99.4	99.5	97.0	99.8

Figure 27 to reviewers (Supplementary Fig. 15b). Allocentric map occupancy and egocentric map occupancy. Green-filled square noted the significant EOR.

(Figure 28 e-f to reviewers). Except for 6 objected regions, 30 non-object regions were defined. Please see the figure below, the result shows that these CEs was more likely show significant tuning to objects but not non-object regions, it supports our idea that activities of labeled neuron show egocentric coding to objects.

As your suggestion, to statistically test our method for detection of significant tunings, a simulation was performed: position of the mice and its head direction, as well as the egocentric tuning and non-egocentric CEs were randomly generated 500 times (same region probability and total time point as real recall trials), followed by a calculation for the detected rate of significant egocentric coding (Figure 28 g-h to reviewers, has been integrated into supplementary figure 16). We found that only need more than 6 egocentric CEs and its portion more than 60% of the total CEs are able to be detected as significant. Moreover, when there was no egocentric CE, only 6.7% of simulations found significant egocentric coding (false positive rate = 6.74%). This data suggested that the method for the detection of significant tunings is statistically reliable.

Figure to reviews 28 (Supplementary Fig. 15e-h). e-f. Activities of labelled neurons are more likely to show significant egocentric coding to objects than to non-object regions. e, The definition of the object position (A-E) and non-object regions (1-30), red numbers indicates all the regions detected as significant. **f,** The comparison of the significant portion between objects and non-object regions. Significant objects portion = $100\% * (a/6)$, Significant non-region portion = $100\% * (b/30)$, here **a** represent the number of significant egocentric coding to objects and **b** represent the number of significant egocentric coding to non-object regions (paired t-test, $n = 13$, $P = 0.0075$, $t(12) = 3.21$). **g-h, Simulation to test whether the egocentric coding significance detection is reliable. g,** A simulation example for one simulation. Mouse position was generated according to real recall occupancy map (Fig. 3h), the head direction was random from $-\pi$ to π radian. Points with calcium event (CEs) include a portion of egocentric tuning CEs to a virtual object (green square and surround green points) and other CEs (black), corresponding head directions were marked by arrows, in total 4200 points (14min * 60 sec * 5fps). **h,** Simulation was repeated 500 times for each pair of total CE number/egocentric tuning CE portion to calculate the rate of ignorance detected. Error bar shows S.E.M.

3. I would like to hear more about the author's thoughts about BMI. I feel the authors wrote BMI without sufficient background. What are the current issues of BMI in application to memory defective diseases? And how does the finding solve them?

Agree, we think we could give some more text on this issue. We have added some content

about BMI in the discussion part: “Our studies showed that brain-machine interface (BMI) devices could improve cognitive functions in healthy people or as a therapy to patients with diseases such as AD. Studies have demonstrated memory enhancement through deep brain stimulation (DBS)¹¹ or transcranial direct current stimulation (tDCS)¹². However, the stimulation settings are highly variable with electrode placements, stimulation waveforms, and spatial-temporal scales¹³. By providing biological insight to the memory process, we suggest novel BMI designs targeting on LEC and proposed specific stimulation protocols as the gamma synchrony. On the other hand, while decoding neural oscillation signals usually employ power or single units¹⁴, instantaneous large-scale multi-pair cortical phase synchrony could be used to decode neural information via new BMI equipment. We propose that the revealed biological mechanism of memory encoding and retrieval could significantly improve the efficiency of BMI for future technologies.”

4. Also, it would be great if the authors provide more discussions on the functional roles of HPC, especially in the context of the Memory Index Theory. This theory assumes the memory index in the hippocampus does not store any information about experiences, which is in line with the authors' findings. If that is the case, what is the contribution of the allocentric spatial map in the hippocampus to episodic memory? I understand addressing this question might not be a scope of the current study, and the findings do not provide a conclusive view, but I think having a deeper discussion on this matter would interest more readers in the broader fields.

Thank you for your suggestion, we agree that our finding is consistent with the memory index theory. Now we provided some discussion on that: “Our discovery on the HPC-regulated distributive cortical engrams shines light on the memory index theory, which proposes that memory is stored in distributed cortical and subcortical modules, while hippocampus registers extrahippocampal network connectivity patterns¹⁵⁻¹⁷. Interestingly, we found HPC/LEC do regulate activities of engrams in multiple cortices via providing γ -synchrony, suggesting the traveling waves in the brain and their synchrony could take a role as the index to register cortical memory units. These observations are also consistent with a recent report that Tanaka and colleagues¹⁸ found the hippocampal CA1 engrams encode the behavior context rather than the specific place information and were coupled to θ rhythm. It is interesting to speculate that allocentric spatial map in place cells and egocentric information coupling to the hippocampal engrams might interact in CA1 to provide integrative information for spatial memory retrieval. In addition, the memory index theory suggests the HPC might not retain and deliver memorized information, but keep the internal index to access those information, which is consistent with our finding that artificially imposed iSOS signal is able to encode and retrieve memory, probably via providing indexing role to the cortical memory units.”

5. This is optional. Related to Fig6, I am very interested in how the egocentric cortical representation is influenced by LEC manipulation.

This is an interesting question. We analyzed our data and found that the number of the significant egocentric coding object decreased when LEC was inhibited. Moreover, the egocentric

maps of this label cortical neurons are trend to be more dispersive (**Figure 29 to reviewers**). This data suggest that the LEC is required for precise egocentric map representation. This part of the data has been put into supplementary figure 18.

Figure to reviews 29 (Supplementary Fig. 18). LEC inhibition distorts egomaps. a, Experimental diagram for LEC inhibition during memory recall in the spatial memory task. Same dataset as figure 6. **b**, number of the significant egocentric coding object with or without LEC inhibition, CNO dosage: 2 mg/kg. (N = 8, paired t-test, $t(7) = 3.4$, $P = 0.0112$). **c**, Two examples of egomaps of the same object with or without LEC inhibition to show distorted egomap. * $P < 0.05$, Error bar shows S.E.M.

Some minor issues:

- Line 99. I think “is essential for the cortical long-range gamma-synchrony” is a bit overstating because HPC-lesion substantially reduced the powers of the oscillations. This sentence needs to be softened, like “is essential for the cortical oscillations.”

We agree, we have revised the sentence to “*is essential for the cortical oscillations.*”

- Line 707-708. “Four screws inserted above the cerebellum were used as reference and ground electrodes during recordings.” Does this mean two of four screws?

Yes, there are four screws were implanted above the cerebellum, two of them were used as reference and the other two were for ground. We drew a small figure to show this, please check the below figure (**Figure 29 to reviewers**). To be clear, we revised this sentence in the method section to: “*Four screws were inserted above the cerebellum, two of them were used as reference and the other two ground electrodes were used as ground. Two references were interconnected and isopotential, two ground screws were also interconnected and isopotential.*”

Figure 29 to reviews only. Positions of implanted electrodes and screws. 4 Screws was implant upon the transverse sinus. We chose transverse sinus for reference screw because the position below is less neuron and more vascular tissue.

3. Line 714. Typo “Carinal” -> “Cranial”

Thanks, already fixed. We have checked and revised throughout the manuscript to avoid grammatical errors, typos. The manuscript was refined and polished by a professional English polishing agency.

I don't see any severe issue in statistical analyses, and the level of the detail is sufficient enough for researchers to reproduce the study (except for a few described above).

Thank you very much for your time to read our response. We also appreciate your careful reading and positive comments!

References

1. Berens, P., Keliris, G.A., Ecker, A.S., Logothetis, N.K. & Tolias, A.S. Feature selectivity of the gamma-band of the local field potential in primate primary visual cortex. *Front Neurosci* **2**, 199-207 (2008).
2. Katzner, S., *et al.* Local origin of field potentials in visual cortex. *Neuron* **61**, 35-41 (2009).
3. Xing, D., Yeh, C.I. & Shapley, R.M. Spatial spread of the local field potential and its laminar variation in visual cortex. *J Neurosci* **29**, 11540-11549 (2009).
4. Buzsaki, G., Anastassiou, C.A. & Koch, C. The origin of extracellular fields and currents--EEG, ECoG, LFP and spikes. *Nat Rev Neurosci* **13**, 407-420 (2012).
5. Zheng, C., Bieri, K.W., Trettel, S.G. & Colgin, L.L. The relationship between gamma frequency and running speed differs for slow and fast gamma rhythms in freely behaving rats. *Hippocampus* **25**, 924-938 (2015).
6. Adhikari, A., Sigurdsson, T., Topiwala, M.A. & Gordon, J.A. Cross-correlation of instantaneous amplitudes of field potential oscillations: a straightforward method to estimate the directionality and lag between brain areas. *J Neurosci Methods* **191**, 191-200 (2010).
7. Hartley, T., Lever, C., Burgess, N. & O'Keefe, J. Space in the brain: how the hippocampal formation supports spatial cognition. *Philos Trans R Soc Lond B Biol Sci* **369**, 20120510 (2014).
8. Igarashi, K.M., Lu, L., Colgin, L.L., Moser, M.B. & Moser, E.I. Coordination of entorhinal-hippocampal ensemble activity during associative learning. *Nature* **510**, 143-147 (2014).
9. Knierim, J.J., Neunuebel, J.P. & Deshmukh, S.S. Functional correlates of the lateral and medial entorhinal cortex: objects, path integration and local-global reference frames. *Philos Trans R Soc Lond B Biol Sci* **369**, 20130369 (2014).
10. Roy, D.S., *et al.* Distinct Neural Circuits for the Formation and Retrieval of Episodic Memories. *Cell* **170**, 1000-1012 e1019 (2017).
11. Suthana, N., *et al.* Memory Enhancement and Deep-Brain Stimulation of the Entorhinal Area. *New England Journal of Medicine* **366**, 502-510 (2012).
12. Antonenko, D., Hayek, D., Netzband, J., Grittner, U. & Flöel, A. tDCS-induced episodic memory enhancement and its association with functional network coupling in older adults. *Scientific Reports* **9** (2019).
13. Mankin, E.A. & Fried, I. Modulation of Human Memory by Deep Brain Stimulation of the Entorhinal-Hippocampal Circuitry. *Neuron* **106**, 218-235 (2020).
14. Flesher, S.N., *et al.* A brain-computer interface that evokes tactile sensations improves robotic arm control. *Science* **372**, 831-836 (2021).
15. Teyler, T.J. & DiScenna, P. The role of hippocampus in memory: A hypothesis. *Neuroscience & Biobehavioral Reviews* **9**, 377-389 (1985).
16. Teyler, T.J. & DiScenna, P. The hippocampal memory indexing theory. *Behavioral Neuroscience* **100**, 147-154 (1986).
17. Goode, T.D., Tanaka, K.Z., Sahay, A. & McHugh, T.J. An Integrated Index: Engrams, Place Cells, and Hippocampal Memory. *Neuron* **107**, 805-820 (2020).
18. Tanaka, K.Z., *et al.* The hippocampal engram maps experience but not place. *Science* **361**, 392-397 (2018).

Reviewers' Comments:

Reviewer #1:

Remarks to the Author:

The authors did a remarkable load of work that significantly improved the original manuscript. I have one remaining concern.

#2. I may miss something. How come optogenetic stimulation did not, but electrical stimulation did show region-specific response? Please explain.

Minor#8. Please specify 'dHPC', instead of 'HPC'.

Thank you for the impressively tremendous work!

Reviewer #2:

Remarks to the Author:

Nature Communications 309301_1

"Acquiring new memories in neocortex of hippocampal-lesioned mice" has been extensively revised for the better, and the new analyses and reported experiments provide compelling evidence that the LEC coordinates synchronized activity in distributed neocortical circuits that are crucial for memory. Although the rebuttal letter states that the manuscript was "refined and polished by a professional English polishing agency," more editing is needed. I've indicated some of the remaining problems below. Beyond these presentation issues and at least one example of unclear or incomplete statistical descriptions (below), the results are quite astonishing. And though I am skeptical that the discoveries would improve cognitive function in victims of Alzheimer's disease, my skepticism could reflect pessimism rather than actual possibilities.

Lines 22 through 27: "confounding hypothesis..." should be rewritten for clarity. Conflicting hypotheses about how the hippocampus contributes to memory storage and retrieval remain. The standard memory consolidation hypothesis proposes..., While the memory indexing theory....

33: ... neurons are identified as elements of memory engrams...

42-43. Please spell out theta and gamma, and include the frequency ranges (which differ in the human and animal literature).

60: first we surveyed...

Including figure numbers in the manuscript would help the reviewers.

61 and captions for figure 1 and supplemental figure 1: More details are required. Figure 1 shows a single starting point, but the text does not make clear if all trials were started from the same point.

68: Please refer here to the evidence described in the next section that the task is hippocampus-dependent.

78: between each cortical area of interest

87: syntax errors. "Because the hippocampus is closely involved..."

97: were substantially reduced compared to those in...

97-98: how does figure 2F show the difference between intact and brain-damaged mice?

111-112: Does "not in the reversed order" refer to brain region, oscillation frequency, or both?

147-148: Response latencies were similar across cortical regions

352-354, supplemental figure 18: What is the "number of the significant egocentric coding object"?

364-366. The sentence needs to be rewritten.

357-363. The paragraph should be reorganized so that mouse behavior and neuronal activity are described separately and more clearly.

Supplemental figure 19b seems to show that LEC inhibition impairs learning, and the statistic reported for the two way ANOVA should include both the main effects of group and training day along with the interaction. The caption for supplemental figure 19d states, "Neither labeled RSC nor VIS neurons showed selectivity to sandbox." This would be easier to understand if figure 5B were called out.

736. The figure 5 caption should move this sentence to the end of line 733.

Lines 817-830: the paragraph switches tense inconsistently and should be corrected.

Line 819: ...to forage for sunflower seeds...

Line 820: The sandboxes A and B...

Line 821: There were no cues on the surrounding walls...

Supplementary figure 12f, why are the dashed lines colored red in the left two panels and blacken the right to?

1100-1101: The same criterion for detecting significant egocentric coding was employed in LEC inhibition by CN or PBS ... employed is misspelled.

1114-1115. Please correct the sentence, it's difficult to understand as is.

Reviewer #3:

Remarks to the Author:

In the revision, the authors provided a significant amount of data. Furthermore, I appreciate that the authors conducted an additional experiment examining the impact of LEC manipulation over the egocentric cortical representation. The result is very interesting. I also appreciate their new discussion sections and enjoyed reading them.

However, their revisions do not fully address my previous concerns (#1 & 2).

- Theta selection (comment #1)

"We found that the spectrogram and the overall synchrony were extremely close to each other, probably due to the synchronization of theta wave."

This is surprising. In Figure 26 to reviewers only, I see almost perfect fits in phase-synchrony when the two areas are swapped. Moreover, the traces of the averaged synchrony from each behavioral condition across different pairs of brain areas also look very similar. This phenomenon is unlikely to happen given the nature of cortical oscillations and strongly suggests that the authors are looking at the effect of volume conductance, noises, or both, as reviewer #1 pointed out.

- Occupancy of the egocentric map (comment #2)

Figures 27 & 28 to reviewers only are also puzzling. The authors found 73.6% of allocentric occupancy on average, which is not very high and means 26.4% of locations are not visited. Nevertheless, the animals could occupy more than 90% of egocentric maps for all the defined objects. As one of the attempts to justify the statistical significance of this measure, they have calculated egocentric tuning to non-object regions covering the entire arena. How is this possible? If the animal did not visit 1/4 of the locations, the authors should have encountered a serious

issue for occupying the egocentric map of these non-object regions.

Reviewer #1 (Remarks to the Author):

The authors did a remarkable load of work that significantly improved the original manuscript. I have one remaining concern.

Thank you very much for your time and suggestions, which really help us to improve the manuscript.

- #2. I may miss something. How come optogenetic stimulation did not, but electrical stimulation did show region-specific response? Please explain.

Thanks for the notification. Evoked potential by the electrical stimulation was confined into the local region, indicating the volume conduction effected would be acceptable in our recording setup (Supplementary figure 7). There are several kinds of optogenetic stimulation in this study, the first one is that we directly activated cell bodies with optical fibers and recorded the evoked potential in various cortical regions (**Figures 1a to reviewers**). Evoked potential can be detected in all 3 brain regions, this is reasonable because the LEC sends fiber to all these regions. In the rescue experiment, to involve more brain regions, we open a cranial window (d = 6 mm) and activated cortical fibers on the surface (**Figures 1b to reviewers**), this would be expected to show no region-specific response because fibers in all regions were activated. To be more clear, we add a small figure on Fig.3a to show the size of the cranial window now.

To investigate whether the synchronization is the essential one or just the activation of the fiber rescued the memory deficit in HPC-lesioned mice, we specifically activated two regions simultaneously or asynchronously (**Figures 1c to reviewers**). To achieve region specificity in this experiment, two optical fibers (diameter = 200 μ m) with a low numerical aperture (N.A. = 0.37) were placed right on the surface of the cortex to confine the emission light to the regions, and we

a, Figure 2I. 470 nm laser-activated neurons expressing oChIEF-mCherry in LEC induced synchronization oscillatory signals in multiple cortical areas simultaneously. **b, Figure 3a,** Scheme of the co-activation LEC fiber. The iSOS were induced by LED on cortical surface to activate oChIEF-expressing widespread axons from LEC L5. **c, Figure 4a.** LEC-axons in two cortical areas were activated by two independent fibers with either synchronized signal or asynchronous signal. **d, Supplementary figure 11c-d.** Asynchronous stimulation was able to induce a large decrease of cortical synchrony.

found that asynchronous stimulation was able to induce a rapid fall of the synchrony, indicating the two lasers were able to evoke potential separately (**Figures 1d to reviewers**). To be more clear, we add some explanation statements in Method: *“To ensure we can activate LEC cortical fibers separately, two optical fibers (diameter = 200 μ m) with a low numerical aperture (N.A. = 0.37) were placed right on the surface of the cortex to confine the emission light to the local region.*

2. **Minor#8. Please specify 'dHPC', instead of 'HPC'.**

Thanks for your kind reminder. To describe it more precisely, we have replaced 'HPC' with 'dHPC' in the result of LFP recording (only dorsal hippocampus was recorded) and kept 'HPC' in the result of the hippocampal lesion (both ventral and dorsal parts were damaged).

Thank you for the impressively tremendous work!

Agree! A great deal of work is the way to the true answer for scientific problems.

Reviewer #2 (Remarks to the Author):

“Acquiring new memories in neocortex of hippocampal-lesioned mice” has been extensively revised for the better, and the new analyses and reported experiments provide compelling evidence that the LEC coordinates synchronized activity in distributed neocortical circuits that are crucial for memory. Although the rebuttal letter states that the manuscript was “refined and polished by a professional English polishing agency,” more editing is needed. I’ve indicated some of the remaining problems below. Beyond these presentation issues and at least one example of unclear or incomplete statistical descriptions (below), the results are quite astonishing. And though I am skeptical that the discoveries would improve cognitive function in victims of Alzheimer’s disease, my skepticism could reflect pessimism rather than actual possibilities.

Thank you very much for your careful reading and practical suggestions, which do help us to improve our manuscript. We revised our expressions and statements accordingly and highlighted them in the revised manuscript.

1. Lines 22 through 27: “confounding hypothesis...” should be rewritten for clarity. Conflicting hypotheses about how the hippocampus contributes to memory storage and retrieval remain. The standard memory consolidation hypothesis proposes..., While the memory indexing theory....

Thanks! For a better expression, we have improved these sentences according to your kind advice. “*Conflicting hypotheses about how the interactions between hippocampus and neocortex contribute to memory storage and retrieval remain. The standard memory consolidation hypothesis proposes hippocampus transiently stores the memorized information, while the memory indexing theory argues that hippocampus only maintains the pointers to memories stored in the neocortex.*”

2. 33: ... neurons are identified as elements of memory engrams...

Fixed! The sentence has been rewritten.

3. 42-43. Please spell out theta and gamma, and include the frequency ranges (which differ in the human and animal literature).

Thank you! To be more clear, we spell out theta and gamma throughout the maintext (still Greek letter in the figure to save room, but indicated in the legend), and provided clear frequency ranges accordingly.

4. 60: first we surveyed... Including figure numbers in the manuscript would help the reviewers.

Agree. We changed the expression and add the figure index after this sentence.

5. 61 and captions for figure 1 and supplemental figure 1: More details are required. Figure 1 shows a single starting point, but the text does not make clear if all trials were started from the same point.

Thanks for your kind reminder. Mice start exploring from the starting point in all trials. To

clarify this, detailed descriptions were added in the text and captions for figure 1 and supplemental figure 1

6. 68: Please refer here to the evidence described in the next section that the task is hippocampus-dependent. “Line 68: We recorded the local field potential (LFP) signals simultaneously from superficial layers of multiple cortices..... to study the coordination between motor, visual and spatial information related regions in the HPC-dependent task.”

Thank you for the reminder, it's unable to know whether this task is HPC-dependent or not before describing the experimental result in figure2, to be more logical, we revised the sentence: “... to study the coordination between motor, visual and spatial information related regions *in this task*.”

7. 78: between each cortical area of interest;
87: syntax errors. “Because the hippocampus is closely involved...;
97: were substantially reduced compared to those in...

Thanks! Syntax errors have been fixed.

8. 97-98: how does figure 2F show the difference between intact and brain-damaged mice?

Maybe you are referring to lines 98-99 about the figure 2F: “*Prominently, learning-induced θ - γ coupling was abolished in HPC-lesioned mice (Fig. 2f, Supplementary Fig. 3)*”, here we are comparing the theta-gamma coupling between hippocampus lesion mice (Fig. 1f) and intact mice (Fig.2f). To make it more clear, we changed the statement to: “...*HPC-lesioned mice (Fig. 1f vs. Fig. 2f, Supplementary Fig. 3 for all kinds of trials)*.”

9. 111-112: Does “not in the reversed order” refer to brain region, oscillation frequency, or both?
Original sentence: 111-112: Thus, the θ rhythm in LEC could regulate cortical synchrony across multi-sensory cortices, but not in the reversed order.

The order refers to brain regions. The sentence was revised to: “*Thus, the theta rhythm in LEC could regulate cortical synchrony across multi-sensory cortices, but not the cortical theta regulated the LEC gamma.*”

10. 147-148: Response latencies were similar across cortical regions.

Revised, thanks for the concise expression.

11. 352-354, supplemental figure 18: What is the “number of the significant egocentric coding object”?

It means the number of objects which were encoded in an egocentric manner. To reduce misunderstanding, we have rewritten this sentence into “*We found that the number of objects which were encoded in an egocentric manner was decreased after LEC inhibition (Supplementary Fig.18a-b)*”.

12. 364-366. The sentence needs to be rewritten. The original 364-366: “Our data indicated that LEC is not only critical for memory engram function during encoding but also the reactivation of cortical

engrams occurred during the memory retrieval period, which was mediated by LEC-mediated γ -synchrony.”

Apologize for the lengthy expression, we have rewritten this sentence: “*Our data indicated that LEC-mediated gamma synchrony is critical for both memory encoding and engram reactivation during memory retrieval.*”

13. 357-363. The paragraph should be reorganized so that mouse behavior and neuronal activity are described separately and more clearly. Original line 357-363: “*Importantly, the LEC activity is also required for the formation of memory. When mice were trained with induced-LEC inhibition for each training day in the spatial task, we labeled the learning activated cortical neurons on day 4 and tested their functions. Interestingly, mice showed significantly reduced performance in recall trials (Supplementary Fig. 19, see details in Method: “LEC Inhibition During Spatial Memory Encoding”). And those labeled neurons were not activated by the training context, indicating the LEC activity was essential for the formation of context-selective cortical neurons (Supplementary Fig. 19d).*”

Thanks for the suggestion. We have improved this paragraph for a better logical organization: “*Importantly, we found LEC activity is essential to the formation of spatial memory. We inhibited LEC activity throughout all the training days of the spatial memory task, and mice showed significantly reduced performance in recall trials (Supplementary Fig. 19, see details in Method: “LEC Inhibition During Spatial Memory Encoding”). Interestingly, we also labeled the learning activated cortical neurons on day 4 and found they were not activated much in the recall trial, indicating the LEC activity was essential for the formation of context-selective cortical neurons (Supplementary Fig. 19d).*”

14. Supplemental figure 19b seems to show that LEC inhibition impairs learning, and the statistic reported for the two-way ANOVA should include both the main effects of group and training day along with the interaction. The caption for supplemental figure 19d states, “Neither labeled RSC nor VIS neurons showed selectivity to sandbox.” This would be easier to understand if figure 5B were called out.

Thanks for your advice, the detail statistic was reported in the legend now: “**b**, *Learning curves. (LEC inhibition: N = 5, Normal mice group = 24, two-way ANOVA, Group factor: $F(1, 27) = 0.9259$, $P = 0.3445$; Time factor: $F(3, 81) = 22.91$, $P < 0.0001$; Interaction: $F(3, 81) = 0.3704$, $P = 0.7746$, Bonferroni’s post-hoc test, $P_{\text{Day1}} = 0.9976$, $P_{\text{Day2}} > 0.9999$, $P_{\text{Day3}} > 0.9999$, $P_{\text{Day4}} > 0.9999$).*”

As you suggested, the activity difference of labeled cortical neurons after LEC inhibition was also emphasized by adding a Fig.5B notification into the corresponding legend: “*Comparing with context-selective RSC and VIS activity in LEC intact mice (Fig. 5B), neither labeled RSC nor VIS neurons showed context selectivity in LEC inhibition mice.*”

15. 736. The figure 5 caption should move this sentence to the end of line 733.

We have rearranged the position of this sentence to make the expression better.

16. Lines 817-830: the paragraph switches tense inconsistently and should be corrected;

Line 819: ...to forage for sunflower seeds....;

Line 820: The sandboxes A and B...;

Line 821: There were no cues on the surrounding walls...

Thanks for your advice, we have improved the expression accordingly.

17. Supplementary figure 12f, why are the dashed lines colored red in the left two panels and blacken the right to?

Apologize for the mistake. We have depicted them with the same color. The average calcium curves were plotted in black and the standard deviations were plotted in orange (**Figure 2 to reviewers**).

f

Figure to reviewers 2. Revised supplementary figure 12f.

18. 1100-1101: The same criterion for detecting significant egocentric coding was employed in LEC inhibition by CNO or PBS ... employed is misspelled.

Fixed! We have corrected the spelling.

19. 1114-1115. Please correct the sentence, it's difficult to understand as is. The original sentence: "We found that only need more than 6 egocentric CEs and its portion more than 60% of the total CEs are able to be detected as significant."

Thanks! To improve the sentence, we revised the sentence into: "*Only 6 (or > 60% of total) CEs that always fire in the same position relative to the animal are able to make a neuron population detected as having significant egocentric tuning (Supplementary Fig 15h).*"

Reviewer #3 (Remarks to the Author):

In the revision, the authors provided a significant amount of data. Furthermore, I appreciate that the authors conducted an additional experiment examining the impact of LEC manipulation over the egocentric cortical representation. The result is very interesting. I also appreciate their new discussion sections and enjoyed reading them. However, their revisions do not fully address my previous concerns (#1 & 2).

Thank you very much for your positive evaluations and useful comments on our work. To further improve the manuscript, we provided some additional data and statements to address these concerns.

1. - Theta selection (comment #1). “We found that the spectrogram and the overall synchrony were extremely close to each other, probably due to the synchronization of theta wave.” This is surprising. In Figure 26 to reviewers only, I see almost perfect fits in phase-synchrony when the two areas are swapped. Moreover, the traces of the averaged synchrony from each behavioral condition across different pairs of brain areas also look very similar. This phenomenon is unlikely to happen given the nature of cortical oscillations and strongly suggests that the authors are looking at the effect of volume conductance, noises, or both, as reviewer #1 pointed out.

Thank you very much for asking this valuable question. In fact, when we estimate the overall synchrony across each behavioral condition, the final overall synchronies were the average of all mice and pairs (**Figure 2a to reviewers**, see method section for detail and comparison between different synchrony calculation methods). Importantly, over 1000 theta cycles from a trial were used to calculate the spectrogram, leading to a final result largely depending on the overall phase-difference across all timepoints but not the theta. Also, the dynamics and the differences between pairs were regardless, so you would not see a significant change when the two areas are swapped. When looking closer into these data, we found that the overall synchronies are different and vary across mice, especially in the gamma band (**Figure 2b to reviewers**).

We observed the dynamics of the power of each channel, if the volume conductance was dominant in these data or largely contaminated by noise, it's expected to see that powers of these channels would covary as the volume conductance effect nearly instantaneously. However, the power of each cortical region is neither identical nor depends on others, suggesting their diversity and independence (**Figure 2c-d to reviewers**).

Furthermore, we investigated the dynamics of synchrony of all pairs (using a 2-second moving window to do the same synchrony estimation) and compared the difference between two theta selections and the difference between pairs (**Figure 2e to reviewers**). We found that there are changes up to 0.1, but the difference has been erased when the differences were averaged across all time points (**Figure 2f to reviewers**). The differences between pairs even show more significant changes up to 0.5 (**Figure 2g to reviewers**). In addition, artificially induced current sink in one of five recorded regions has little effect on other channels (Supplementary Fig. 7), which is consistent with report literatures^{1, 2}.

Figure 4 to reviewers. The diversity and dynamics of LFP power and synchrony. **a**, A brief demonstration for calculation of the overall synchrony. f , frequency, t , time, n , the number of theta troughs, $P_{\text{electrod}1}(f, t)$ and $P_{\text{electrod}2}(f, t)$ corresponding the phase for frequency f at time t of two electrodes. k is the iteration of theta trough ID. **b**, Diversity of synchronies. 3 pairs of synchronies from each mouse ($n = 22$). One color represent one mouse. **c**, Dynamics of LFP power. Synchronies were calculated with 2s window and 0.2 step. 60s data from a training trial of a hippocampus intact mouse. **d**, Filtered theta traces of 3 regions. The envelop shows the instantaneous amplitude extract from Hilbert transform. Here 30 seconds is 0-30 seconds of panel c. **e**, Dynamics of LFP phase synchrony. Line figures on the right are the average synchrony across all time point. Red, selction1, Blue selection2. **f**, The difference between two selections. Line figures on the right are the average difference across all time point. **g**, Difference between three pairs. Line figures on the right are the average difference across all time point.

The fact that the synchrony is correlated to the firing memory-related cell, suggesting that the synchrony dynamics were not a stochastic process. Taken these data together, we believed that the contamination from volume conductance would be low and acceptable.

2. - Occupancy of the egocentric map (comment #2). Figures 27 & 28 to reviewers only are also puzzling. The authors found 73.6% of allocentric occupancy on average, which is not very high and means 26.4% of locations are not visited. Nevertheless, the animals could occupy more than 90% of egocentric maps for all the defined objects. As one of the attempts to justify the statistical significance of this measure, they have calculated egocentric tuning to non-object regions covering the entire arena. How is this possible? If the animal did not visit 1/4 of the locations, the authors should have encountered a serious issue for occupying the egocentric map of these non-object regions.

Thank you for this notification. This is not intuitive, but the big egocentric occupancy is actually can be detected in the case with low allocentric occupancy because the animal can stay in a small region but with different head directions so that their sight could cover larger areas. To make it more clear, here we present a small video to show the way that we calculate the allocentric and egocentric occupancy, and the egocentric tuning to non-object regions (**Movie 1 to reviewers**). This is an example analysis of a recall trial from one mouse (data of 4 trials of each mouse were pooled together in the real calculation), this animal explored only 10.6% of the context but caused egocentric maps with over 80% occupancy. To calculate the egocentric tuning to non-object regions, regions without objects underwent the same procedure as the calculation we did for the object region. To be more clear, this video was added into the supplement as a supplementary movie 1.

Movie 1 to reviewers (Supplementary Movie 1). Generation of egocentric map. If the video is unable to be play on this PDF, please see the Supplementary Movie 1. The red square represents the biggest block, the blue square represents a non-object region. The green square represents the food zone. The video is made from a 3.5 minutes recall trial.

References

1. Katzner, S., *et al.* Local origin of field potentials in visual cortex. *Neuron* **61**, 35-41 (2009).
2. Dubey, A. & Ray, S. Cortical Electrocorticogram (ECoG) Is a Local Signal. *The Journal of Neuroscience* **39**, 4299-4311 (2019).

Reviewers' Comments:

Reviewer #1:

Remarks to the Author:

The authors made a significant improvement to address that cortical gamma synchrony orchestrates memory encoding in a hippocampal theta-dependent mechanism. However, the manuscript is still poorly written, and many readers may have a hard time understanding this elegant study. For example, the first paragraph of the discussion comprehensively summarizes the essence of the work, but the abstract does not do its job.

Reviewer #3:

Remarks to the Author:

In this round of revision, the authors provided additional analyses to address my previous concerns. They are convincing enough, and I really appreciate their efforts to clarify my points. If I can request, I think "the supplementary movie 1 for the reviewer" would be a great visual material to avoid readers' confusion and should be included as one of the supplemental materials. Either way, I believe their results are solid and very interesting. It is worth publishing in its present form.

Reviewer #1 (Remarks to the Author):

The authors made a significant improvement to address that cortical gamma synchrony orchestrates memory encoding in a hippocampal theta-dependent mechanism. However, the manuscript is still poorly written, and many readers may have a hard time understanding this elegant study. For example, the first paragraph of the discussion comprehensively summarizes the essence of the work, but the abstract does not do its job.

Thanks for your suggestion, we asked a colleague to review our manuscript. And we revised the abstract accordingly *“The hippocampus interacts with the neocortical network for memory retrieval and consolidation. Here, we found the lateral entorhinal cortex (LEC) modulates learning-induced cortical long-range gamma synchrony (20 - 40Hz) in a hippocampal-dependent manner. The long-range gamma synchrony, which was coupled to the theta (7 - 10Hz) rhythm and enhanced upon learning and recall, was mediated by inter-cortical projections from layer 5 neurons of the LEC to layer 2 neurons of the sensory and association cortices. Artificially induced cortical gamma synchrony across cortical areas significantly improved memory encoding in hippocampal lesioned mice for originally hippocampal-dependent tasks. Mechanistically, we found that activities of cortical c-Fos labeled neurons, which showed egocentric map properties, were modulated by LEC-mediated gamma synchrony during memory recall, implicating a role of cortical synchrony to generate an integrative memory representation from disperse features. Our findings reveal the hippocampal mediated organization of cortical memories and suggest a brain-machine interface approach could be a potential therapy for memory defective diseases.”*.

Reviewer #3 (Remarks to the Author):

In this round of revision, the authors provided additional analyses to address my previous concerns. They are convincing enough, and I really appreciate their efforts to clarify my points. If I can request, I think “the supplementary movie 1 for the reviewer” would be a great visual material to avoid readers' confusion and should be included as one of the supplemental materials. Either way, I believe their results are solid and very interesting. It is worth publishing in its present form.

Thank you very much for your comment, which really helps us to improve the manuscript. Yes, the supplementary movie 1 for the reviewer is included in the supplementary information as Supplementary Movie 1 now.